# Persistence in risk and effect of COVID-19 vaccination on long-term health consequences after SARS-CoV-2 infection

Ivan Chun Hang Lam [1,15], Ran Zhang[2,15], Kenneth Keng Cheung Man [1,3,4,5], Carlos King Ho Wong [1,2,3,6], Celine Sze Ling Chui[3,7,8,9], Francisco Tsz Tsun Lai [1,2,3,9], Xue Li [1,3,9,10], Esther Wai Yin Chan [1,3,11,12], Chak Sing Lau [13], Ian Chi Kei Wong [1,3,9,14,16] ✉ & Eric Yuk Fai Wan [1,2,3,9,16] ✉

The persisting risk of long-term health consequences of SARS-CoV-2 infection and the protection against such risk conferred by COVID-19 vaccination remains unclear. Here we conducted a retrospective territory-wide cohort study on 1,175,277 patients with SARS-CoV-2 infection stratified by their vaccination status and non-infected controls to evaluate the risk of clinical sequelae, cardiovascular and all-cause mortality using a territory-wide public healthcare database with population-based vaccination records in Hong Kong. A progressive reduction in risk of all-cause mortality was observed over one year between patients with SARS-CoV-2 infection and controls. Patients with complete vaccination or have received booster dose incurred a lower risk of health consequences including major cardiovascular diseases, and all-cause mortality than unvaccinated or patients with incomplete vaccination 30-90 days after infection. Completely vaccinated and patients with booster dose of vaccines did not incur significant higher risk of health consequences from 271 and 91 days of infection onwards, respectively, whilst un-vaccinated and incompletely vaccinated patients continued to incur a greater risk of clinical sequelae for up to a year following SARS-CoV-2 infection. This study provided real-world evidence supporting the effectiveness of COVID-19 vaccines in reducing the risk of long-term health consequences of SARS-CoV-2 infection and its persistence following infection.

Since the outbreak of the Coronavirus disease 2019 (COVID-19) pandemic caused by the SARS-CoV-2 virus in late 2019, substantial research has been undertaken to uncover the health consequences associated with SARS-CoV-2 infection. The current body of evidence has reported that patients with SARS-CoV-2 infection may incur a greater risk of acute and post-acute health consequences involving multiple organ systems and associated mortality following SARS-CoV-2 infection[1–3].

A universal definition of post-COVID-19 condition remains to be determined owing to the discrepancy in definition published by different healthcare regulatory bodies[4]. Nevertheless, the current literature referred to clinical presentations develop within 30 days of initial infection as acute clinical sequelae, while complications that developed or persisted beyond the acute phase of SARS-CoV-2 infection were referred to as post-acute clinical sequelae[5]. Despite the current evidence suggesting that most patients may recover from SARS-CoV-2 infection within two to four weeks of symptoms appearance, the increased risk of incident cardiovascular, neurological, psychiatric diseases, diabetes and all-cause mortality was shown to persist for up to two years[6–10]. Patients with a severe SARS-CoV-2 infection or

critically ill patients are at particular risk of developing long-term adverse outcome beyond their acute infection. Nevertheless, evidence emerged from the earlier stage of the pandemic were limited by the selection of under-represented samples as the study population[11]. Moreover, findings reported from the current body of literature were based largely on the assumption of a constant risk increase of clinical sequelae over a prolonged duration which may not be able to account for the change in risk over time, thus hindering the representativeness on the burden of the long-term health consequences of SARS-CoV-2 infection. Such speculation was evident from the gradual improvement in pulmonary function in most patients who recovered from severe SARS-CoV-2 infection over a three-monthly observation period for 12 months[12]. Meanwhile, a recent study in Israel have reported a considerably reduced risk of PASC towards the later period of SARS-CoV-2 infection over a course of one year amongst patients with mild infection[13].

Shortly after the outbreak of the pandemic, the global initiative to develop vaccines against SARS-CoV-2 infection followed by the international vaccination campaign has been shown to successfully reduce the risk of primary infection, disease severity and hospitalization associated with SARS-CoV-2 infection[14]. A two dose vaccination regimen was initially recommended for the vast majority of brand of COVID-19 vaccines including the BioNtech and CoronaVac offered in Hong Kong based on the efficacy profile in preventing primary infection established from earlier clinical trials[15,16]. Nevertheless, a third and even fourth booster dose of the vaccine was later introduced in certain countries to restore immunity within the population in the face of the Omicron variant of SARS-CoV-2[17,18]. Despite the cumulative evidence of the covid-19 vaccines' ability to reduce disease burden during acute infection, its effect in preventing the adverse outcome in the post-acute phase of SARS-CoV-2 infection remained largely unknown owing to the inconsistent findings from the existing studies[19,20]. While a protective effect of COVID-19 vaccination on incident health outcomes has been reported in several population and community-based studies, the extent of risk reduction, especially from the booster dose of vaccines remains to be evaluated[21,22].

This population-based study aims to evaluate the progressive risk of health consequences associated with SARS-CoV-2 infection over 1 year and compare the risk and persistence of such risk differences between patients of different COVID-19 vaccination statuses.

## Results

A total of 1,175,277 patients with SARS-CoV-2 infection were identified in this study, of those, 124,443, 101,379, 457,896, and 491,559 patients were unvaccinated, had one, two and three or more doses of COVID-19 vaccination record prior to infection, respectively (Fig. 1). The median time between the latest dose of vaccination and SARS-CoV-2 infection for patients who have received one, two and three or more doses of COVID-19 vaccination was 18 (interquartile range 12–42), 175 (119–208) and 101 (37–170) days, respectively. The median follow-up period for controls, patients with SARS-CoV-2 who were unvaccinated, have received one, two and three or more doses of COVID-19 vaccination were 318 (173–329), 320 (253–331), 324 (312–329), 325 (312–330), and 171 (135–316) days, respectively. The baseline characteristics before and after weighting were summarized in Tables 1 and 2, respectively. The SMDs of all baseline characteristics after weighting were less than 0.1, indicating a good balance between groups of patients with different vaccination statuses. The number of patients at risk at each observation period between days 0–30, 31–90, 91–180, 181–270, 271–365 were 2,815,023, 2,801,396, 2,764,243, 2,119,507, and 1,876,858, respectively.

Overall, a graded reduction in the risk of clinical sequelae among infected patients was observed over time and with the increased number of doses of COVID-19 vaccines received prior to infection. Compared to non-infected controls, patients with SARS-CoV-2 infection were observed to incur a greater risk of clinical sequelae including major cardiovascular diseases [HR dose=0: 4.64 (4.00, 5.38); dose=1: 3.13 (2.60, 3.76); dose=2: 2.53 (2.21, 2.89); dose ≥3: 1.99 (1.72, 2.29)], and all-cause mortality [HR dose=0: 18.89 (18.07, 19.74); dose=1: 8.96 (8.46, 9.48); dose=2: 3.95 (3.71, 4.20); dose ≥3: 1.74 (1.50, 2.02)] across patients with different COVID-19 vaccination status during the acute phase of infection. A graded decrease in risk was

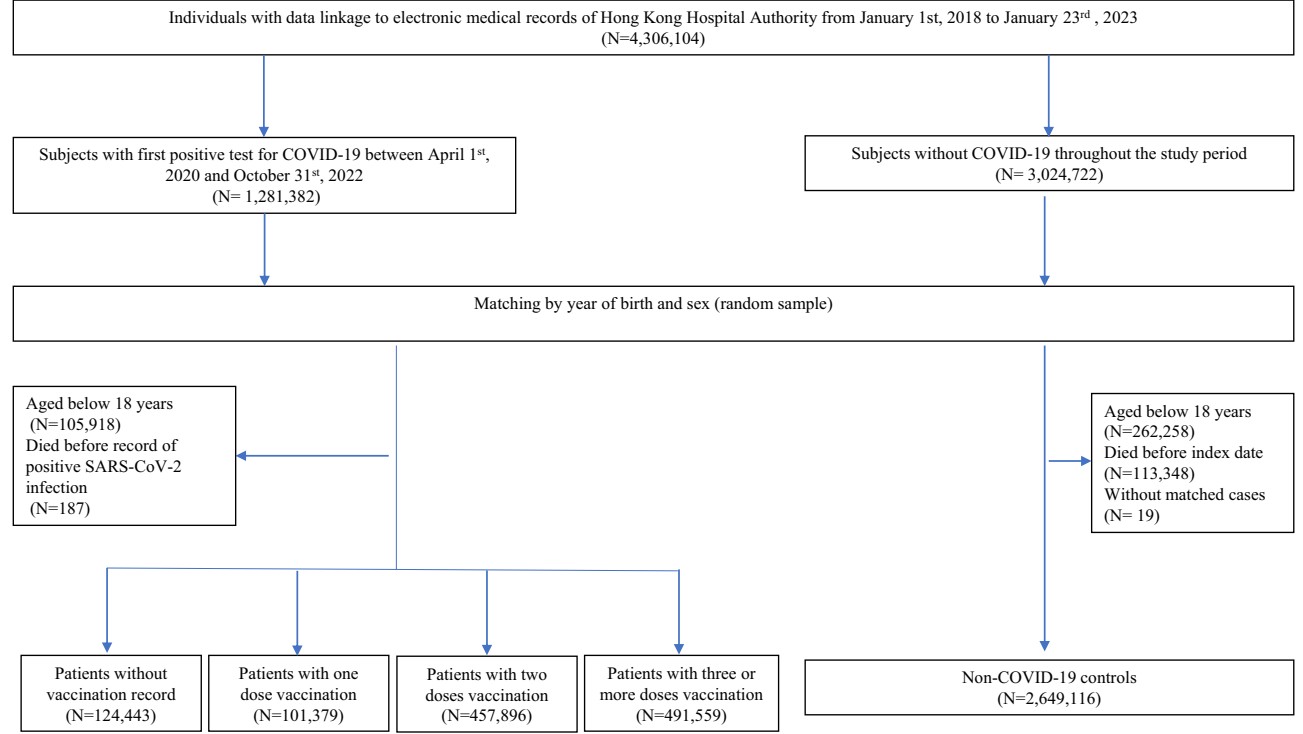

**Fig. 1 | Flowchart for identifying study population.**

**Table 1 | Baseline characteristics of patients with COVID-19 stratified by vaccine status and controls from the Hong Kong Hospital Authority before propensity-score weighting**

| Baseline characteristics | Hong Kong Hospital Authority (April 1, 2020 to October 31, 2022) | | | | | |
| --- | --- | --- | --- | --- | --- | --- |
| | Controls (N = 2,649,116) | COVID-19 | | | | SMD |
| | | Unvaccinated (N = 124,443) | Incomplete vaccination (N = 101,379) | Fully vaccinated (N = 457,896) | Received booster dose (N = 491,559) | |
| | N/mean (%/SD) | N/mean (%/SD) | N/mean (%/SD) | N/mean (%/SD) | N/mean (%/SD) | |
| Age (years) | 54.6 (18.17) | 60.2 (21.93) | 59.6 (19.88) | 50.2 (17.70) | 52.1 (16.07) | 0.293 |
| Sex, male | 1,158,800 (43.7) | 55,091 (44.3) | 44,557 (44.0) | 201,942 (44.1) | 227,573 (46.3) | 0.022 |
| Charlson Comorbidity Index | 1.67 (1.80) | 2.61 (2.53) | 2.38 (2.20) | 1.33 (1.69) | 1.35 (1.57) | 0.354 |
| **Pre-existing morbidities** | | | | | | |
| Respiratory disease | 48,373 (1.8) | 5942 (4.8) | 3759 (3.7) | 8285 (1.8) | 7877 (1.6) | 0.096 |
| Diabetes | 305,099 (11.5) | 21,585 (17.3) | 18,407 (18.2) | 49,344 (10.8) | 48,954 (10.0) | 0.133 |
| Cardiovascular disease | 198,820 (7.5) | 24,098 (19.4) | 14,776 (14.6) | 27,798 (6.1) | 27,318 (5.6) | 0.229 |
| Myocardial infarction | 17,961 (0.7) | 3019 (2.4) | 1443 (1.4) | 2592 (0.6) | 2256 (0.5) | 0.085 |
| Chronic heart failure | 24,386 (0.9) | 5846 (4.7) | 2,613 (2.6) | 3518 (0.8) | 2455 (0.5) | 0.139 |
| Peripheral vascular disease | 6899 (0.3) | 1185 (1.0) | 574 (0.6) | 859 (0.2) | 749 (0.2) | 0.057 |
| Cerebrovascular disease | 91,852 (3.5) | 12,920 (10.4) | 7461 (7.4) | 12,618 (2.8) | 10,975 (2.2) | 0.181 |
| COPD | 48,202 (1.8) | 5900 (4.7) | 3742 (3.7) | 8265 (1.8) | 7868 (1.6) | 0.096 |
| Dementia | 7068 (0.3) | 2795 (2.2) | 1359 (1.3) | 1308 (0.3) | 581 (0.1) | 0.108 |
| Paralysis | 5364 (0.2) | 1105 (0.9) | 485 (0.5) | 720 (0.2) | 499 (0.1) | 0.058 |
| Diabetes without chronic complication | 291,744 (11.0) | 19,919 (16.0) | 17,417 (17.2) | 47,374 (10.3) | 47,199 (9.6) | 0.123 |
| Diabetes with chronic complication | 19,421 (0.7) | 2523 (2.0) | 1477 (1.5) | 2974 (0.6) | 2562 (0.5) | 0.071 |
| Chronic kidney disease | 29,575 (1.1) | 4957 (4.0) | 2740 (2.7) | 4305 (0.9) | 3862 (0.8) | 0.113 |
| Mild liver disease | 3657 (0.1) | 523 (0.4) | 255 (0.3) | 608 (0.1) | 493 (0.1) | 0.031 |
| Moderate-severe liver disease | 2420 (0.1) | 396 (0.3) | 168 (0.2) | 436 (0.1) | 315 (0.1) | 0.028 |
| Ulcers | 25,808 (1.0) | 2754 (2.2) | 1722 (1.7) | 4182 (0.9) | 3949 (0.8) | 0.061 |
| Rheumatoid arthritis and other inflammatory polyarthropathies | 10,659 (0.4) | 912 (0.7) | 626 (0.6) | 1816 (0.4) | 1858 (0.4) | 0.025 |
| Acquired immune deficiency syndrome (AIDS) | 6 (0.0) | 0 (0.0) | 0 (0.0) | 0 (0.0) | 0 (0.0) | 0.001 |
| Malignancy | 75,108 (2.8) | 7166 (5.8) | 4,994 (4.9) | 11,615 (2.5) | 11,003 (2.2) | 0.098 |
| Metastatic solid tumor | 11,699 (0.4) | 1698 (1.4) | 937 (0.9) | 1771 (0.4) | 1324 (0.3) | 0.063 |
| **Medication history** | | | | | | |
| Renin–angiotensin system agents | 370,280 (14.0) | 27,128 (21.8) | 22,749 (22.4) | 59,417 (13.0) | 61,874 (12.6) | 0.151 |
| Beta-blockers | 233,781 (8.8) | 20,025 (16.1) | 15,684 (15.5) | 37,189 (8.1) | 35,296 (7.2) | 0.158 |
| Calcium channel blockers | 523,591 (19.8) | 35,937 (28.9) | 31,214 (30.8) | 84,331 (18.4) | 86,862 (17.7) | 0.173 |
| Diuretics | 76,619 (2.9) | 11,741 (9.4) | 6564 (6.5) | 11,775 (2.6) | 10,650 (2.2) | 0.166 |
| Nitrates | 56,049 (2.1) | 7452 (6.0) | 4482 (4.4) | 8262 (1.8) | 8424 (1.7) | 0.121 |
| Lipid-lowering agents | 538,003 (20.3) | 35,491 (28.5) | 30,986 (30.6) | 82,681 (18.1) | 92,072 (18.7) | 0.164 |
| Insulins | 43,083 (1.6) | 5867 (4.7) | 3435 (3.4) | 7039 (1.5) | 5991 (1.2) | 0.108 |
| Antidiabetic drugs | 281,408 (10.6) | 19,480 (15.7) | 17,638 (17.4) | 47,601 (10.4) | 47,592 (9.7) | 0.122 |
| Oral anticoagulants | 30,393 (1.1) | 4928 (4.0) | 2797 (2.8) | 4220 (0.9) | 3855 (0.8) | 0.113 |
| Antiplatelets | 205,166 (7.7) | 22,446 (18.0) | 15,121 (14.9) | 30,357 (6.6) | 30,444 (6.2) | 0.202 |
| Immunosuppressants | 11,624 (0.4) | 1241 (1.0) | 819 (0.8) | 2032 (0.4) | 2295 (0.5) | 0.036 |
| **Number of past healthcare visits** | | | | | | |
| Hospital admission | 0.3 (2.0) | 1.2 (5.3) | 0.8 (4.6) | 0.3 (2.5) | 0.3 (2.3) | 0.125 |
| Doctor consultation | 2.6 (3.7) | 3.8 (4.8) | 4.0 (4.6) | 2.9 (4.0) | 2.7 (3.6) | 0.184 |
| **Doses of COVID-19 vaccines received** | | | | | | |
| 0 | 441,450 (16.7) | 124,443 (100.0) | 0 (0.0) | 0 (0.0) | 0 (0.0) | |
| 1 | 198,158 (7.5) | 0 (0.0) | 101,379 (100.0) | 0 (0.0) | 0 (0.0) | |
| 2 | 914,720 (34.5) | 0 (0.0) | 0 (0.0) | 457,896 (100.0) | 0 (0.0) | |
| 3 or above | 1,094,788 (41.3) | 0 (0.0) | 0 (0.0) | 0 (0.0) | 491,559 (100.0) | |

*SMD* standard mean difference, *COPD* chronic obstructive pulmonary disease.
Age, Charlson Comorbidity Index, Inpatient hospital and Specialist and general outpatient visits are presented in mean ± SD; SMD ≤ 0.1 is considered a good balance between cohorts.

observed in patients with a greater number of doses of COVID-19 vaccination with unvaccinated and incompletely vaccinated patients incurring a greater risk of most clinical sequelae than those completely vaccinated and those who received booster doses of vaccines. Most notably, there was an approximately five-fold reduction in risk of all-cause mortality between unvaccinated patients (18.89; 18.07,19.74) and patients with complete vaccination (3.95; 3.71, 4.20) during the acute phase of infection. Such risk was further reduced amongst patients who received booster dose of vaccines (1.74; 1.50, 2.02).

**Table 2 | Baseline characteristics of patients with COVID-19 stratified by vaccine status and controls from the Hong Kong Hospital Authority after propensity-score weighting**

| Baseline characteristics | Hong Kong Hospital Authority (April 1, 2020 to October 31, 2022) | | | | | |
| --- | --- | --- | --- | --- | --- | --- |
| | Controls (N = 3,828,075) | COVID-19 | | | | SMD |
| | | Unvaccinated (N = 3,809,700) | Incomplete vaccination (N = 3,801,677) | Fully vaccinated (N = 3,818,571) | Received booster dose (N = 3,854,149) | |
| | N/mean (%/SD) | N/mean (%/SD) | N/mean (%/SD) | N/mean (%/SD) | N/mean (%/SD) | |
| Age (years) | 54.1 (18.19) | 52.4 (20.79) | 53.4 (19.75) | 54.0 (18.02) | 54.1 (16.90) | 0.040 |
| Sex, male | 1,690,639 (44.2) | 1,681,641 (44.1) | 1,692,265 (44.5) | 1,689,322 (44.2) | 1,716,080 (44.5) | 0.005 |
| Charlson Comorbidity Index | 1.65 (1.82) | 1.59 (1.95) | 1.64 (1.86) | 1.65 (1.86) | 1.74 (2.06) | 0.032 |
| **Pre-existing morbidities** | | | | | | |
| Respiratory disease | 69,540 (1.8) | 97,239 (2.6) | 91,668 (2.4) | 81,724 (2.1) | 84,607 (2.2) | 0.024 |
| Diabetes | 444,079 (11.6) | 428,379 (11.2) | 438,048 (11.5) | 446,612 (11.7) | 472,078 (12.2) | 0.014 |
| Cardiovascular disease | 291,084 (7.6) | 331,874 (8.7) | 307,673 (8.1) | 292,762 (7.7) | 304,869 (7.9) | 0.019 |
| Myocardial Infarction | 27,309 (0.7) | 30,844 (0.8) | 27,559 (0.7) | 28,154 (0.7) | 31,849 (0.8) | 0.007 |
| Chronic heart failure | 37,531 (1.0) | 53,104 (1.4) | 43,631 (1.1) | 43,140 (1.1) | 42,721 (1.1) | 0.016 |
| Peripheral vascular disease | 10,610 (0.3) | 13,600 (0.4) | 10,275 (0.3) | 9430 (0.2) | 11,916 (0.3) | 0.009 |
| Cerebrovascular disease | 132,775 (3.5) | 179,833 (4.7) | 153,715 (4.0) | 136,713 (3.6) | 136,145 (3.5) | 0.031 |
| COPD | 69,276 (1.8) | 96,613 (2.5) | 91,266 (2.4) | 81,544 (2.1) | 84,507 (2.2) | 0.024 |
| Dementia | 9916 (0.3) | 38,782 (1.0) | 28,411 (0.7) | 17,978 (0.5) | 8948 (0.2) | 0.055 |
| Paralysis | 7705 (0.2) | 11,966 (0.3) | 9190 (0.2) | 8585 (0.2) | 9048 (0.2) | 0.010 |
| Diabetes without chronic complication | 423,588 (11.1) | 405,241 (10.6) | 418,941 (11.0) | 427,648 (11.2) | 447,257 (11.6) | 0.013 |
| Diabetes with chronic complication | 30,253 (0.8) | 29,564 (0.8) | 25,827 (0.7) | 28,255 (0.7) | 39,233 (1.0) | 0.016 |
| Chronic kidney disease | 45,952 (1.2) | 48,784 (1.3) | 45,533 (1.2) | 48,248 (1.3) | 70,173 (1.8) | 0.022 |
| Mild liver disease | 5453 (0.1) | 5909 (0.2) | 5303 (0.1) | 6099 (0.2) | 7418 (0.2) | 0.006 |
| Moderate-severe liver disease | 3666 (0.1) | 3509 (0.1) | 2822 (0.1) | 4673 (0.1) | 6894 (0.2) | 0.014 |
| Ulcers | 37,358 (1.0) | 41,500 (1.1) | 39,446 (1.0) | 42,146 (1.1) | 46,942 (1.2) | 0.011 |
| Rheumatoid arthritis and other inflammatory polyarthropathies | 15,516 (0.4) | 17,100 (0.4) | 16,950 (0.4) | 16,209 (0.4) | 18,742 (0.5) | 0.006 |
| Acquired immune deficiency syndrome (AIDS) | 8 (0.0) | 0 (0.0) | 0 (0.0) | 0 (0.0) | 0 (0.0) | 0.001 |
| Malignancy | 110,554 (2.9) | 94,420 (2.5) | 99,599 (2.6) | 112,459 (2.9) | 172,683 (4.5) | 0.048 |
| Metastatic solid tumor | 18,870 (0.5) | 9,682 (0.3) | 11,332 (0.3) | 21,434 (0.6) | 56,238 (1.5) | 0.062 |
| **Medication history** | | | | | | |
| Renin–angiotensin system agents | 543,451 (14.2) | 535,752 (14.1) | 550,713 (14.5) | 544,896 (14.3) | 561,725 (14.6) | 0.007 |
| Beta-blockers | 343,990 (9.0) | 354,237 (9.3) | 354,979 (9.3) | 345,695 (9.1) | 362,166 (9.4) | 0.008 |
| Calcium channel blockers | 764,595 (20.0) | 732,992 (19.2) | 766,221 (20.2) | 764,390 (20.0) | 782,898 (20.3) | 0.012 |
| Diuretics | 118,650 (3.1) | 117,662 (3.1) | 122,698 (3.2) | 120,500 (3.2) | 129,769 (3.4) | 0.008 |
| Nitrates | 85,421 (2.2) | 85,906 (2.3) | 87,713 (2.3) | 86,476 (2.3) | 90,767 (2.4) | 0.004 |
| Lipid-lowering agents | 781,588 (20.4) | 765,893 (20.1) | 784,093 (20.6) | 785,049 (20.6) | 803,700 (20.9) | 0.008 |
| Insulins | 66,414 (1.7) | 71,182 (1.9) | 69,032 (1.8) | 67,438 (1.8) | 76,617 (2.0) | 0.009 |
| Antidiabetic drugs | 415,093 (10.8) | 421,824 (11.1) | 423,609 (11.1) | 416,006 (10.9) | 431,842 (11.2) | 0.006 |
| Oral anticoagulants | 46,658 (1.2) | 48,834 (1.3) | 48,463 (1.3) | 47,388 (1.2) | 51,978 (1.3) | 0.005 |
| Antiplatelets | 305,831 (8.0) | 296,699 (7.8) | 309,166 (8.1) | 310,017 (8.1) | 321,256 (8.3) | 0.009 |
| Immunosuppressants | 18,178 (0.5) | 20,464 (0.5) | 19,832 (0.5) | 18,872 (0.5) | 20,145 (0.5) | 0.004 |
| **Number of past healthcare visits** | | | | | | |
| Hospital admission | 0.4 (3.80) | 0.5 (2.07) | 0.4 (2.11) | 0.4 (2.81) | 0.6 (6.78) | 0.036 |
| Doctor consultation | 2.8 (4.48) | 2.9 (5.95) | 2.9 (3.69) | 2.8 (3.59) | 2.9 (3.79) | 0.015 |
| **Doses of COVID-19 vaccines received** | | | | | | |
| 0 | 632,151 (16.5) | 3,809,700 (100.0) | 0 (0.0) | 0 (0.0) | 0 (0.0) | |
| 1 | 286,526 (7.5) | 0 (0.0) | 3,801,677 (100.0) | 0 (0.0) | 0 (0.0) | |
| 2 | 1,328,800 (34.7) | 0 (0.0) | 0 (0.0) | 3,818,571 (100.0) | 0 (0.0) | |
| 3 or above | 1,580,598 (41.3) | 0 (0.0) | 0 (0.0) | 0 (0.0) | 3,854,149 (100.0) | |

*SMD* standard mean difference, *COPD* chronic obstructive pulmonary disease.
The number of individuals presented represents the pseudo-population of each stratum of patients with COVID-19 after propensity-score weighting; Age, Charlson Comorbidity Index, Inpatient hospital and Specialist and general outpatient visits are presented in mean ± SD; SMD ≤ 0.1 is considered a good balance between cohorts.

The risk of all-cause mortality reduced progressively over one year between patients with SARS-CoV-2 infection and non-COVID-19 controls [0–30 d: 18.89 (18.07, 19.74); 31–90 d: 3.79 (3.56, 4.03); 91–180 d: 2.11 (1.97, 2.26); 181–270d: 1.97 (1.83, 2.13); 271–365 d: 2.12 (1.92, 2.33)]. A sharp decline in the risks of the outcomes were observed during the post-acute phase between 31 and 90 days of infection compared to the acute phase especially in patients who received complete and booster dose of vaccination. Nevertheless, the risk of several clinical sequelae including heart failure (dose=2: 1.43; 1.13, 1.81; dose ≥3: 1.35; 1.05, 1.75) and all-cause mortality (dose=2: 1.11; 1.02, 1.21) remained significantly greater in patients with SARS-CoV-2 infection. The risk of clinical sequelae further reduced in the subsequent observation period with no significant greater risk of clinical sequelae observed among completely vaccinated and patients with booster dose of vaccines from 271 and 91 days onwards, respectively. Meanwhile, the increased risk of certain clinical sequelae including all-cause mortality (2.12; 1.92, 2.33) continued to persist for up to a year amongst unvaccinated patients. (Tables 3–7 and Fig. 2) The cumulative incidence plots for individual observation windows were shown in Supplementary Fig. 1. Sensitivity analyses reported largely consistent findings for the aforementioned outcomes (Supplementary Tables 2–11). A moderate increase in risk of lung cancer and lymphoma was observed during the first 30 days of infection. (Supplementary Table 12).

Patients aged above 65 or with a CCI of four or above are at a greater risk of clinical sequelae during the acute and post-acute phase of SARS-CoV-2 infection especially amongst unvaccinated or incompletely vaccinated patients. Male and female incurred broadly comparable risk of acute and post-acute sequelae with respect to their vaccination status. (Supplementary Tables 13–18).

## Discussion

This study examined the risk of long-term health consequences of SARS-CoV-2 infection involving multiple organ systems between patients with a history of SARS-CoV-2 infection and non-infected controls over the course of 365 days. The findings of this study showed a reduction in the risk of the majority of clinical sequelae over the course of the observation window suggesting the gradual subside of the risk of long-term health consequences over a year. COVID-19 vaccination, especially the uptake of the booster dose, was found to be effective in reducing the risk of health consequences. Patients who received three or more doses of vaccines did not incur any significant risk increased in clinical sequelae from 91 days onwards from their initial infection. On the other hand, unvaccinated patients were at a greater risk of several clinical sequelae including all-cause mortality up to one year following infection. Such findings provided evidence to support the potential protection from vaccination in reducing the risk of long-term health consequences following SARS-CoV-2 infection.

Previous studies that characterized and evaluated the risk of post-acute sequelae have reported a considerable increase in risk of clinical sequelae involving multiple organ systems[1,2,7,8,10]. In particular, patients with a severe SARS-CoV-2 infection during the acute phase were reported to incur further risk of post-acute clinical sequelae following infection[23]. Nevertheless, evidence that emerged in the early stage of the pandemic has been generated based on data with limited representation of the general population such as discharged hospitalized and older patients with SARS-CoV-2 infection. Since these factors were indicative of patients with more severe disease outcomes compared to the general population, the findings from these studies have shown great discrepancies in the prevalence and risk of clinical sequelae following infection[11,24]. More recent evidence from several large-scale, nationwide population-based studies have indicated a gradual improvements in their recovery status characterized by the reduction in prevalence of self-reported symptoms and proportion of infected individuals reporting non-recovery from health outcomes associated

with SARS-CoV-2 infection[25,26]. A reduction in risk of post-infection clinical sequelae was also reported 6 months following their initial infection amongst individuals with a mild SARS-CoV-2 infection[13]. With consideration of the comparable disease severity induced by the dominant XBB variant to the other sublineages of the omicron strain of SARS-CoV-2, the current body of evidence may indicate the gradual subside and lesser clinical burden on the long-term health consequences caused by such milder strain of SARS-CoV-2 virus than that reported in the earlier stage of the pandemic[27]. Nonetheless, high-risk patients including older people, lack of vaccination, immunosuppression, and individuals with certain underlying comorbidities may still be more vulnerable to poor clinical prognosis from the adverse clinical outcomes following SARS-CoV-2 infection. This emphasizes the need for closer monitoring for these patient groups and provide timely treatment if necessary.

As the pandemic evolved, the global COVID-19 vaccination campaigns have shown to be effective in preventing serious illness and death associated with SARS-CoV-2 infection. Recent research on the booster dose of COVID-19 vaccination has reported substantial extra protection against initial, severe SARS-CoV-2 infection and the rate of mortality by over 80% with a greater risk reduction estimated amongst high-risk population with multi-morbidity, in addition to that conferred by the first and second doses[28–31]. The bivalent omicron-containing booster vaccine was also shown to confer broader immunity against the Omicron variant of SARS-CoV-2 and reduce hospitalizations or deaths due to SARS-CoV-2 infection[32,33]. Previous studies investigating the impact of COVID-19 vaccines on the risk of developing clinical sequelae associated with SARS-CoV-2 infection have reported inconsistent findings due to the difference in the methodological approach employed to evaluate two key aspects: the effect of vaccination on the risk of incident clinical sequelae following SARS-CoV-2 infection and the prevalence of post-infection sequelae outcomes among COVID-19 survivors[19,20]. Further to the current evidence supporting the protection against both acute and post-acute health consequences of SARS-CoV-2 infection conferred by COVID-19 vaccination, the findings of this study demonstrated a graded reduction in the incidence and the persistence in risk according to the number of vaccines doses received prior to infection. The protective effect was more pronounced among the older people and patients with a greater degree of morbidity. Nevertheless, further studies are warranted to evaluate the effect of COVID-19 vaccination in reducing the prevalence of clinical sequelae and understand the potential protective effect of COVID-19 vaccination in patients who have developed onset of clinical sequelae associated with SARS-CoV-2 infection.

Despite the recapitulation of evidence demonstrating the association of COVID-19 vaccination with a reduced risk of clinical sequelae and adverse clinical outcomes during the acute phase of SARS-CoV-2 infection, its protective effect against clinical sequelae in the post-acute phase of infection remained largely unknown. One putative mechanism for the protection effects against the long-term health consequences from vaccination observed could be attributed to the protection against severe SARS-CoV-2 infection during the acute infection may subsequently reduce the risk of long-term health outcomes. The most common COVID-19 vaccines development approach was based on the spike protein of the virus as an antigen accessible by antibodies and immunological cells in the body. The BioNtech COVID-19 vaccines contains mRNA molecules carrying the information for the SARS-CoV-2 spike protein whilst the CoronaVac consisted of whole attenuated SARS-CoV-2 virus capable of provoking an immune response upon administration. The spike proteins attracts antibodies and provoke a high response of sub-types CD8+ killer and CD4 helper T cells, signaling the production of cytokines and proliferations of T memory cells to mediate a lasting immunity, preventing severe health consequences from SARS-CoV-2 infection and the associated irreversible damage to vital organ systems in our body, constituting to a

**Table 3 | Incidence rate and hazard ratio of clinical sequelae in COVID-19 and non-COVID-19 cohort after weighting during 0–30 days of SARS-CoV-2 infection**

0–30 days

| Clinical sequelae | Controls (REF) | | COVID-19 | | | | | | | | | | |
| | | | Unvaccinated | | | Received single dose | | | Received two doses | | | Received booster dose(s) | | |
| | Event | Incidence per 1000 person-years | Event | Incidence per 1000 person-years | Hazard ratio | Event | Incidence per 1000 person-years | Hazard ratio | Event | Incidence per 1000 person-years | Hazard ratio | Event | Incidence per 1000 person-years | Hazard ratio |
|---|---|---|---|---|---|---|---|---|---|---|---|---|---|---|
| Major CVD | 1312 | 4.37 (4.14, 4.61) | 5941 | 20.38 (19.87, 20.90) | 4.64 (4.00, 5.38) | 4038 | 13.69 (13.28, 14.12) | 3.13 (2.60, 3.76) | 3300 | 11.05 (10.68, 11.43) | 2.53 (2.21, 2.89) | 2612 | 8.67 (8.35, 9.01) | 1.99 (1.72, 2.29) |
| Stroke | 818 | 2.61 (2.44, 2.79) | 3605 | 11.90 (11.52, 12.29) | 4.52 (3.74, 5.48) | 2437 | 7.94 (7.63, 8.26) | 3.03 (2.40, 3.83) | 2287 | 7.35 (7.05, 7.65) | 2.81 (2.39, 3.31) | 1702 | 5.41 (5.15, 5.67) | 2.07 (1.71, 2.51) |
| Myocardial infarction | 300 | 0.93 (0.83, 1.04) | 1895 | 6.00 (5.74, 6.28) | 6.42 (4.99, 8.26) | 1560 | 4.91 (4.67, 5.16) | 5.27 (3.88, 7.16) | 980 | 3.06 (2.87, 3.25) | 3.29 (2.53, 4.27) | 925 | 2.85 (2.67, 3.04) | 3.07 (2.38, 3.96) |
| Heart failure | 272 | 0.84 (0.75, 0.95) | 1326 | 4.22 (4.00, 4.46) | 4.99 (3.72, 6.68) | 913 | 2.88 (2.70, 3.08) | 3.41 (2.36, 4.93) | 643 | 2.01 (1.86, 2.17) | 2.38 (1.75, 3.25) | 354 | 1.09 (0.98, 1.21) | 1.30 (0.88, 1.92) |
| Atrial fibrillation | 154 | 0.48 (0.41, 0.56) | 668 | 2.14 (1.98, 2.31) | 4.43 (2.95, 6.65) | 597 | 1.90 (1.75, 2.06) | 3.93 (2.51, 6.15) | 425 | 1.34 (1.22, 1.47) | 2.77 (1.89, 4.06) | 509 | 1.58 (1.45, 1.72) | 3.28 (2.34, 4.60) |
| Coronary artery disease | 483 | 1.55 (1.41, 1.69) | 2116 | 6.88 (6.60, 7.18) | 4.43 (3.50, 5.60) | 1669 | 5.41 (5.15, 5.67) | 3.49 (2.62, 4.63) | 1267 | 4.07 (3.85, 4.30) | 2.63 (2.11, 3.27) | 1098 | 3.51 (3.30, 3.72) | 2.27 (1.82, 2.82) |
| Deep vein thrombosis | 41 | 0.13 (0.09, 0.17) | 288 | 0.91 (0.81, 1.02) | 7.18 (3.53, 14.60) | 273 | 0.85 (0.76, 0.96) | 6.78 (3.19, 14.39) | 110 | 0.34 (0.28, 0.41) | 2.70 (1.25, 5.81) | 72 | 0.22 (0.17, 0.27) | 1.75 (0.77, 3.96) |
| Cardiovascular mortality | 1092 | 3.36 (3.17, 3.57) | 6361 | 20.00 (19.51, 20.49) | 5.92 (5.17, 6.78) | 3440 | 10.74 (10.39, 11.10) | 3.19 (2.65, 3.84) | 2118 | 6.55 (6.28, 6.84) | 1.95 (1.63, 2.32) | 1057 | 3.23 (3.04, 3.43) | 0.96 (0.74, 1.26) |
| Chronic pulmonary disease | 115 | 0.36 (0.30, 0.43) | 361 | 1.16 (1.05, 1.29) | 3.22 (1.88, 5.50) | 688 | 2.20 (2.04, 2.37) | 6.10 (3.65, 10.21) | 399 | 1.26 (1.14, 1.39) | 3.51 (2.33, 5.27) | 234 | 0.73 (0.64, 0.83) | 2.03 (1.30, 3.16) |
| Acute respiratory distress syndrome | 189 | 0.58 (0.50, 0.67) | 1405 | 4.45 (4.23, 4.69) | 7.58 (5.44, 10.55) | 715 | 2.25 (2.08, 2.41) | 3.83 (2.53, 5.80) | 449 | 1.40 (1.27, 1.53) | 2.39 (1.66, 3.43) | 431 | 1.33 (1.20, 1.45) | 2.27 (1.59, 3.23) |
| Seizure | 103 | 0.32 (0.26, 0.38) | 741 | 2.37 (2.20, 2.54) | 7.40 (4.68, 11.69) | 573 | 1.81 (1.66, 1.96) | 5.67 (3.20, 10.06) | 233 | 0.73 (0.64, 0.82) | 2.28 (1.43, 3.65) | 296 | 0.91 (0.81, 1.02) | 2.86 (1.51, 5.43) |
| End-stage renal disease | 37 | 0.11 (0.08, 0.16) | 238 | 0.75 (0.66, 0.85) | 6.50 (2.86, 14.76) | 55 | 0.17 (0.13, 0.22) | 1.50 (0.37, 6.06) | 94 | 0.29 (0.24, 0.35) | 2.54 (1.02, 6.30) | 87 | 0.27 (0.21, 0.33) | 2.33 (0.63, 8.62) |
| Acute kidney injury | 87 | 0.27 (0.22, 0.33) | 803 | 2.54 (2.37, 2.73) | 9.45 (6.26, 14.27) | 818 | 2.57 (2.40, 2.75) | 9.57 (6.02, 15.20) | 313 | 0.97 (0.87, 1.09) | 3.63 (2.24, 5.86) | 387 | 1.19 (1.07, 1.31) | 4.43 (2.77, 7.09) |
| Pancreatitis | 56 | 0.17 (0.13, 0.22) | 270 | 0.85 (0.76, 0.96) | 4.89 (2.41, 9.93) | 137 | 0.43 (0.36, 0.51) | 2.47 (0.80, 7.62) | 146 | 0.45 (0.38, 0.53) | 2.61 (1.38, 4.92) | 133 | 0.41 (0.34, 0.48) | 2.35 (1.31, 4.22) |
| All-cause mortality | 5379 | 16.56 (16.12, 17.00) | 99,819 | 313.79 (311.85, 315.74) | 18.89 (18.07, 19.74) | 47,563 | 148.53 (147.20, 149.87) | 8.96 (8.46, 9.48) | 21,135 | 65.40 (64.52, 66.28) | 3.95 (3.71, 4.20) | 9418 | 28.81 (28.23, 29.40) | 1.74 (1.50, 2.02) |

*CI* confidence interval, *REF* reference group, *Major CVD* major cardiovascular disease, Composite outcome of stroke, heart failure and coronary artery disease.
Hazard ratio (HR) and 95% confidence interval (95% CI) were estimated by Cox regression; HR > 1 (or <1) indicates patients with COVID-19 had a higher (lower) risk of sequelae compared to the non-COVID-19 control cohort.
Unvaccinated, incomplete vaccinated, fully vaccinated and received booster dose of COVID-19 vaccines refers to patients who have received 0 dose, 1 doses, 2 doses and ≥3 doses of BioNtech or CoronaVac COVID-19 vaccines, respectively.

**Table 4 | Incidence rate and hazard ratio of clinical sequelae in COVID-19 and non-COVID-19 cohort after weighting during 31–90 days of SARS-CoV-2 infection**

### 31–90 days

| Clinical sequelae | Controls (REF) | | COVID-19 | | | | | | | | | | |
| --- | --- | --- | --- | --- | --- | --- | --- | --- | --- | --- | --- | --- | --- |
| | | | Unvaccinated | | | Incompletely vaccinated | | | Fully vaccinated | | | Received booster dose | | |
| | Event | Incidence per 1000 person-years | Event | Incidence per 1000 person-years | Hazard ratio | Event | Incidence per 1000 person-years | Hazard ratio | Event | Incidence per 1000 person-years | Hazard ratio | Event | Incidence per 1000 person-years | Hazard ratio |
| Major CVD | 3124 | 5.31 (5.12, 5.50) | 5814 | 10.23 (9.97, 10.49) | 1.93 (1.67, 2.22) | 4397 | 7.60 (7.38, 7.83) | 1.43 (1.21, 1.70) | 3294 | 5.63 (5.44, 5.82) | 1.06 (0.94, 1.20) | 3455 | 5.87 (5.68, 6.07) | 1.11 (0.98, 1.25) |
| Stroke | 1775 | 2.89 (2.76, 3.03) | 2863 | 4.84 (4.66, 5.02) | 1.67 (1.37, 2.05) | 2400 | 3.99 (3.83, 4.15) | 1.38 (1.09, 1.74) | 1759 | 2.89 (2.75, 3.02) | 1.00 (0.85, 1.18) | 1597 | 2.60 (2.47, 2.73) | 0.90 (0.75, 1.08) |
| Myocardial infarction | 695 | 1.10 (1.02, 1.18) | 1166 | 1.89 (1.78, 2.00) | 1.72 (1.29, 2.29) | 683 | 1.10 (1.02, 1.18) | 1.00 (0.64, 1.55) | 716 | 1.14 (1.06, 1.22) | 1.04 (0.80, 1.34) | 701 | 1.11 (1.03, 1.19) | 1.01 (0.78, 1.31) |
| Heart failure | 724 | 1.15 (1.07, 1.24) | 2273 | 3.71 (3.56, 3.86) | 3.22 (2.61, 3.98) | 1223 | 1.97 (1.86, 2.08) | 1.71 (1.29, 2.28) | 1027 | 1.64 (1.54, 1.74) | 1.43 (1.13, 1.81) | 982 | 1.56 (1.46, 1.66) | 1.35 (1.05, 1.75) |
| Atrial fibrillation | 474 | 0.76 (0.69, 0.83) | 1119 | 1.84 (1.73, 1.95) | 2.43 (1.80, 3.28) | 706 | 1.14 (1.06, 1.23) | 1.51 (1.01, 2.26) | 633 | 1.02 (0.94, 1.10) | 1.34 (1.01, 1.78) | 507 | 0.81 (0.74, 0.88) | 1.07 (0.78, 1.45) |
| Coronary artery disease | 1227 | 2.00 (1.90, 2.12) | 2347 | 3.91 (3.75, 4.07) | 1.95 (1.57, 2.42) | 1533 | 2.53 (2.41, 2.66) | 1.26 (0.94, 1.69) | 1343 | 2.20 (2.09, 2.32) | 1.10 (0.91, 1.33) | 1606 | 2.63 (2.50, 2.76) | 1.31 (1.11, 1.55) |
| Deep vein thrombosis | 118 | 0.19 (0.15, 0.22) | 433 | 0.70 (0.63, 0.77) | 3.76 (2.19, 6.44) | 221 | 0.35 (0.31, 0.40) | 1.90 (0.96, 3.73) | 237 | 0.38 (0.33, 0.43) | 2.02 (1.25, 3.28) | 106 | 0.17 (0.14, 0.20) | 0.90 (0.48, 1.68) |
| Cardiovascular mortality | 1324 | 2.08 (1.97, 2.20) | 2555 | 4.11 (3.95, 4.27) | 1.97 (1.63, 2.39) | 1570 | 2.50 (2.38, 2.63) | 1.20 (0.93, 1.55) | 1254 | 1.98 (1.87, 2.09) | 0.95 (0.77, 1.18) | 693 | 1.09 (1.01, 1.17) | 0.52 (0.38, 0.72) |
| Chronic pulmonary disease | 308 | 0.49 (0.44, 0.55) | 881 | 1.45 (1.36, 1.55) | 2.94 (2.03, 4.26) | 633 | 1.03 (0.95, 1.11) | 2.09 (1.26, 3.45) | 385 | 0.62 (0.56, 0.69) | 1.26 (0.88, 1.81) | 276 | 0.44 (0.39, 0.50) | 0.90 (0.60, 1.33) |
| Acute respiratory distress syndrome | 539 | 0.85 (0.78, 0.93) | 945 | 1.53 (1.44, 1.63) | 1.80 (1.26, 2.57) | 892 | 1.43 (1.34, 1.52) | 1.68 (1.14, 2.48) | 625 | 0.99 (0.92, 1.07) | 1.16 (0.88, 1.55) | 715 | 1.13 (1.05, 1.21) | 1.32 (0.97, 1.81) |
| Seizure | 290 | 0.46 (0.41, 0.51) | 706 | 1.15 (1.07, 1.24) | 2.51 (1.62, 3.89) | 537 | 0.86 (0.79, 0.94) | 1.89 (1.09, 3.27) | 351 | 0.56 (0.50, 0.62) | 1.22 (0.84, 1.77) | 197 | 0.31 (0.27, 0.36) | 0.68 (0.40, 1.15) |
| End-stage renal disease | 62 | 0.10 (0.08, 0.13) | 101 | 0.16 (0.13, 0.20) | 1.65 (0.58, 4.73) | 139 | 0.22 (0.19, 0.26) | 2.27 (0.86, 5.95) | 63 | 0.10 (0.08, 0.13) | 1.02 (0.42, 2.47) | 21 | 0.03 (0.02, 0.05) | 0.34 (0.10, 1.10) |
| Acute kidney injury | 203 | 0.32 (0.28, 0.37) | 742 | 1.20 (1.12, 1.29) | 3.74 (2.53, 5.52) | 513 | 0.82 (0.75, 0.89) | 2.55 (1.54, 4.23) | 252 | 0.40 (0.35, 0.45) | 1.24 (0.77, 2.00) | 258 | 0.41 (0.36, 0.46) | 1.27 (0.76, 2.10) |
| Pancreatitis | 145 | 0.23 (0.19, 0.27) | 475 | 0.77 (0.70, 0.84) | 3.37 (1.93, 5.88) | 265 | 0.42 (0.37, 0.48) | 1.86 (0.90, 3.83) | 154 | 0.24 (0.21, 0.28) | 1.07 (0.60, 1.92) | 176 | 0.28 (0.24, 0.32) | 1.22 (0.75, 1.97) |
| All-cause mortality | 7708 | 12.12 (11.85, 12.39) | 28,568 | 45.94 (45.40, 46.47) | 3.79 (3.56, 4.03) | 13,463 | 21.43 (21.07, 21.79) | 1.77 (1.61, 1.94) | 8539 | 13.49 (13.21, 13.78) | 1.11 (1.02, 1.21) | 6728 | 10.55 (10.30, 10.80) | 0.87 (0.76, 0.99) |

CI confidence interval, REF reference group, Major CVD major cardiovascular disease, Composite outcome of stroke, heart failure and coronary artery disease.
Hazard ratio (HR) and 95% confidence interval (95% CI) were estimated by Cox regression. HR > 1 (or <1) indicates patients with COVID-19 had a higher (lower) risk of sequelae compared to the non-COVID-19 control cohort.
Unvaccinated, incomplete vaccinated, fully vaccinated and received booster dose of COVID-19 vaccines refers to patients who have received 0 dose, 1 doses, 2 doses and ≥3 doses of BioNtech or CoronaVac COVID-19 vaccines, respectively.

**Table 5 | Incidence rate and hazard ratio of clinical sequelae in COVID-19 and non-COVID-19 cohort after weighting during 91–180 days of SARS-CoV-2 infection**

**91–180 days**

| Clinical sequelae | Controls (REF) | | COVID-19 | | | | | | | | | | | |
| | | | Unvaccinated | | | Incomplete vaccination | | | Fully vaccinated | | | Received booster dose | | |
| | Event | Incidence per 1000 person-years | Event | Incidence per 1000 person-years | Hazard ratio | Event | Incidence per 1000 person-years | Hazard ratio | Event | Incidence per 1000 person-years | Hazard ratio | Event | Incidence per 1000 person-years | Hazard ratio |
|---|---|---|---|---|---|---|---|---|---|---|---|---|---|---|
| Major CVD | 4239 | 5.58 (5.42, 5.75) | 6525 | 8.24 (8.04, 8.44) | 1.48 (1.29, 1.70) | 5820 | 6.91 (6.74, 7.09) | 1.25 (1.07, 1.45) | 4709 | 5.63 (5.47, 5.79) | 1.01 (0.92, 1.12) | 3223 | 5.00 (4.83, 5.18) | 0.89 (0.79, 0.99) |
| Stroke | 2461 | 3.11 (2.99, 3.23) | 3270 | 3.96 (3.83, 4.10) | 1.28 (1.05, 1.56) | 3196 | 3.65 (3.52, 3.77) | 1.18 (0.96, 1.45) | 2637 | 3.03 (2.92, 3.15) | 0.98 (0.86, 1.12) | 1561 | 2.33 (2.21, 2.44) | 0.74 (0.63, 0.87) |
| Myocardial infarction | 913 | 1.12 (1.05, 1.19) | 1499 | 1.74 (1.65, 1.83) | 1.55 (1.16, 2.07) | 1156 | 1.27 (1.20, 1.35) | 1.14 (0.82, 1.58) | 938 | 1.05 (0.98, 1.11) | 0.93 (0.74, 1.17) | 899 | 1.30 (1.22, 1.39) | 1.17 (0.92, 1.47) |
| Heart failure | 834 | 1.03 (0.96, 1.10) | 2135 | 2.49 (2.39, 2.60) | 2.44 (1.95, 3.05) | 1516 | 1.68 (1.59, 1.76) | 1.65 (1.24, 2.19) | 1299 | 1.45 (1.38, 1.53) | 1.43 (1.16, 1.75) | 716 | 1.04 (0.97, 1.12) | 1.00 (0.77, 1.30) |
| Atrial fibrillation | 676 | 0.84 (0.77, 0.90) | 1047 | 1.23 (1.16, 1.31) | 1.47 (1.07, 2.02) | 1071 | 1.19 (1.12, 1.26) | 1.42 (1.03, 1.96) | 956 | 1.08 (1.01, 1.14) | 1.28 (1.02, 1.61) | 800 | 1.17 (1.09, 1.25) | 1.41 (0.96, 2.05) |
| Coronary artery disease | 1699 | 2.15 (2.05, 2.25) | 2688 | 3.20 (3.09, 3.33) | 1.49 (1.21, 1.84) | 2403 | 2.72 (2.62, 2.83) | 1.27 (1.01, 1.59) | 1856 | 2.13 (2.04, 2.23) | 0.99 (0.85, 1.16) | 1514 | 2.27 (2.16, 2.39) | 1.05 (0.89, 1.25) |
| Deep vein thrombosis | 149 | 0.18 (0.15, 0.21) | 263 | 0.30 (0.27, 0.34) | 1.66 (0.87, 3.16) | 343 | 0.38 (0.34, 0.42) | 2.06 (1.05, 4.04) | 98 | 0.11 (0.09, 0.13) | 0.60 (0.30, 1.17) | 135 | 0.19 (0.16, 0.23) | 1.07 (0.47, 2.47) |
| Cardiovascular mortality | 1557 | 1.90 (1.80, 1.99) | 2870 | 3.30 (3.18, 3.42) | 1.74 (1.44, 2.10) | 1673 | 1.83 (1.74, 1.92) | 0.96 (0.75, 1.24) | 1214 | 1.34 (1.27, 1.42) | 0.71 (0.58, 0.87) | 397 | 0.57 (0.52, 0.63) | 0.30 (0.21, 0.44) |
| Chronic pulmonary disease | 432 | 0.54 (0.49, 0.59) | 955 | 1.13 (1.06, 1.20) | 2.12 (1.46, 3.09) | 864 | 0.97 (0.90, 1.03) | 1.83 (1.19, 2.81) | 559 | 0.63 (0.58, 0.69) | 1.19 (0.89, 1.59) | 425 | 0.63 (0.57, 0.69) | 1.15 (0.78, 1.68) |
| Acute respiratory distress syndrome | 714 | 0.87 (0.81, 0.94) | 1447 | 1.68 (1.59, 1.77) | 1.92 (1.43, 2.58) | 1017 | 1.12 (1.05, 1.19) | 1.28 (0.89, 1.84) | 1022 | 1.14 (1.07, 1.21) | 1.30 (1.04, 1.62) | 594 | 0.86 (0.79, 0.93) | 0.99 (0.70, 1.40) |
| Seizure | 389 | 0.48 (0.43, 0.53) | 1046 | 1.22 (1.15, 1.30) | 2.59 (1.81, 3.71) | 936 | 1.03 (0.97, 1.10) | 2.20 (1.46, 3.34) | 412 | 0.46 (0.42, 0.50) | 0.97 (0.70, 1.36) | 300 | 0.43 (0.39, 0.48) | 0.89 (0.48, 1.65) |
| End-stage renal disease | 77 | 0.09 (0.08, 0.12) | 132 | 0.15 (0.13, 0.18) | 1.62 (0.58, 4.52) | 182 | 0.20 (0.17, 0.23) | 2.15 (0.83, 5.62) | 39 | 0.04 (0.03, 0.06) | 0.47 (0.16, 1.34) | 0 | NA | NA |
| Acute kidney injury | 355 | 0.43 (0.39, 0.48) | 638 | 0.74 (0.68, 0.80) | 1.71 (1.14, 2.57) | 498 | 0.55 (0.50, 0.60) | 1.27 (0.81, 1.99) | 441 | 0.49 (0.45, 0.54) | 1.14 (0.80, 1.61) | 144 | 0.21 (0.18, 0.25) | 0.48 (0.26, 0.88) |
| Pancreatitis | 181 | 0.22 (0.19, 0.25) | 205 | 0.24 (0.21, 0.27) | 1.07 (0.52, 2.21) | 186 | 0.20 (0.18, 0.23) | 0.92 (0.37, 2.27) | 194 | 0.22 (0.19, 0.25) | 0.97 (0.58, 1.63) | 104 | 0.15 (0.12, 0.18) | 0.69 (0.36, 1.29) |
| All-cause mortality | 9958 | 12.13 (11.89, 12.37) | 22,224 | 25.56 (25.23, 25.90) | 2.11 (1.97, 2.26) | 13,014 | 14.22 (13.98, 14.47) | 1.17 (1.07, 1.29) | 8752 | 9.69 (9.49, 9.90) | 0.80 (0.74, 0.87) | 5537 | 7.98 (7.77, 8.19) | 0.66 (0.57, 0.75) |

CI confidence interval, REF reference group, Major CVD major cardiovascular disease, Composite outcome of stroke, heart failure and coronary artery disease.
Hazard ratio (HR) and 95% confidence interval (95% CI) were estimated by Cox regression, HR > 1 (or <1) indicates patients with COVID-19 had higher (lower) risk of sequelae compared to the non-COVID-19 control cohort.
Unvaccinated, incomplete vaccinated, fully vaccinated, and received booster dose of COVID-19 vaccines refers to patients who have received 0 dose, 1 doses, 2 doses and ≥3 doses of BioNtech or CoronaVac COVID-19 vaccines, respectively.

**Table 6 | Incidence rate and hazard ratio of clinical sequelae in COVID-19 and non-COVID-19 cohort after weighting during 181–270 days of SARS-CoV-2 infection**

**181–270 days**

| Clinical sequelae | Controls (REF) | | COVID-19 | | | | | | | | | | |
| --- | --- | --- | --- | --- | --- | --- | --- | --- | --- | --- | --- | --- | --- |
| | | | Unvaccinated | | | Incomplete vaccination | | | Fully vaccinated | | | Received booster dose | | |
| | Event | Incidence per 1000 person-years | Event | Incidence per 1000 person-years | Hazard ratio | Event | Incidence per 1000 person-years | Hazard ratio | Event | Incidence per 1000 person-years | Hazard ratio | Event | Incidence per 1000 person-years | Hazard ratio |
| Major CVD | 3462 | 5.79 (5.60, 5.99) | 4973 | 8.28 (8.05, 8.51) | 1.43 (1.23, 1.66) | 4720 | 7.54 (7.33, 7.76) | 1.30 (1.12, 1.52) | 3574 | 5.77 (5.58, 5.96) | 1.00 (0.90, 1.11) | 2529 | 4.82 (4.64, 5.01) | 0.83 (0.71, 0.98) |
| Stroke | 1927 | 3.09 (2.96, 3.23) | 2865 | 4.57 (4.41, 4.74) | 1.48 (1.22, 1.80) | 2277 | 3.49 (3.35, 3.64) | 1.13 (0.91, 1.40) | 1935 | 3.00 (2.87, 3.14) | 0.97 (0.84, 1.12) | 1311 | 2.39 (2.26, 2.52) | 0.77 (0.62, 0.96) |
| Myocardial infarction | 826 | 1.29 (1.20, 1.37) | 893 | 1.36 (1.28, 1.45) | 1.06 (0.78, 1.44) | 1223 | 1.81 (1.71, 1.91) | 1.40 (1.04, 1.89) | 695 | 1.05 (0.97, 1.12) | 0.81 (0.64, 1.04) | 1108 | 1.96 (1.84, 2.08) | 1.54 (0.60, 3.91) |
| Heart Failure | 769 | 1.20 (1.12, 1.29) | 1889 | 2.90 (2.77, 3.04) | 2.42 (1.94, 3.02) | 1501 | 2.23 (2.12, 2.34) | 1.86 (1.44, 2.40) | 818 | 1.23 (1.15, 1.32) | 1.03 (0.81, 1.31) | 684 | 1.21 (1.12, 1.30) | 1.01 (0.51, 2.00) |
| Atrial fibrillation | 544 | 0.85 (0.79, 0.93) | 969 | 1.50 (1.41, 1.60) | 1.76 (1.29, 2.40) | 748 | 1.12 (1.04, 1.20) | 1.31 (0.91, 1.90) | 788 | 1.20 (1.11, 1.28) | 1.40 (1.11, 1.78) | 383 | 0.68 (0.62, 0.75) | 0.79 (0.51, 1.23) |
| Coronary Artery Disease | 1462 | 2.35 (2.23, 2.47) | 1706 | 2.68 (2.55, 2.81) | 1.14 (0.90, 1.44) | 2019 | 3.08 (2.94, 3.21) | 1.31 (1.04, 1.65) | 1601 | 2.48 (2.36, 2.61) | 1.06 (0.90, 1.24) | 1110 | 2.03 (1.92, 2.15) | 0.87 (0.69, 1.10) |
| Deep vein thrombosis | 114 | 0.18 (0.15, 0.21) | 118 | 0.18 (0.15, 0.21) | 1.02 (0.51, 2.05) | 86 | 0.13 (0.10, 0.16) | 0.72 (0.30, 1.73) | 137 | 0.20 (0.17, 0.24) | 1.16 (0.68, 1.99) | 49 | 0.09 (0.06, 0.11) | 0.48 (0.17, 1.34) |
| Cardiovascular mortality | 984 | 1.52 (1.43, 1.62) | 1761 | 2.67 (2.54, 2.79) | 1.76 (1.41, 2.19) | 1155 | 1.70 (1.60, 1.80) | 1.12 (0.85, 1.47) | 645 | 0.96 (0.89, 1.04) | 0.64 (0.48, 0.83) | 471 | 0.82 (0.75, 0.90) | 0.54 (0.31, 0.92) |
| Chronic pulmonary disease | 361 | 0.57 (0.51, 0.63) | 822 | 1.28 (1.19, 1.37) | 2.25 (1.56, 3.23) | 635 | 0.95 (0.88, 1.03) | 1.68 (1.11, 2.53) | 382 | 0.58 (0.53, 0.64) | 1.02 (0.74, 1.41) | 195 | 0.35 (0.30, 0.40) | 0.62 (0.38, 1.02) |
| Acute respiratory distress syndrome | 562 | 0.87 (0.80, 0.95) | 859 | 1.31 (1.23, 1.40) | 1.50 (1.06, 2.13) | 582 | 0.86 (0.79, 0.93) | 0.98 (0.65, 1.49) | 493 | 0.74 (0.68, 0.81) | 0.85 (0.64, 1.12) | 469 | 0.83 (0.76, 0.91) | 0.96 (0.45, 2.06) |
| Seizure | 320 | 0.50 (0.45, 0.55) | 1036 | 1.59 (1.50, 1.69) | 3.21 (2.27, 4.53) | 781 | 1.16 (1.08, 1.24) | 2.33 (1.54, 3.53) | 268 | 0.40 (0.36, 0.45) | 0.81 (0.55, 1.19) | 213 | 0.38 (0.33, 0.43) | 0.75 (0.25, 2.24) |
| End-stage renal disease | 52 | 0.08 (0.06, 0.11) | 98 | 0.15 (0.12, 0.18) | 1.83 (0.74, 4.51) | 109 | 0.16 (0.13, 0.19) | 1.98 (0.88, 4.44) | 91 | 0.14 (0.11, 0.17) | 1.68 (0.84, 3.38) | 0 | NA | NA |
| Acute kidney injury | 302 | 0.47 (0.42, 0.52) | 452 | 0.69 (0.63, 0.75) | 1.47 (0.97, 2.22) | 351 | 0.52 (0.47, 0.58) | 1.11 (0.67, 1.84) | 272 | 0.41 (0.36, 0.46) | 0.87 (0.57, 1.33) | 99 | 0.17 (0.14, 0.21) | 0.37 (0.17, 0.81) |
| Pancreatitis | 173 | 0.27 (0.23, 0.31) | 296 | 0.45 (0.40, 0.50) | 1.68 (0.91, 3.10) | 163 | 0.24 (0.21, 0.28) | 0.90 (0.37, 2.18) | 148 | 0.22 (0.19, 0.26) | 0.83 (0.51, 1.35) | 83 | 0.14 (0.12, 0.18) | 0.53 (0.24, 1.17) |
| All-cause mortality | 8058 | 12.45 (12.18, 12.72) | 16,197 | 24.53 (24.15, 24.91) | 1.97 (1.83, 2.13) | 9673 | 14.20 (13.92, 14.48) | 1.14 (1.04, 1.26) | 5597 | 8.36 (8.14, 8.58) | 0.67 (0.61, 0.74) | 7931 | 13.89 (13.59, 14.20) | 1.11 (0.76, 1.63) |

CI confidence interval, REF reference group, Major CVD major cardiovascular disease, Composite outcome of stroke, heart failure and coronary artery disease.
Hazard ratio (HR) and 95% confidence interval (95% CI) were estimated by Cox regression, HR > 1 (or <1) indicates patients with COVID-19 had a higher (lower) risk of sequelae compared to the non-COVID-19 control cohort.
Unvaccinated, incomplete vaccinated, fully vaccinated and received booster dose of COVID-19 vaccines refers to patients who have received 0 dose, 1 doses, 2 doses and ≥3 doses of BioNtech or CoronaVac COVID-19 vaccines, respectively.

**Table 7 | Incidence rate and hazard ratio of clinical sequelae in COVID-19 and non-COVID-19 cohort after weighting during 271–365 days of SARS-CoV-2 infection**

**271–365 days**

| Clinical sequelae | Controls (REF) | | COVID-19 | | | | | | | | | | |
| --- | --- | --- | --- | --- | --- | --- | --- | --- | --- | --- | --- | --- | --- |
| | | | Unvaccinated | | | Incomplete vaccination | | | Fully vaccinated | | | Received booster dose | | |
| | Event | Incidence per 1000 person-years | Event | Incidence per 1000 person-years | Hazard ratio | Event | Incidence per 1000 person-years | Hazard ratio | Event | Incidence per 1000 person-years | Hazard ratio | Event | Incidence per 1000 person-years | Hazard ratio |
| Major CVD | 2090 | 6.02 (5.77, 6.29) | 2405 | 6.59 (6.33, 6.86) | 1.16 (0.95, 1.42) | 2370 | 7.13 (6.85, 7.42) | 1.17 (0.95, 1.45) | 1833 | 5.36 (5.12, 5.61) | 0.89 (0.77, 1.02) | 1642 | 4.88 (4.65, 5.12) | 0.80 (0.59, 1.07) |
| Stroke | 1099 | 3.04 (2.86, 3.22) | 1180 | 3.12 (2.94, 3.30) | 1.07 (0.80, 1.43) | 934 | 2.70 (2.53, 2.88) | 0.88 (0.63, 1.23) | 973 | 2.74 (2.57, 2.91) | 0.90 (0.74, 1.09) | 1014 | 2.86 (2.69, 3.04) | 0.93 (0.59, 1.45) |
| Myocardial infarction | 509 | 1.37 (1.25, 1.49) | 482 | 1.22 (1.12, 1.34) | 0.96 (0.63, 1.46) | 656 | 1.83 (1.70, 1.98) | 1.32 (0.89, 1.98) | 463 | 1.27 (1.16, 1.39) | 0.92 (0.68, 1.25) | 245 | 0.67 (0.59, 0.75) | 0.48 (0.30, 0.77) |
| Heart Failure | 573 | 1.54 (1.42, 1.68) | 1070 | 2.73 (2.57, 2.90) | 1.90 (1.44, 2.52) | 946 | 2.65 (2.49, 2.83) | 1.70 (1.25, 2.30) | 659 | 1.81 (1.67, 1.95) | 1.16 (0.89, 1.52) | 395 | 1.08 (0.98, 1.19) | 0.69 (0.33, 1.45) |
| Atrial fibrillation | 305 | 0.83 (0.74, 0.92) | 504 | 1.30 (1.19, 1.41) | 1.64 (1.05, 2.57) | 413 | 1.17 (1.06, 1.29) | 1.41 (0.86, 2.31) | 321 | 0.88 (0.79, 0.98) | 1.07 (0.75, 1.52) | 565 | 1.55 (1.43, 1.68) | 1.86 (0.64, 5.37) |
| Coronary Artery Disease | 864 | 2.40 (2.24, 2.56) | 783 | 2.04 (1.90, 2.19) | 0.89 (0.63, 1.26) | 1004 | 2.89 (2.72, 3.07) | 1.20 (0.86, 1.67) | 684 | 1.93 (1.79, 2.08) | 0.80 (0.63, 1.01) | 651 | 1.85 (1.71, 1.99) | 0.76 (0.51, 1.14) |
| Deep vein thrombosis | 56 | 0.15 (0.12, 0.19) | 140 | 0.35 (0.30, 0.42) | 2.50 (1.08, 5.76) | 31 | 0.09 (0.06, 0.12) | 0.56 (0.14, 2.31) | 30 | 0.08 (0.06, 0.11) | 0.54 (0.19, 1.53) | 28 | 0.08 (0.05, 0.11) | 0.49 (0.12, 2.04) |
| Cardiovascular mortality | 95 | 0.25 (0.21, 0.31) | 194 | 0.49 (0.43, 0.56) | 1.77 (0.85, 3.69) | 52 | 0.14 (0.11, 0.19) | 0.59 (0.18, 1.90) | 8 | 0.02 (0.01, 0.04) | 0.08 (0.01, 0.59) | 0 | NA | NA |
| Chronic pulmonary disease | 287 | 0.78 (0.69, 0.87) | 385 | 0.99 (0.90, 1.10) | 1.37 (0.81, 2.31) | 470 | 1.33 (1.22, 1.46) | 1.69 (1.03, 2.77) | 345 | 0.95 (0.86, 1.06) | 1.22 (0.85, 1.75) | 109 | 0.30 (0.25, 0.36) | 0.38 (0.19, 0.75) |
| Acute respiratory distress syndrome | 278 | 0.75 (0.66, 0.84) | 361 | 0.92 (0.83, 1.01) | 1.32 (0.77, 2.26) | 499 | 1.39 (1.27, 1.52) | 1.83 (1.19, 2.83) | 305 | 0.83 (0.74, 0.93) | 1.11 (0.75, 1.63) | 596 | 1.63 (1.50, 1.76) | 2.12 (0.61, 7.41) |
| Seizure | 180 | 0.48 (0.42, 0.56) | 268 | 0.69 (0.61, 0.77) | 1.51 (0.83, 2.76) | 316 | 0.88 (0.79, 0.99) | 1.81 (0.99, 3.30) | 143 | 0.39 (0.33, 0.46) | 0.80 (0.51, 1.27) | 126 | 0.34 (0.28, 0.40) | 0.69 (0.19, 2.55) |
| End-stage renal disease | 41 | 0.11 (0.08, 0.15) | 162 | 0.41 (0.35, 0.48) | 4.12 (1.73, 9.80) | 56 | 0.16 (0.12, 0.20) | 1.41 (0.42, 4.79) | 4 | 0.01 (0.00, 0.03) | 0.11 (0.01, 0.81) | 58 | 0.16 (0.12, 0.20) | 1.38 (0.19, 10.30) |
| Acute kidney injury | 150 | 0.40 (0.34, 0.47) | 179 | 0.45 (0.39, 0.52) | 1.21 (0.64, 2.29) | 93 | 0.26 (0.21, 0.32) | 0.63 (0.25, 1.60) | 96 | 0.26 (0.21, 0.32) | 0.65 (0.34, 1.22) | 419 | 1.14 (1.03, 1.25) | 2.70 (0.49, 14.84) |
| Pancreatitis | 79 | 0.21 (0.17, 0.26) | 190 | 0.48 (0.42, 0.55) | 2.46 (1.12, 5.39) | 118 | 0.33 (0.27, 0.39) | 1.53 (0.55, 4.25) | 34 | 0.09 (0.07, 0.13) | 0.44 (0.17, 1.11) | 20 | 0.05 (0.03, 0.08) | 0.24 (0.03, 1.75) |
| All-cause mortality | 4615 | 12.32 (11.97, 12.68) | 9718 | 24.52 (24.04, 25.01) | 2.12 (1.92, 2.33) | 4411 | 12.25 (11.89, 12.61) | 0.98 (0.86, 1.13) | 3398 | 9.22 (8.92, 9.54) | 0.75 (0.66, 0.84) | 1440 | 3.88 (3.68, 4.09) | 0.31 (0.20, 0.47) |

CI confidence interval, REF reference group, Major CVD major cardiovascular disease, Composite outcome of stroke, heart failure and coronary artery disease.

Hazard ratio (HR) and 95% confidence interval (95% CI) were estimated by Cox regression, HR > 1 (or <1) indicates patients with COVID-19 had a higher (lower) risk of sequelae compared to the non-COVID-19 control cohort.

Unvaccinated, incomplete vaccinated, fully vaccinated and received booster dose of COVID-19 vaccines refers to patients who have received 0 dose, 1 doses, 2 doses and ≥3 doses of BioNtech or CoronaVac COVID-19 vaccines, respectively.

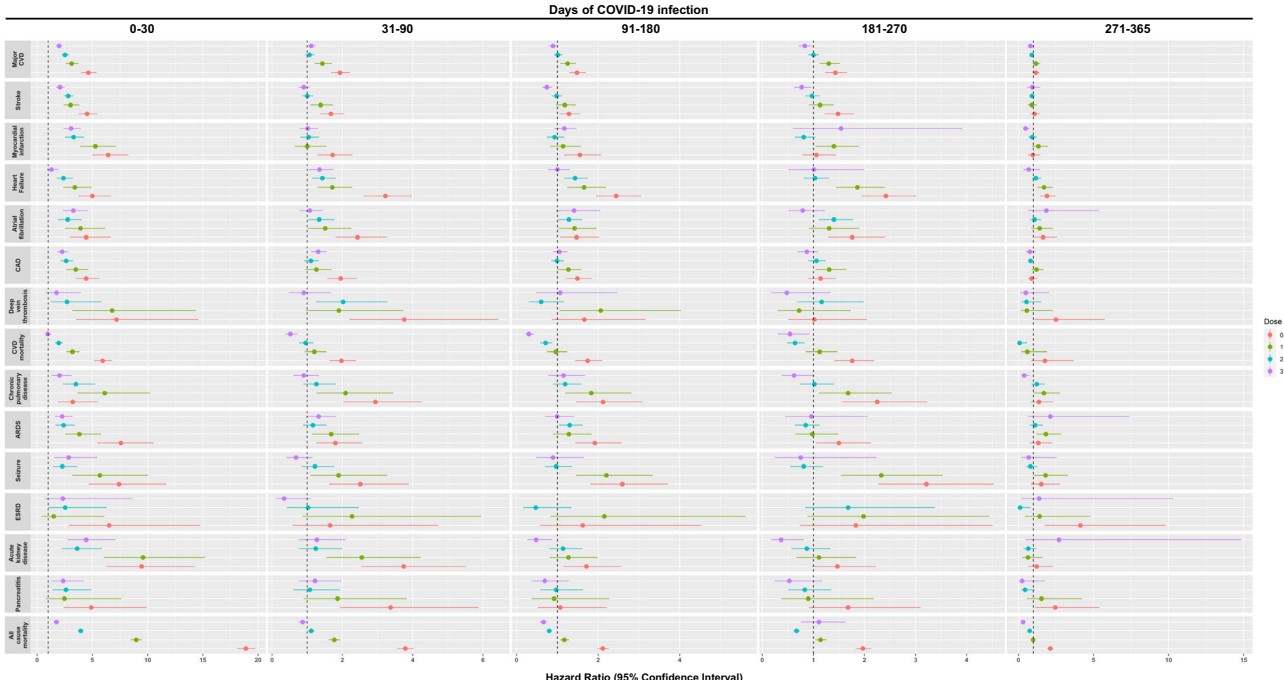

**Fig. 2 | Hazard ratio (and 95% confidence interval) of clinical sequelae compared to the control groups between 0 and 30, 31 and 90, 91 and 180, 181 and 270, 271 and 365 days of SARS-CoV-2 infection.** Note: Major CVD major cardiovascular disease, Composite outcome of stroke, heart failure and coronary artery disease, ARDS acute respiratory distress syndrome, CAD coronary artery disease, ESRD end-stage renal disease. Hazard ratio (HR) and 95% confidence interval (95% CI) were estimated by Cox regression, HR > 1 (or <1) indicates patients with COVID-19 had higher (lower) risk of sequelae compared to the non-COVID-19 control cohort. Dose 0, 1, 2, 3 refers to study population who have received 0 dose, 1 dose, 2 doses, and ≥3 doses of BioNtech or CoronaVac COVID-19 vaccines, respectively.

reduced risk of subsequent clinical sequelae beyond the acute phase of SARS-CoV-2 infection[34].

The findings of this study demonstrated a gradual reduction in the risk of long-term health consequences associated with SARS-CoV-2 over one year, indicating a lesser disease burden compared to that reported in earlier studies as well as the effect of COVID-19 vaccination in reducing the risk of clinical sequelae beyond the acute phase of SARS-CoV-2 infection. The comprehensive records of vaccination records provided by the Department of Health ensures the accuracy of information on the vaccination status of individuals reported in this study. As the pandemic progresses, our findings provided real-world evidence supporting the effectiveness of the COVID-19 vaccines in the prevention of long-term health consequences following SARS-CoV-2 infection. Nevertheless, our study is subject to several limitations. First, detection bias might be inherent in this study due to the potential under-reporting of existing underlying conditions prior to a diagnosis of SARS-CoV-2 infection. In addition, the increased healthcare contacts from receiving further examination amongst patients diagnosed with SARS-CoV-2 infection could result in the increased diagnosis of condition which could have persisted prior to infection. For instance, patients presenting with sequelae may have developed certain diseases prior to their diagnosis of SARS-CoV-2 infection; yet they did not receive a diagnosis for those conditions until a confirmed diagnosis of SARS-CoV-2 infection, resulting in the over-attribution of disease diagnosis as post-infection sequelae. Nevertheless, the history of chronic diseases in the HKHA has been recorded with high completeness as demonstrated in previous study, thus ensuring the accuracy and reliability of data to distinguish existing comorbidities and sequelae of SARS-CoV-2 infection[35]. Given the sufficiently long observation period, any existing comorbidities of subjects that were not captured is considered unlikely. Such error would also have a minimal effect on sequelae observed during the post-acute phase of infection. Furthermore, sequelae reported in this study including stroke, MI and seizure were largely of great disease severity which would typically result in distinct symptoms upon the onset of disease. Thus, the incidence of such sequelae would not be affected by the increased surveillance on patients following SARS-CoV-2 infection. Second, potential selection bias may also arise from the increased SARS-CoV-2 testing amongst individuals with prevalent comorbidities. Third, the emergence of novel variant of SARS-CoV-2 is strongly correlated with better vaccination coverage and booster dose of vaccination. Given the lower severity and risk of health consequences associated with Delta and Omicron variant emerged later in the pandemic, this could lead to potential confounding bias in our findings. Nevertheless, sensitivity analysis which adjusted for the likely variant of SARS-CoV-2 amongst the infected patients have reported a largely consistent result as the main analysis suggesting that such potential confounding bias should not impact the study's overall conclusions. Fourth, the estimation of period-specific HR is subjected to possible built-in selection bias from the censoring of patients upon the incident of clinical sequelae and the systematic difference in distribution of unknown ubiquitous factors between survivors from separate cohorts exist generally especially toward the later stage of follow-up. This could contribute to the reduction in the magnitude of HR estimated in the later observation windows. Further study is warranted to evaluate the effect of the built-in selection bias described[36,37]. Lastly, residual confounding bias can remain even after weighing subjects according to their propensity scores. Several important unmeasured confounders, namely obesity, smoking, socioeconomic status, educational level and strains of SARS-CoV-2 virus found in individual patients, could not be accounted for in this study owing to data availability, which may have introduced bias to our results.

This study examined the progressive risk of acute and post-acute sequelae following SARS-CoV-2 infection at 3 monthly intervals up to a year amongst patients with different vaccination status. The risk of clinical sequelae was observed to reduce gradually over the observation period. Complete vaccination and the uptake of booster dose of COVID-19 vaccines were found to further reduce the risk and

persistence in risk of long-term health consequences of SARS-CoV-2 infection. The findings of this study indicated a lesser disease burden caused by health consequences of SARS-CoV-2 infection compared to that reported in earlier study and provided real-world evidence supporting the effectiveness of COVID-19 vaccines in reducing the risk of long-term health consequences following infection.

## Methods

### Data source

In this retrospective cohort study, routine electronic medical records were retrieved from the Hong Kong Hospital Authority (HKHA). The Hospital Authority is a statutory body that manages all public hospitals and their ambulatory clinics in Hong Kong. The service is available to all HK residents (> 7.2 million) covering ~80% of all routine hospital admissions[38]. Electronic medical records from the HKHA database consisted of disease diagnoses recorded in planned or unplanned doctor consultations from in- and outpatient hospitals and emergency visits, thus allowing timely capture of all medical records of all users of the public health services in HK. Records were obtained from the Hong Kong Deaths Registry to identify mortality in this study. Information on vaccination status was provided by the Department of Health, The Government of Hong Kong Special Administrative Region whilst records of confirmed cases of SARS-CoV-2 infection were obtained from the Centre for Health Protection of the Government, the Hong Kong Special Administrative Region and HKHA. Anonymized unique patient identifiers were used to integrate these databases. These population-based databases have been used in previous studies on the long-term sequelae of COVID-19 infection, COVID-19 vaccines safety surveillance and effectiveness[3,6,38–42].

### Study design and population

Individuals with data linkage to electronic medical records of Hong Kong Hospital Authority from January 1, 2018 to January 23, 2023 were eligible for this study. A cohort study was conducted to evaluate the risk of health consequences between patients with and without SARS-CoV-2 infection aged 18 years or above. Patients with an incident SARS-CoV-2 infection (confirmed by rapid antigen test [RAT] or polymerase chain reaction [PCR] test in throat swab, nasopharyngeal aspirate, or deep throat sputum specimens) between April 1, 2020 and October 31, 2022 were matched to non-infected controls without a positive SARS-CoV-2 test record throughout the study period with the exact birth-year and sex. All individuals without a record of positive test record of the same birth-year and sex were selected as matched controls. Patients with SARS-CoV-2 infection were further stratified into (1) unvaccinated (0 dose), (2) incompletely vaccinated (1 dose), (3) completely (2 doses), and (4) vaccinated with booster doses (≥ 3 doses) according to the number of BioNtech or CoronaVac vaccines received prior to first SARS-CoV-2 infection. The index date of patients with SARS-CoV-2 infection was defined as the date of first diagnosis date of SARS-CoV-2 infection. The identical index date was assigned to randomly selected corresponding matched controls as the pseudo-index date.

All subjects were followed up from the index date until the date of death, the occurrence of outcome, SARS-CoV-2 re-infection or the end of the separate observation periods at 30, 90, 180, 270, and 365 days after the index date or the end of the study period January 31, 2023, whichever occurred earlier.

Anonymized longitudinal clinical healthcare data since 2016 and the earliest date of data availability were obtained for all subjects from HKHA. Relevant data included baseline demographic (sex, age and Charlson Comorbidity Index); pre-existing morbidities captured by clinical diagnosis codes (cardiovascular, cerebrovascular, respiratory, chronic kidney, liver diseases, rheumatoid arthritis and malignancy; Supplementary Table 1), history of long-term medication (renin–angiotensin-system agents, beta-blockers, calcium channel blockers, diuretics, nitrates, lipid-lowering agents, insulins, anti-diabetic drugs, oral anticoagulants, antiplatelets and immunosuppressants) and COVID-19 vaccination status before index date.

This study was reported according to the Reporting of studies Conducted using Observational Routinely-collected Data (RECORD), extended from the Strengthening the Reporting of Observational Studies in Epidemiology (STROBE) guideline.

### Outcomes of clinical diagnosis

The outcomes of this study were selected based on previous evidence on the risk of clinical sequelae associated with SARS-CoV-2 infection which includes incidences of major cardiovascular diseases (a composite outcome of stroke, heart failure and coronary heart disease), stroke, myocardial infarction (MI), heart failure, atrial fibrillation, coronary artery disease, deep vein thrombosis (DVT), chronic pulmonary disease, acute respiratory distress syndrome, seizure, end-stage renal disease, acute kidney injury, pancreatitis, cardiovascular and all-cause mortality[1,8–10,43]. Outcomes were identified based on the International Classification of Diseases, Ninth Revision, Clinical Modification (ICD-9-CM; Supplementary Table 1).

### Statistical analyses

Inverse Probability Treatment Weighting (IPTW)[44] based on age, sex, Charlson Comorbidity index (CCI), history of separate class of medication (renin–angiotensin system agents, beta-blockers, calcium channel blockers, diuretics, nitrates, lipid-lowering agents, insulins, antidiabetic drugs, oral anticoagulants, antiplatelets and immunosuppressants), the number of hospital admission and doctor consultation within one year of index date was applied to account for potential confounding factors. Standardized mean difference (SMD) between cases and controls was estimated, SMD ≤ 0.1 was regarded as sufficient balance between case and control groups[45]. Subjects with a history of outcome of interest were excluded from the analysis of the specific conditions whilst continued to be considered at risk for other disease outcomes. The incidence rate (per 1000 person-years), hazard ratio (HR) and 95% confidence interval (CI) of each outcome were estimated between COVID and non-COVID-19 cohorts separately for each of the observation period using Cox proportional hazard regressions. Sensitivity analysis was performed by only including individuals with a positive PCR SARS-CoV-2 screening test results, cases of SARS-CoV-2 infection from the Omicron wave in Hong Kong[46], unvaccinated patients with COVID-19 and matched control with the same vaccination status, adjusting for the likely variant of SARS-CoV-2 responsible for the infection, excluding patients who received their last dose of vaccine more than 6 months before SARS-CoV-2 infection owing to the waning of immunity following vaccination[47,48], and controlling for the false discovery rate at 0.05 through Benjamin-Hochberg procedure[49]. Lung cancer, brain cancer, and lymphoma which were considered to have a prolonged latent period for their development were included as negative control outcomes to detect possible testing bias. Subgroup analyses were predefined taking account of the risk factors of post-COVID-19 condition[50]. Patients were stratified by (1) age (≤65, >65), (2) sex, (3) Charlson Comorbidity index (CCI; <4, ≥4).

All statistical analyses were performed using R version 4.1.2 (R Foundation for Statistical Computing, Vienna, Austria). All significance tests were two-tailed. A P value less than 0.05 or 95% CI excluding 1.0 were taken to indicate statistical significance. At least two investigators (ICHL, RZ, and EYFW) conducted each of the statistical analyses independently for quality assurance.

### Data access

EYFW and ICKW had full access to all the data in the study and took responsibility for the integrity of the data and the accuracy of the data analysis.

### Ethical approval

Ethical approval for this study was granted by the Institutional Review Board of the University of HK/HA HK West Cluster (UW20-556 and UW21-149) and Department of Health, HK (L/M21/2021 and L/M175/2022) with an exemption for informed consent from participants as patients' confidentiality was maintained in this retrospective cohort study.

### Reporting summary

Further information on research design is available in the Nature Portfolio Reporting Summary linked to this article.

### Data availability

The data contains confidential information and hence cannot be shared with the public due to third-party use restrictions. Local academic institutions, government departments, or non-governmental organizations may apply for the access to data through the Hospital Authority's data-sharing portal (https://www3.ha.org.hk/data).

### Code availability

The code used in this study is available on Zenodo (https://doi.org/10.5281/zenodo.10132693).

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

## Acknowledgements

The authors thank the Hospital Authority for the generous provision of data for this study. This work was supported by HMRF Research on COVID-19, The Hong Kong Special Administrative Region (HKSAR) Government (Principal Investigator: EWYC; Ref No. COVID1903011); Collaborative Research Fund, University Grants Committee, the HKSAR Government (Principal Investigator: ICKW; Ref. No. C7154-20GF); and Research Grant from the Health Bureau, the HKSAR Government (Principal Investigator: I.C.K.W.; Ref. No. COVID19F01). I.C.K.W. and F.T.T.L. are partially supported by the Laboratory of Data Discovery for Health (D$^2$4H) funded by the AIR@InnoHK administered by the Innovation and Technology Commission. The funders did not have any role in design and conduct of the study; collection, management, analysis, and interpretation of the data; preparation, review, or approval of the manuscript; and decision to submit the manuscript for publication.

## Author contributions

I.C.H.L., E.Y.F.W., and I.C.K.W. had the original idea for the study, contributed to the development of the study, extracted data from the source database, constructed the study design and the statistical model, reviewed the literature, and act as guarantors for the study. I.C.H.L., R.Z., and E.Y.F.W. accessed and verified the data, and performed statistical analysis. I.C.H.L., R.Z., E.Y.F.W., and I.C.K.W. wrote the first draft of the manuscript. I.C.K.W. is the principal investigator and provided oversight for all aspects of this project. K.K.C.M., C.K.H.W., C.S.L.C., F.T.T.L., X.L., E.W.Y.C., C.S.L., E.Y.F.W., and I.C.K.W. provided critical input to the analyses, study design, and discussion. All authors contributed to the interpretation of the analysis, critically reviewed and revised the manuscript, and approved the final manuscript to be submitted.

## Competing interests

K.K.C.M. reported grants from the Hong Kong Research Grant Council, the CW Maplethorpe Fellowship, UK National Institute for Health and Care Research, European Commission Framework Horizon 2020, Innovation and Technology Commission of the Government of the Hong Kong Special Administrative Region, and personal fees from IQVIA Ltd outside the submitted work. CKHW. reports the receipt of General Research Fund, Research Grant Council, Government of Hong Kong SAR; EuroQol Research Foundation; AstraZeneca and Boehringer Ingelheim, all outside the submitted work. C.S.L.C. has received grants from the Health Bureau of the Hong Kong Government, Hong Kong Research Grant Council, Hong Kong Innovation and Technology Commission, Pfizer, IQVIA, and Amgen; and personal fees from PrimeVigilance; outside the submitted work. F.T.T.L. has been supported by the RGC Postdoctoral Fellowship under the Hong Kong Research Grants Council and has received research grants from the Health Bureau of the Government of the Hong Kong Special Administrative Region, outside the submitted work. X.L. has received research grants from Hong Kong Health and Medical Research Fund (HMRF, HMRF Fellowship Scheme, HKSAR), Research Grants Council Early Career Scheme (RGC/ECS, HKSAR), Janssen and Pfizer; internal funding from the University of Hong Kong; consultancy fees from Merck Sharp & Dohme and Pfizer, unrelated to this work. E.W.C. reports grants from Research Grants Council (RGC, Hong Kong), Research Fund Secretariat of the Food and Health Bureau, National Natural Science Fund of China, Wellcome Trust, Bayer, Bristol-Myers Squibb, Pfizer, Janssen, Amgen, Takeda, and Narcotics Division of the Security Bureau of the Hong Kong Special Administrative Region; honorarium from Hospital Authority; outside the submitted work. ICKW reports grants from Amgen, Bristol-Myers Squibb, Pfizer, Janssen, Bayer, GSK and Novartis, the Hong Kong RGC, and the Hong Kong Health and Medical Research Fund in Hong Kong, National Institute for Health Research in England, European Commission, National Health and Medical Research Council in Australia, consulting fees from IQVIA and World Health Organization, payment for expert testimony for Appeal Court of Hong Kong and is a non-executive director of Jacobson Medical in Hong Kong and Therakind in England, outside of the submitted work; no other relationships or activities that could appear to have influenced the submitted work. EYFW has received research grants from the Health Bureau of the Government of the Hong Kong Special Administrative Region, and the Hong Kong Research Grants Council, outside the submitted work. The remaining authors declare no competing interests.

## Additional information

[1]Centre for Safe Medication Practice and Research, Department of Pharmacology and Pharmacy, Li Ka Shing Faculty of Medicine, The University of Hong Kong, Hong Kong SAR, China. [2]Department of Family Medicine and Primary Care, School of Clinical Medicine, Li Ka Shing Faculty of Medicine, The University of Hong Kong, Hong Kong SAR, China. [3]Laboratory of Data Discovery for Health (D24H), Hong Kong Science and Technology Park, Sha Tin, Hong Kong SAR, China. [4]Research Department of Practice and Policy, School of Pharmacy, University College London, London, UK. [5]Centre for Medicines Optimisation Research and Education, University College London Hospitals NHS Foundation Trust, London, UK. [6]Department of Infectious Disease Epidemiology & Dynamics, Faculty of Epidemiology and Population Health, London School of Hygiene and Tropical Medicine, London, UK. [7]School of Nursing, Li Ka Shing Faculty of Medicine, The University of Hong Kong SAR, Hong Kong, China. [8]School of Public Health, Li Ka Shing Faculty of Medicine, The University of Hong Kong, Hong Kong SAR, China. [9]Advanced Data Analytics for Medical Science (ADAMS) Limited, Hong Kong, China. [10]Department of Medicine, School of Clinical Medicine, Li Ka Shing Faculty of Medicine, The University of Hong Kong, Hong Kong SAR, China. [11]Department of Medicine, The University of Hong Kong-Shenzhen Hospital, Shenzhen, China. [12]The University of Hong Kong Shenzhen Institute of Research and Innovation, Hong Kong SAR, China. [13]Division of Rheumatology and Clinical Immunology, Department of Medicine, School of Clinical Medicine, Li Ka Shing Faculty of Medicine, The University of Hong Kong, Hong Kong SAR, China. [14]Aston Pharmacy School, Aston University, Birmingham, UK. [15]These authors contributed equally: Ivan Chun Hang Lam, Ran Zhang. [16]These authors jointly supervised this work: Ian Chi Kei Wong, Eric Yuk Fai Wan. ✉e-mail: wongick@hku.hk; yfwan@hku.hk

