## [Peer Review File · Nature Communications]

Persistence in risk and effect of COVID-19 vaccination on long-term health consequences after SARS-CoV-2 infectionREVIEWER COMMENTS

Reviewer #1 (Remarks to the Author):

Lam and colleagues have conducted an interesting study that answers a really important question around effectiveness of vaccine beyond the short-term protection against the acute illness. It is quite well-done and well-reported. The main findings are: COVID-19 infection increases the risk for all-cause mortality 2-16 times over the subsequent 1 year following infection; and fully vaccinated people have reduced risk of major cardiovascular diseases and all-cause mortality compared to unvaccinated or incompletely vaccinated people in dose-dependent manner.

However, there is one big issue that needs to be rectified before this manuscript can be accepted for publication. The authors have used the terminology "long-term clinical sequelae" interchangeably with "PASC" (post-acute sequelae of COVID-19), which is not accurate. Although the definition of PASC is still being shaped, its defining symptoms are confirmed by multitude of studies from around the world. In fact, the core outcome set for PASC were determined by an international Delphi study (see ref 1). The twelve core outcomes that should be used in clinical research and reporting are: fatigue; pain; post-exertion symptoms; work or occupational and study changes; survival; recovery; and functioning, symptoms, and conditions for each of cardiovascular, respiratory, nervous system, cognitive, mental health, and physical outcomes. Lam et al did not report on these core outcomes but mainly focused on the risk for major CVDs and all-cause mortality within the one year following acute infection. So, I recommend that the authors should use the term "long term health consequences of COVID-19" in vaccinated and unvaccinated cohorts... or similar in this manuscript, and not refer to PASC or long COVID outcomes.

Reviewer #2 (Remarks to the Author):

The authors report a retrospective study comparing a large dataset of covid-19 infected patients age 18 and above during 1/4/2020 - 31/10/2022 with matched controls by age and sex adjusted using propensity score. Hazard ratio of a list of reported outcomes including all-cause mortality was calculated for several time intervals. Infected patients were stratified by vaccination status at index date in order to estimate hazard ratios for different vaccination status.

The authors showed that the risk for all-cause mortality is elevated in the infection group year follow infection though decrease with time but still elevated even year post infection and that as vaccine coverage increases the risk for post-acute sequelae decreases. The dataset is large and the questions are important especially for the vaccination effect on long term sequelae and mortality. However the current analysis raises major concerns due to potential biases , which may lead to incorrect conclusions.

Major issues:

1. Covid-19 Infected vs control comparison

I am concerned about a possible introduction of a testing bias that may strongly affect the effect of disease outcomes. Specifically I am worried about the scenario where hospitalized patients get tested more frequently than the general population (for example if each hospitalized patient gets tested upon admission to the hospital) combined with incomplete detection in the general population this may lead to a bias toward higher detection of covid-19 within patients with elevated risk for health outcomes.

The fact that a patient had a health problem may elevate the covid-19 detection probability and therefore lead to an incorrect correlation between covid-19 and the health problem.

The same bias may also occur with doctor visits instead of hospitalized.

The bias may be more emphasized during the Omicron period , where a large fraction of the population was infected with covid-19 combined with potential incomplete disease reporting.

What was the testing policy during the various covid phases ? Were all the control patients tested at least once ? routinely ?

Can authors add additional outcomes that are not expected to be covid-19 dependent such as cancer , fractures etc .. We would expect that using the same analysis on these outcomes will not show an increased risk in covid-19 patients. Is this the case ?

Can the author compare the frequency of the outcomes during the study period to the frequency in previous years ? for example the authors observed 5-8 fold elevated risk for End-stage renal disease in covid-19 patients from all groups (unvaccinated , fully vaccinate , booster) compared to controls (supplementary table 3) if this is indeed the case and given the high prevalence of covid-19 infection we would expect to see higher prevalence of End-stage renal disease in the population during the study period compared to previous years , is this the case ?

Another concern is that the selection of the controls involved peaking into the future "COVID-19 controls without a positive SARS-CoV-2 PCR test record throughout the study period" (study design) this may cause some biases ,to overcome control should be selected based only on no positive pcr test up to index date (see ref 14 in your article).

2. Vaccine group comparison

The analysis in its current form may suffer from a bias, due to different index dates of the different vaccine status groups.

I am worried that there is a correlation between time and the vaccine status.as vaccines were not available at the beginning of the study period and booster was taken a few months after the 2 dose this may lead to different index times for the different vaccine groups. As variants weaken with time and medical treatment improves with time this may lead to incorrect conclusions about the protective effect of the vaccine.

In order to overcome this potential bias and also the biases mentioned in the previous section I would recommend applying this analysis only on covid-19 infected patients after matching on the infected date (index date) between the groups (unvaccinated , 2 dose , booster). i.e. the hazard ratio will be directly calculated between each pair of vaccine group infected patients.

This analysis may provide interesting and important information about the vaccine effect on outcomes and mortality.

3. Confounders

It seems that several important confounders are missing for example smoking , obesity , number of hospitalization last year and number of doctor visit that may affect the for the risk for several outcomes including mortality if not balanced between the groups (in addition to the socioeconomic confounder that is mentioned as missing in the article) i would recommend correcting for these confounders.

Minor issues

1. Vaccinated (received ≤ 1 dose) group - the combination of the unvaccinated patients with patients receiving 1 dose is unclear.On the one hand, vaccinated with 1 dose provide some coverage, on the other hand, it is unclear why only one dose was given, is it because they were infected ? Is it because they had side effects after vaccination ? How large is this group of 1 dose ? It seems that quantitatively the authors have enough patients so it might be better to separate

these groups and maybe neglect the 1 dose group at all.

2. Variant - the analysis was performed on multiple variants. It has been shown that different variants are associated with different risks for outcomes.

Since as i understood hong kong had zero strategy till the Omicron wave maybe it is better to exclude patients that were infected before the Omicron time period.

3. Follow up - it is not clear what is the last date the cohort was followed, infected were taken till 31/10/2022. Until when they were followed ? It may be very informative to show for each group (control , Unvaccinated , fully , booster) the average following time.

4. Reinfection - was reinfection taken into account in the analysis ? If not , I would recommend stopping the followup when reinfection occurred.

5. Time from vaccine - it is informative for the infected vaccinated groups to add the average time from last vaccine , since there is a waning affect it could be that the time between vaccine booster to infection was lower that 2 dose to infection hence the difference between 2 dose and booster is due to unsimilar time difference to infection

6. Data source - it is not clear from the paragraph (data source) "in-patients electronic medical records were retrieved from the Hong 123 Kong Hospital Authority (HKHA)" and (strength and limitation) "owing to the in-patient diagnosis code used to identify disease outcomes in the HKHA". Does the data contain community patients , patients with mild covid infection that were not attending to hospital due to covid or any other reason but only visit/report his family doctor will be included in the analysis ? If included then the paragraph (data source) is a little bit confusing try to rephrase ?

7. Treatment after infection - do the authors have any information regarding medication taken after infection such as Paxlovid ? Is there a difference between the groups in the chance of taking such treatment ?

8. FDR - as the number of tests big i would recommend applying multiple hypothesis correction such as FDR.

9. Table 1,Supplementary table 2 - In the method the authors mention that for every covid patient there is matched control looking at Supplementary table 2 the number of controls is 2,649,069 where is it coming from ? if there are a total of 1,166,987 cases ?

In general the numbers in table 1 are after propensity score weighting. It might be more informative to replace Supplementary table 2 with table 1 as they reflect the study cohort before manipulation.

Reviewer #3 (Remarks to the Author):

This study by Lam, Zhang et al. evaluated the incidence of various clinical sequelae of SARS-CoV-2 infection and all-cause mortality over the course of the first year post-infection in a large population-based cohort in Hong Kong. The authors leveraged a large amount of available routine healthcare data and administrative data to estimate the protective effect of COVID-19 vaccination on these outcomes. They found that COVID-19 vaccination led to a reduction in the incidence of most evaluated clinical sequelae as well as all-cause mortality compared to a non-infected control population, with a stronger reduction with higher levels of vaccination and a decreasing risk overall with increasing time since infection. While the authors provide an extensive analysis and population-based studies including control groups are crucial to better estimate the longer-term risks of post-COVID-19 condition and clinical sequelae after SARS-CoV-2 infection for public health policy, there are several concerns that should be addressed:

Major points

1. There are four very important aspects when evaluating the potential effect of COVID-19 vaccination on post-acute sequelae of SARS-CoV-2 infection (PASC) over time, which are currently insufficiently discussed:

a. Timing of vaccination: Some existing studies have evaluated the effect of COVID-19 vaccination on incident PASC (infection and PASC onset after vaccination), while others have evaluated the effect of vaccination on prevalent PASC (infection and PASC onset prior to vaccination). The authors do not discuss this aspect and make a generalized statement that findings regarding the effect of vaccination are unknown and inconsistent, referring to two systematic reviews (Notarte et al. 2022, *eClinMed*; Byambasuren et al. 2023, *BMJ Med*). While I agree that the evidence is not fully clear to date, I think it can be stated that most prior studies have found a preventive effects of vaccination on (incident) PASC. Since the authors are primarily interested in this effect, the existing evidence should be discussed in greater detail both in the introduction and in the discussion, and the findings of the study should be discussed in the context of these previous findings. The authors may focus especially on population-based studies and consider evidence that emerged after the search window of the cited systematic reviews.

b. Reinfections: Reinfections may affect the occurrence of PASC both through acute sequelae of infection/COVID-19 and through presenting a repeated exposure that may lead to PASC in the post-acute phase. The authors do not comment on reinfections, whether such have occurred over the study timeframe, how reinfections were handled in the analysis (e.g., censoring may be appropriate), and how reinfections may have influenced the findings. This aspect needs to be explicitly addressed in the analysis and discussed by the authors.

c. Emergence of novel variants: Over the study timeframe, novel SARS-CoV-2 variants have emerged that have resulted in a large number of cases. As the authors state, the Omicron variant has been shown to result in milder disease. Various studies have evaluated the changes in the risk of PASC with novel strains and have found at least suggestive evidence that Delta and Omicron led to a lower risk of PACS. Since the emergence of these viruses is strongly correlated with better vaccination coverage and booster vaccination, this potentially leads to important confounding. While the authors also acknowledge this in the limitations, it is a point that requires attention in the analysis. Even in the absence of viral genetic data, the authors should perform a sensitivity analysis that attempts to account for the dominant strain at different infection timepoints and discuss this point in greater detail.

2. From the provided descriptions, it is unclear how the statistical analyses to calculate incidence rates and hazard ratios were conducted. Since the 1-year follow-up timeframe was separated into different periods (0-30, 31-90, 91-180, 181-270, 271-365 days), how were the incidences calculated for each of the period? For example, what was the population at risk for each period; e.g., were individuals with an incident PASC outcome in previous periods excluded for later periods? Were separate Cox regression analyses performed for each period or were period-specific hazard ratios estimated using one model including all data? Were models with time-varying hazard considered (see e.g. Taquet et al. 2022, *Lancet Psych*)? The authors have to provide more details about the estimation method for incidence rates and the models used to estimate relative effects and justify their approach. A visualization of relevant cumulative incidence curves should be provided.

Minor points

3. While perhaps often used interchangeably in the literature, I consider the distinction between COVID-19 and SARS-CoV-2 infection an important one. In the article, the authors refer both to SARS-CoV-2 infection and to COVID-19 (compared to a "non-COVID-19 group"). Since the authors also state that they used "in-patient electronic medical records" (page 6, line 122), it is unclear to me whether they indeed included all SARS-CoV-2 infected individuals in the "COVID-19" group, or only those hospitalized with COVID-19. I suspect that the former is the case, but this should be clearly reported since it does have an impact on the interpretation of the findings (COVID-19 would be those with moderate to severe disease, while the other may include mildly symptomatic

and asymptomatic individuals). I would strongly suggest to use the term "SARS-CoV-2 infection" consistently throughout the article if the comparison is between infected vs. non-infected individuals. If the comparison is between individuals with COVID-19 vs. no COVID-19, this should be explained and justified.

4. Both in introduction and discussion, the authors repeatedly and primarily refer to the work of Al-Aly and colleagues who have performed various analyses within the US Veterans Health Administration database. While their work is highly important in this context, the population underlying their study is not representative of the general population, which is an important limitation of their findings. Further evidence exists that is not discussed in the article. The authors should more comprehensively discuss the available literature, especially focusing on studies that have addressed population-based samples. This may include Ballering et al. 2022 (Lancet, [https://doi.org/10.1016/S0140-6736\(22\)01214-4](https://doi.org/10.1016/S0140-6736(22)01214-4)), Wu et al. 2022 (Sci Rep, <https://doi.org/10.1038/s41598-022-15727-0>), Ballouz et al. 2023 (BMJ, <https://doi.org/10.1136/bmj-2022-074425>), Holm et al. 2023 (Comm Med, <https://doi.org/10.1038/s43856-023-00326-5>), and Hastie et al. 2023 (Nat Comm, <https://doi.org/10.1038/s41467-023-39193-y>) as a starting point, as well as e.g. the GBD analysis (Global Burden of Disease Long COVID Collaborators 2022, JAMA). I would also like to alert the authors that they cite a study concerned with reinfection (Bowe et al. 2022, Nat Med; which is otherwise not addressed, see above), and a study that has been retracted (Huang et al. 2021, Lancet).

5. It is unclear why the timeframe from 1 Apr 2020 to 31 Oct 2022 was used for the study. To my knowledge, the first case in Hong Kong occurred in Jan 2020. Cases from Sep-Oct 2022 at this point could not yet have reached one year of follow-up. The authors should thus justify the timeframe of the study. Furthermore, they should comment on the follow-up time of participants (median follow-up, consider person-time at risk) as well as the number individuals at risk for each analysis timeframe (0-30d, 31-90d, ...).

6. It is unclear why individuals with no COVID-19 vaccination prior to infection (0 doses) and individuals with incomplete vaccination (1 dose) were pooled into one group. To enable better comparability with other studies, it would be good to see the results for these two groups separately, since even incomplete vaccination may be protective compared to no vaccination. Otherwise, the authors should justify their choice.

7. No information is provided on the linkage of routine health data. The authors should make an explicit statement that data were linked and how this was achieved (unique identifier, name & date of birth, probability matching)? The authors state that longitudinal health data used to determine prior existing (co)morbidities was "anonymized" (page 7, line 148). How could the linkage be achieved with anonymized data? The authors should also elaborate whether linkage failed for any participant and how many were excluded at this stage.

8. The matching for the comparator group of uninfected individuals ("non-COVID-19 group") is insufficiently described. What was the matching ratio (1:1, 2:1) or were all individuals with matching sex and date of birth as those in the "COVID-19 group" included? Furthermore, the authors should justify why they used sex and date of birth for the matching. Other factors may also play a role (socioeconomic environment, comorbidities), while the exact date of birth may not be the most important factor (age in years would probably be sufficient and allow for a greater number of matches so that other factors could be incorporated more easily). The authors should elaborate this in the methods and justify their choice.

9. It is not fully clear how censoring was performed. What is meant by "the latest date of the respective observation period" (page 7, line 146-147); does this refer to the total follow-up observation period for each individual after diagnosis (or pseudo-diagnosis) of infection or to the separate observation periods for calculating incidences (0-30d, 31-90d, ...)? Were individuals censored if they had any PASC outcome event (censored at the same timepoint for all outcomes), or were they censored only for the respective evaluated PASC outcome event analyzed (varying censoring for each PASC outcome)? The authors should specify this more clearly.

10. In line with the above, the authors should provide a flowchart of the selection of individuals for analysis, including any exclusions at the linkage and matching stages as well as due to missing information and attrition.

11. The authors used inverse probability treatment weighting (IPTW) to control for potential confounding, using age, sex, CCI, and medication history to estimate IPTW weights. The authors should outline how these covariables were included (age as groups or as continuous, CCI as groups or continuous, medication history as yes/no for all medications or yes/no for individual medications) in the weights estimation. As the authors also state, other confounding factors such as socioeconomic status, educational level and health-related behaviors may be further important factors. Given the detail of the described administrative data, why could this not be included at least partially e.g. through the use of social area codes? Last, SMDs are presented to evaluate balance between "case and control" groups (page 8, line 173; Table 2). Do these SMDs relate to the comparison between "non-COVID-19" and "COVID-19" groups? If so, SMDs should be evaluated and ideally presented for all relevant group comparisons made (i.e., I would also want to see how well the IPTW method performed for individuals with booster compared to incompletely vaccinated, etc.).

12. The authors state that "subjects with a history of outcome of interest were excluded from the analysis of specific conditions" (page 8, lines 173-174). Does this mean that participants with a pre-existing specific condition (e.g. myocardial infarction) were excluded from analyses for that specific PASC outcome but not for others (myocardial infarction, but not for stroke), or for all PASC outcomes (myocardial infarction, stroke, etc.)? The authors should make clearer which individuals were excluded for which outcome analysis. They should also consider presenting how many individuals (or person-time) were included in each outcome analysis.

13. The uncertainties regarding the statistical approach also impact the interpretation of the findings. For example, the authors state that the study found that "the uptake of the booster dose, was found to be effective in reducing the risk and persistence of PASC" (page 11, lines 238-239). If the authors indeed evaluated the incidence of PASC, no inference about its persistence can be made (this would be a statement about prevalence among infected over time).

14. The list of pre-existing morbidities, evaluated PASC outcomes, and the outcome list in Supplementary Table 1 do not align. Based on this, it is not possible to judge which definitions were used for the baseline characteristics and the outcome evaluations.

15. Authors state to have performed subgroup analyses based on age groups ≤ 65 years and >65 years (in addition to sex and CCI <4 vs. ≤ 4). In the results, results are reported for "patients aged above 40" (page 11, line 228). Is this a typo or were additional analyses conducted?

16. It is not fully clear how the sensitivity analysis including only individuals with a positive PCR was conducted. I assume that for SARS-CoV-2 infected individuals, the population from the primary analysis was restricted to those individuals. However, how was the control population defined? Was a new matched comparator sample selected (which could theoretically include individuals that had a positive RAT or aspirate) or was the same comparator group used as in primary analysis? The authors should make this more clear and consider presenting a population characteristics table for the sensitivity analysis in the supplementary material.

17. The discussion should be revised to make sure that all statements are supported by the findings. For example, the statement "Such findings are consistent with the existing literatures which reported a reduction in risk of PASC conditions 6 months following mild and re-infection of COVID-19" (page 12, lines 252-253) is unrelated to the finding stated in the previous sentence. Furthermore, the discussion focuses somewhat on the use of mRNA vaccinations (e.g., "The promotion of the uptake of the booster dose of the bivalent messenger RNA (mRNA) should also be encouraged" (page 13, lines 276-277)), which were not separately analyzed. Last, the strengths and limitations section contains several repetitions of the results that do not directly provide arguments on strengths and limitations (page 14, lines 297-301, lines 303-305) and could thus be shortened.

18. The authors should also check the statements related to detection bias. Especially the statement "undercapturing of COVID-19 diagnosis of asymptomatic cases could result in potential misclassification between existing co-morbidities and post-infection sequelae" (page 14, lines 306-320) is not clear to me. Perhaps I misunderstood, but if the authors indeed evaluated newly diagnosed incident PASC and exclude individuals with pre-existing conditions (as the authors state), then pre-existing co-morbidities should not influence the results? Where it does make sense is that overattribution may happen among those with diagnosed SARS-CoV-2 infection since the infection-induced healthcare contacts may result in an increased likelihood of receiving further examinations and consecutively a diagnosis of other conditions.

19. The population sizes reported in Table 2 do not align with the number of individuals included in the analysis. I assume that the numbers correspond to the pseudopopulation size derived from IPTW. The authors should make this explicit and also report the actual number of individuals (as in Supplementary Table 2).

20. The language of the article is in many places imprecise and ambiguous. While I empathize with the authors and the difficulties involved in scientific writing in foreign languages, it precludes a thorough interpretation of the methods and results of the article. This is likely reflected in my comments above. The article should be thoroughly checked for grammar, spelling, and the meaning conveyed by the wording by a native English speaker. Also consistency in terminology should be ensured (e.g. "sex" vs. "gender").

RE: NCOMMS-23-39376– The persistence in risk and effect of COVID-19 vaccination on long-term health consequences one year after SARS-CoV-2 infection: a territory-wide cohort study in Hong Kong

On behalf of the co-authors, I would like to thank the editors and the reviewers for their valuable comments and suggestions. In response, we have made the necessary adjustments in the revised manuscript, and hope that you will now find it suitable for publication. We provide here, our point-to-point responses to the earlier version of this paper reviewed.

REVIEWER COMMENTS

Reviewer #1 (Remarks to the Author):

Lam and colleagues have conducted an interesting study that answers a really important question around effectiveness of vaccine beyond the short-term protection against the acute illness. It is quite well-done and well-reported. The main findings are: COVID-19 infection increases the risk for all-cause mortality 2-16 times over the subsequent 1 year following infection; and fully vaccinated people have reduced risk of major cardiovascular diseases and all-cause mortality compared to unvaccinated or incompletely vaccinated people in dose-dependent manner.

However, there is one big issue that needs to be rectified before this manuscript can be accepted for publication. The authors have used the terminology “long-term clinical sequelae” interchangeably with “PASC” (post-acute sequelae of COVID-19), which is not accurate. Although the definition of PASC is still being shaped, its defining symptoms are confirmed by multitude of studies from around the world. In fact, the core outcome set for PASC were determined by an international Delphi study (see ref 1). The twelve core outcomes that should be used in clinical research and reporting are: fatigue; pain; post-exertion symptoms; work or occupational and study changes; survival; recovery; and functioning, symptoms, and conditions for each of cardiovascular, respiratory, nervous system, cognitive, mental health, and physical outcomes. Lam et al did not report on these core outcomes but mainly focused on the risk for major CVDs and all-cause mortality within the one year following acute infection. So, I recommend that the authors should use the term “long term health consequences of COVID-19” in vaccinated and unvaccinated cohorts... or similar in this manuscript, and not refer to PASC or long COVID outcomes.

Thank you for your reviewing and providing your insightful comments on the use of the terms PASC. Taking your suggestion into consideration, we have avoided labelling our study outcomes as post-acute sequelae of SARS-CoV-2 (PASC) and long COVID. The relevant terms used to describe clinical sequelae reported developed following SARS-CoV-2 infection has been reworded to as “health consequences” or “clinical sequelae” throughout the manuscript.

Reviewer #2 (Remarks to the Author):

The authors report a retrospective study comparing a large dataset of covid-19 infected patients age 18 and above during 1/4/2020 - 31/10/2022 with matched controls by age and sex adjusted using propensity score. Hazard ratio of a list of reported outcomes including all-cause mortality was calculated for several time intervals. Infected patients were stratified by vaccination status at index date in order to estimate hazard ratios for different vaccination status.

The authors showed that the risk for all-cause mortality is elevated in the infection group year follow infection though decrease with time but still elevated even year post infection and that as vaccine coverage increases the risk for post-acute sequelae decreases. The dataset is large and the questions are important especially for the vaccination effect on long term sequelae and mortality. However the current analysis raises major concerns due to potential biases , which may lead to incorrect conclusions.

Thank you for reviewing our manuscript and highlighting the strength and potential bias in our study. Please find our response and revision on our manuscript based on your comments.

Major issues:

1. Covid-19 Infected vs control comparison

I am concerned about a possible introduction of a testing bias that may strongly affect the effect of disease outcomes. Specifically I am worried about the scenario where hospitalized patients get tested more frequently than the general population (for example if each hospitalized patient gets tested upon admission to the hospital) combined with incomplete detection in the general population this may lead to a bias toward higher detection of covid-19 within patients with elevated risk for health outcomes.

The fact that a patient had a health problem may elevate the covid-19 detection probability and therefore lead to an incorrect correlation between covid-19 and the health problem.

The same bias may also occur with doctor visits instead of hospitalized. The bias may be more emphasized during the Omicron period , where a large fraction of the population was infected with covid-19 combined with potential incomplete disease reporting.

What was the testing policy during the various covid phases ? Were all the control patients tested at least once ? routinely ?

Thank you for your comments and questions regarding the possibility of testing bias. We would like to clarify that the Government of Hong Kong has conducted routine screening tests for COVID-19 among the general population throughout the entire pandemic period. The records of confirmed COVID-19 diagnoses were provided by the Centre for Health Protection (CHP) under the Department of Health of Hong Kong which documented all confirmed COVID-19 cases identified from both mandatory and voluntary reporting of positive results on polymerase chain reaction (PCR) taken in public hospitals and community testing centres as well as rapid antigen tests (RAT) used

in private hospitals and within the community. Therefore, records of COVID-19 identified from these databases provided comprehensive coverage of cases in Hong Kong during the entire study period from both hospital and community settings.

We acknowledge your concern about the under detection of cases in non-hospitalised patients. We would like to emphasise that the inclusion period of this study covered the height of pandemic outbreak in HK during the Omicron period when the Government of Hong Kong had introduced stringent guidelines and policy in relation to the report of positive cases of COVID-19. Frequent and regular testings for COVID-19 were encouraged for all residents in Hong Kong especially for individuals with symptoms of COVID-19 including fever, cough and sneezing.^{1,2} Free Rapid Antigen Testing (RAT) kits were also issued to all residents in HK to facilitate and encourage the regular screening and testing of potential cases of COVID-19. Patients who have tested positive for COVID-19 are also required to self-report to the CHP for the issue of home quarantine order whilst mandatory testing is required for the entry of specified premises such as hospitals, schools, and care-homes. Given the array of regulations on the testing, reporting cases of COVID-19 and easy access to the RAT self-testing kit, any under-capturing of cases of COVID-19 from the community settings should not have a major influence on the conclusion of this study.

Furthermore, we agreed that the more frequent hospital or doctor visits among patients with other co-morbidities could lead to a higher chance of COVID-19 detection and other subsequent clinical sequelae after a COVID-19 diagnosis, thus introducing selection bias in our study. To account for such potential confounding factors, we have further adjusted for the number of hospital and doctor visits of individuals one year prior to the index date in our Inverse Probability Treatment Weighting (IPTW) model along with the Charlson comorbidity index to account for the difference in the degree of morbidities between the study cohorts.

Based on the concerns you have raised regarding the data sources, potential confounding and testing bias in our study, we have made the following revision to the study design and discussed further the limitations of this study.

Data sources: *“Information on vaccination status was provided by the Department of Health, The Government of Hong Kong Special Administrative Region whilst records of confirmed cases of SARS-CoV-2 infection were obtained from the Centre for Health Protection of the Government of the Hong Kong Special Administrative Region. Anonymized unique patient identifiers were used to integrate these databases. These population-based databases have been used in previous studies on the long-term sequelae of COVID-19 infection, COVID-19 vaccines safety surveillance and effectiveness.”* (Page. 7, Lines 135-140)

Statistical analyses: *“Inverse Probability Treatment Weighting (IPTW) based on age, sex, Charlson Comorbidity index (CCI), history of separate class of medication (renin-angiotensin-system agents, beta blockers, calcium channel blockers, diuretics, nitrates, lipid lowering agents, insulins, antidiabetic drugs, oral anticoagulants, antiplatelets and immunosuppressants), the number of hospital admission and doctor consultation within one year of index date was applied to account for potential confounding factors.”* (Page. 9, Lines 177-181)

References

- [1] Cheung PH, Chan CP, Jin DY. Lessons learned from the fifth wave of COVID-19 in Hong Kong in early 2022. *Emerg Microbes Infect.* 2022 Dec;11(1):1072-1078. doi: 10.1080/22221751.
- [2] Wong SC, Au AK, Lo JY, Ho PL, Hung IF, To KK, Yuen KY, Cheng VC. Evolution and Control of COVID-19 Epidemic in Hong Kong. *Viruses.* 2022 Nov 14;14(11):2519. doi: 10.3390/v14112519.

Can authors add additional outcomes that are not expected to be covid-19 dependent such as cancer , fractures etc .. We would expect that using the same analysis on these outcomes will not show an increased risk in covid-19 patients. Is this the case ?

Thank you for your valuable comments and suggestions regarding the use of negative outcomes as controls to assess the increased risk of various health conditions resulting from COVID-19 infection. However, due to the considerable uncertainties surrounding the clinical implications following COVID-19 infection, especially those associated with the mid to long-term. It is impossible to identify clinical outcomes with a complete null association with COVID-19 infection based on the current knowledge of the clinical outcomes and health consequences of COVID-19 infection.

With considerations on the use of cancer and fracture as potential negative outcomes for this study that the reviewer kindly suggested, previous studies have indicated that COVID-19 infection may contribute to the development of certain endocrine-related diseases, resulting in alterations in parathyroid hormone levels, which in turn affect bone formation and mineral density. This could potentially elevate the risk of fractures following SARS-CoV-2 infection.^{1,2} Additionally, earlier literature has highlighted several potential pathophysiological pathways influenced by COVID-19 infection, which could contribute to an increased risk of developing cancers amongst COVID-19 survivors.³ The association between COVID-19 infection and the increased risk of cancer has also been reported in a previous mendelian randomization study.⁴

Given the uncertainty in outcomes which are not associated with COVID-19 infection which can be considered as a negative outcome for our current study, we have made the decision not to conduct such analyses to avoid the synthesis of any misleading results and messages to the readers.

References:

- [1] Mirza SA, Sheikh AAE, Barbera M, Ijaz Z, Javaid MA, Shekhar R, Pal S, Sheikh AB. COVID-19 and the Endocrine System: A Review of the Current Information and Misinformation. *Infect Dis Rep.* 2022 Mar 11;14(2):184-197. doi: 10.3390/idr14020023.
- [2] Haudenschild, AK, Christiansen, BA, Orr, S, et al. Acute bone loss following SARS-CoV-2 infection in mice. *J Orthop Res.* 2023; 41: 1945-1952. doi:10.1002/jor.25537
- [3] Jahankhani K, Ahangari F, Adcock IM, Mortaz E. Possible cancer-causing capacity of COVID-19: Is SARS-CoV-2 an oncogenic agent? *Biochimie.* 2023 Oct;213:130-138. doi: 10.1016/j.biochi.2023.
- [4] Li, J, Bai, H, Qiao, H, et al. Causal effects of COVID-19 on cancer risk: a Mendelian randomization study. *J Med Virol.* 2023; 95:e28722. doi:10.1002/jmv.28722

Can the author compare the frequency of the outcomes during the study period to the frequency in previous years ? for example the authors observed 5-8 fold elevated risk for End-stage renal disease in covid-19 patients from all groups (unvaccinated , fully vaccinate , booster) compared to controls (supplementary table 3) if this is indeed the case and given the high prevalence of covid-19 infection we would expect to see higher prevalence of End-stage renal disease in the population during the study period compared to previous years , is this the case ?

Thank you for your suggestions. We have conducted further analysis to evaluate and compare the prevalence of end-stage renal disease amongst the study population during our current study period and the pre-pandemic period between Jan 1st 2016 – 2020. As you have correctly hypothesized, the result shown below reported an over 57% increase in the prevalence of ESRD observed within the

population during the study period compared to the pre-pandemic period. This lends further support to the association of COVID-19 infection and various clinical sequelae including ESRD reported in this current study.

	Number of cases in previous years	Prevalence in previous years	Number of cases in study period	Prevalence in study period	Difference in percentage
End-stage renal disease	2,220	0.058%	3,487	0.091%	0.033%

Another concern is that the selection of the controls involved peaking into the future “COVID-19 controls without a positive SARS-CoV-2 PCR test record throughout the study period” (study design) this may cause some biases ,to overcome control should be selected based only on no positive pcr test up to index date (see ref 14 in your article).

Thank you for raising the concern and suggestions for an alternative design to address such a potential issue. In view of your concern and to assess whether such potential sampling bias could have a significant impact on the study findings, we have conducted further validation analysis based on the methodology and study design adopted by Dagan et al. The findings from the study presented below reported a largely consistent findings which led to the same conclusion as our main analysis.

References:

[1] Dagan, N. et al. BNT162b2 mRNA Covid-19 Vaccine in a Nationwide Mass Vaccination Setting. *New England Journal of Medicine* 384, 1412-1423 (2021).

Incidence rate and hazard ratio of clinical sequelae in COVID-19 and non-COVID-19 cohort after matching during 0-30, 31-90, 91-180, 181-270,271-365 days of COVID-19 infection

Clinical sequelae	Controls (REF)		COVID-19											
	Event	Incidence per 1,000 person-years	Unvaccinated			Received single dose			Received two doses			Received booster dose(s)		
			Event	Incidence per 1,000 person-years	Hazard ratio	Event	Incidence per 1,000 person-years	Hazard ratio	Event	Incidence per 1,000 person-years	Hazard ratio	Event	Incidence per 1,000 person-years	Hazard ratio
0-30 days														
Major CVD	3,381	8.99 (8.69,9.30)	368	1,037.28 (933.99,1,148.86)	8.52 (7.60,9.55)	153	676.84 (573.84,792.99)	5.56 (4.70,6.56)	303	238.93 (212.78,267.40)	1.96 (1.73,2.22)	276	235.93 (208.92,265.47)	1.94 (1.70,2.20)
Cardiovascular Mortality	1,885	4.71 (4.50,4.93)	363	853.19 (767.67,945.64)	10.59 (9.40,11.94)	85	330.99 (264.39,409.28)	4.11 (3.29,5.12)	113	84.83 (69.91,101.98)	1.05 (0.87,1.28)	50	40.81 (30.29,53.80)	0.51 (0.38,0.67)
All-cause Mortality	9,874	24.71 (24.22,25.20)	4,971	11,683.82 (11,361.25,12,013.22)	30.77 (29.58,32.01)	1,232	4,797.46 (4,533.28,5,073.02)	12.51 (11.76,13.32)	1,025	769.44 (723.05,818.02)	2.00 (1.87,2.13)	299	244.05 (217.16,273.33)	0.63 (0.56,0.71)
31-90 days														
Major CVD	5,698	8.02 (7.82,8.23)	513	24.31 (22.25,26.51)	3.09 (2.83,3.39)	242	17.55 (15.41,19.91)	2.22 (1.95,2.53)	568	7.25 (6.66,7.87)	0.92 (0.84,1.00)	567	7.83 (7.20,8.50)	0.99 (0.91,1.08)
Cardiovascular Mortality	1,979	2.62 (2.51,2.74)	437	17.50 (15.90,19.22)	6.81 (6.14,7.56)	123	7.88 (6.55,9.40)	3.07 (2.56,3.68)	172	2.09 (1.79,2.43)	0.81 (0.70,0.95)	77	1.01 (0.80,1.27)	0.39 (0.31,0.50)
All-cause Mortality	11,745	15.55 (15.27,15.84)	5,403	217.49 (211.73,223.37)	14.30 (13.84,14.77)	1,060	67.99 (63.96,72.21)	4.47 (4.20,4.76)	1,396	16.97 (16.09,17.88)	1.12 (1.06,1.18)	775	10.21 (9.51,10.96)	0.67 (0.62,0.72)
91-180 days														
Major CVD	6,438	6.96 (6.80,7.14)	559	18.56 (17.06,20.17)	2.64 (2.42,2.88)	281	13.78 (12.22,15.49)	1.93 (1.72,2.18)	742	6.47 (6.01,6.95)	0.92 (0.85,0.99)	498	5.92 (5.41,6.46)	0.89 (0.81,0.97)
Cardiovascular Mortality	1,897	1.93 (1.84,2.02)	327	9.18 (8.22,10.24)	4.69 (4.17,5.27)	87	3.77 (3.02,4.65)	1.90 (1.53,2.35)	148	1.23 (1.04,1.44)	0.62 (0.53,0.74)	41	0.47 (0.33,0.63)	0.26 (0.19,0.35)
All-cause Mortality	12,395	12.60 (12.38,12.82)	2,615	73.60 (70.80,76.47)	5.76 (5.52,6.01)	712	30.85 (28.63,33.20)	2.39 (2.21,2.57)	1,071	8.89 (8.37,9.44)	0.69 (0.65,0.74)	526	5.97 (5.47,6.51)	0.50 (0.46,0.54)
181-270 days														
Major CVD	4,969	7.72 (7.51,7.94)	472	17.13 (15.62,18.75)	2.17 (1.97,2.38)	251	12.71 (11.18,14.38)	1.70 (1.50,1.93)	626	5.95 (5.49,6.43)	0.77 (0.71,0.84)	226	5.97 (5.22,6.80)	0.80 (0.70,0.92)
Cardiovascular Mortality	1,098	1.60 (1.50,1.69)	206	6.32 (5.48,7.24)	3.99 (3.44,4.63)	66	2.95 (2.28,3.75)	1.94 (1.51,2.48)	85	0.77 (0.62,0.95)	0.49 (0.40,0.62)	14	0.36 (0.19,0.60)	0.22 (0.13,0.38)
All-cause Mortality	9,883	14.38 (14.09,14.66)	2,170	66.65 (63.87,69.51)	4.58 (4.37,4.79)	663	29.66 (27.44,32.00)	2.12 (1.96,2.30)	791	7.18 (6.69,7.70)	0.50 (0.47,0.54)	169	4.30 (3.67,5.00)	0.31 (0.26,0.36)
271-365 days														
Major CVD	2,548	6.29 (6.04,6.53)	255	12.70 (11.19,14.35)	2.06 (1.81,2.34)	126	10.07 (8.38,11.98)	1.58 (1.32,1.89)	317	4.45 (3.98,4.97)	0.72 (0.64,0.80)	92	4.51 (3.64,5.53)	0.71 (0.57,0.87)
Cardiovascular Mortality	104	0.24 (0.20,0.29)	27	1.15 (0.76,1.67)	4.90 (3.20,7.50)	4	0.28 (0.08,0.73)	1.15 (0.42,3.13)	5	0.07 (0.02,0.16)	0.29 (0.12,0.71)	0	NA	NA
All-cause Mortality	5,101	11.80 (11.48,12.13)	1,426	60.85 (57.73,64.09)	5.28 (4.97,5.60)	377	26.73 (24.10,29.57)	2.23 (2.01,2.48)	475	6.39 (5.83,6.99)	0.55 (0.50,0.60)	73	3.45 (2.71,4.34)	0.29 (0.23,0.36)

Note: CI: confidence interval, REF: Reference group; Major CVD: Major Cardiovascular Disease; Composite outcome of stroke, heart failure and coronary artery disease
Hazard Ratio (HR) and 95% Confidence Interval (95% CI) were estimated by Cox regression, HR >1 (or <1) indicates patients with COVID-19 had higher (lower) risk of sequelae compared to the non-COVID-19 control cohort
Un-vaccinated, Incomplete vaccinated, Fully Vaccinated and received booster dose of COVID-19 vaccines refers to patients who have received 0 dose, 1 doses, 2 doses and ≥3 doses of BioNtech or CoronaVac COVID-19 vaccines, respectively

2. Vaccine group comparison

The analysis in its current form may suffer from a bias, due to different index dates of the different vaccine status groups.

I am worried that there is a correlation between time and the vaccine status.as vaccines were not available at the beginning of the study period and booster was taken a few months after the 2 dose this may lead to different index times for the different vaccine groups. As variants weaken with time and medical treatment improves with time this may lead to incorrect conclusions about the protective effect of the vaccine.

In order to overcome this potential bias and also the biases mentioned in the previous section I would recommend applying this analysis only on covid-19 infected patients after matching on the infected date (index date) between the groups (unvaccinated , 2 dose , booster). i.e. the hazard ratio will be directly calculated between each pair of vaccine group infected patients.

This analysis may provide interesting and important information about the vaccine effect on outcomes and mortality.

Thank you for your comments. We acknowledged your concerns and agreed on the potential association between the number of vaccination doses received and the variant of COVID-19 causing the infection. One objective of this study is to evaluate the potential protective effect of COVID-19 vaccination against clinical sequelae following COVID-19 infection. COVID-19 vaccinations were therefore expected not to confer any protection against any clinical sequelae associated with COVID-19 amongst individuals who have not been previously infected. Such an assumption was supported by the null effectiveness and side effects from COVID-19 vaccination. Hence, we believe that comparison of patients with and without COVID-19 infection of the same vaccination status may not directly address or mitigate such potential bias in our results. We further believe that controlling the vaccination status between cases and controls could induce collider bias with the vaccination status of the individual being the collider between the variant of COVID-19 and the clinical sequelae following infection in the scenario described by Hernán et al highlighting the potential of selection bias when conditioning on a collider.¹

As detailed in previous reports on the situation of the pandemic reported in previous studies most cases of COVID-19 in Hong Kong arose during the 5th wave of the pandemic caused by the spread of the Omicron variant within the community began in January 2022, one year after the first introduction of COVID-19 vaccination in Hong Kong, it is believed that the vast majority of cases included in this study were caused by the Omicron variant of COVID-19.² Nevertheless, information on the exact variant of COVID-19 in patients infected were not currently available from our data provider. To elucidate and evaluate the effect of such possible bias caused by the potential correlation between the vaccination status and the variant of COVID-19 in the infected group, we have conducted additional sensitivity analysis by adjusting for the likely variant of COVID-19 responsible for the infection based on the reported dominant strain of SARS-CoV-2 worldwide at the date of infection. The results from the analysis showed broadly consistent results as our main analysis, indicating that such potential bias should not have a significant influence on the overall conclusion of this study.

The methods and the results of the additional sensitivity analysis have been described in the relevant section shown below and included as part of our revised manuscript under supplementary table 8.

Method: *“Sensitivity analysis was performed by only including individuals with a positive PCR SARS-CoV-2 screening test results, cases of SARS-CoV-2 infection from the Omicron wave in Hong Kong, adjusting for the likely variant of SARS-CoV-2 responsible for the infection, excluding patients who received their last dose of vaccine more than 6 months before SARS-CoV-2 infection owing to the waning of immunity following vaccination, and controlling for the false discovery rate at 0.05 through Benjamin-Hochberg procedure.”* (Page.9, Lines 186-191)

References

- [1] Hernán M A, Monge S. Selection bias due to conditioning on a collider BMJ 2023; 381 :p1135
- [2] Xie, R., Edwards, K.M., Adam, D.C. et al. Resurgence of Omicron BA.2 in SARS-CoV-2 infection-naive Hong Kong. Nat Commun 14, 2422 (2023).
- [3] Tabatabai M, Juarez PD, Matthews-Juarez P, Wilus DM, Ramesh A, Alcendor DJ, Tabatabai N, Singh KP. An Analysis of COVID-19 Mortality During the Dominancy of Alpha, Delta, and Omicron in the USA. J Prim Care Community Health. 2023 Jan-Dec;14:21501319231170164.

3. Confounders

It seems that several important confounders are missing for example smoking , obesity , number of hospitalization last year and number of doctor visit that may affect the for the risk for several outcomes including mortality if not balanced between the groups (in addition to the socioeconomic confounder that is mentioned as missing in the article) i would recommend correcting for these confounders.

Thank you for your comments and raising the possibility of under-adjustment of potential confounders in our study. We have taken your suggestion onboard and further adjusted for the number of in-patients hospital and doctor visit in our revised model to account for the difference in the probability of being diagnosed with COVID-19 from the difference in healthcare visit by patients with different level of health awareness and degree of morbidity. The findings generated from our revised model were largely consistent with that generated from our previous model which had led to the same conclusion. Unfortunately, certain potential confounders including smoking and obesity cannot be accounted for in this study as the databases used for this current study do not possess the most up-to data on the smoking status and a lack of data on Body Mass Index (BMI) of individuals. As the use of such potential incomplete or out-of-date data would undoubtedly cause misclassification of patients which could compromise the accuracy of our study findings, we have decided to withhold the use of such incomplete or out-of-date data, we have decided to withhold the use of such data and have acknowledged such potential limitation in our revised manuscript.

Based on your suggestion, we have made the following revisions to our manuscript:

Method: *“Inverse Probability Treatment Weighting (IPTW) based on age, sex and Charlson Comorbidity index (CCI) and history of separate class of medication (renin-angiotensin-system agents, beta blockers, calcium channel blockers, diuretics, nitrates, lipid lowering agents, insulins, antidiabetic drugs, oral anticoagulants, antiplatelets and immunosuppressants) and the number of in-patients hospital and doctor consultation within one year of index date of each study individual was applied to account for potential confounding factors”.* (Page. 8-9, Lines 175-180)

Strength and Limitation: *“Several important unmeasured confounders, namely socioeconomic, smoking status, obesity and strains of SARS-CoV-2 virus found in individual patient, could not be*

accounted for in this study owing to data availability, which may have introduced bias to our results.” (Page. 16, Lines 353-356)

Minor issues

1. Vaccinated (received ≤ 1 dose) group - the combination of the unvaccinated patients with patients receiving 1 dose is unclear. On the one hand, vaccinated with 1 dose provide some coverage, on the other hand, it is unclear why only one dose was given, is it because they were infected? Is it because they had side effects after vaccination? How large is this group of 1 dose? It seems that quantitatively the authors have enough patients so it might be better to separate these groups and maybe neglect the 1 dose group at all.

Thank you for your comment and suggestion. We agreed that there could be a partial protection effect against PASC conferred by the uptake of one dose of either of these vaccines and that such results would provide insightful findings to the readers. We have classified those unvaccinated and received one dose into two separate cohorts of patients for comparison in our revised the following description on the methodology and all the relevant analyses based on your suggestion.

Method: *Patients with COVID-19 infection were further stratified into (1) unvaccinated (0 dose), (2) incompletely vaccinated (1 dose), (3) completely (2 doses), and (4) vaccinated with booster doses (≥ 3 doses) according to the number of BioNtech or CoronaVac vaccines received prior to first COVID-19 infection.* (Page. 7, Lines 146-149)

2. Variant - the analysis was performed on multiple variants. It has been shown that different variants are associated with different risks for outcomes. Since as i understood hong kong had zero strategy till the Omicron wave maybe it is better to exclude patients that were infected before the Omicron time period.

Thank you for your suggestion. As you have mentioned correctly, the number of cases of COVID-19 in Hong Kong remained low prior to the 5th wave of COVID-19 which began in January 2022 owing to the success in containing the Alpha and Delta variants through its zero-COVID policy. It is reported that throughout the whole of 2021, fewer than 4,000 cases of COVID-19 have been reported.¹ Thus, it is expected that the vast majority of cases of COVID-19 included in our current study were caused by the Omicron variant of COVID-19. In view of your concern about the difference in risk of clinical sequelae associated with different strains of COVID-19, we have conducted further sensitivity analysis by excluding cases of COVID-19 prior to the Omicron wave in Hong Kong (before 31st December, 2021),² and their corresponding controls and adjusted for the likely variant of COVID-19 responsible for the infection based on the dominant strain of COVID-19 worldwide at the date of COVID-19 infection. The results from these additional sensitivity analyses were consistent between and with the main analysis suggesting that any potential confounding factors due to the difference in variant of COVID-19 should not affect the main findings of this study. The relevant description and results have been included in the following section of the revised manuscript and in Supplementary Table 8.

Method: *“Sensitivity analysis was performed by only including individuals with a positive PCR SARS-CoV-2 screening test results, cases of SARS-CoV-2 infection from the Omicron wave in Hong Kong, adjusting for the likely variant of SARS-CoV-2 responsible for the infection, excluding patients who received their last dose of vaccine more than 6 months before SARS-CoV-2 infection owing to the waning of immunity following vaccination, and controlling for the false discovery rate at 0.05 through Benjamin-Hochberg procedure.”* (Page.9, Lines 186-191)

References

- [1] Burki T. Hong Kong's fifth COVID-19 wave-the worst yet. *Lancet Infect Dis.* 2022 Apr;22(4):455-456.
- [2] Xie, R., Edwards, K.M., Adam, D.C. et al. Resurgence of Omicron BA.2 in SARS-CoV-2 infection-naive Hong Kong. *Nat Commun* 14, 2422 (2023).

3. Follow up - it is not clear what is the last date the cohort was followed, infected were taken till 31/10/2022. Until when they were followed ? It may be very informative to show for each group (control , Unvaccinated , fully , booster) the average following time.

Thank you for raising this potential source of confusion and suggestion. The last date of follow-up for all individuals was defined as the date of death, occurrence of outcome, SARS-CoV-2 re-infection events or the latest date of the respective observation period of the separate observation periods at 30, 90, 180, 270 and 365 days after the index date or the end of the study period January 31st, 2023, whichever occurs earlier. In view of the wording in the description of the study period which could lead to confusion on the study design, we have revised the wordings to clarify the end of follow-up of our study in the manuscript. In addition, we have reported the median time of follow-up for each study group in the study as per your suggestion.

Method: *“All subjects were followed up from the index date until the date of death, the occurrence of outcome, SARS-CoV-2 re-infection events or the latest date of the respective observation period end of the separate observation periods at 30, 90, 180, 270 and 365 days after the index date or the end of the study period January 31st, 2023, whichever occurred earlier.”* (Page. 7, Lines 152-154)

Results: *“The median follow-up period for controls, patients with SARS-CoV-2 who were unvaccinated, have received one, two and three or more doses of COVID-19 vaccination were 318 (173-329), 320 (253-331), 324 (312-329), 325 (312-330) and 171 (135-316) days, respectively.”* (Page. 10, Lines 209-212)

4. Reinfection - was reinfection taken into account in the analysis ? If not , I would recommend stopping the followup when reinfection occurred.

Thank you for the suggestion. Upon further investigation, it was discovered that the number of cases of SARS-CoV-2 reinfection was relatively low, accounting for only 5.5% of the total number of patients with SARS-CoV-2 infection in this study. This could be attributed to the fact that the majority of cases in Hong Kong emerged during the relatively recent wave of the COVID-19 pandemic which began in in early 2022. Nevertheless, we have taken your advice to censor patients upon re-infection of SARS-CoV-2 in the main, sensitivity and subgroup analyses of this study. The findings from the revised analysis reported consistent findings as our previous findings generated from our previous analysis which has led to the same conclusion. The following relevant section on the methodology has also been revised to reflect the changes

Method: *“All subjects were followed up from the index date until the date of death, the occurrence of outcome, SARS-CoV-2 re-infection events or the latest date of the respective observation period end of the separate observation periods at 30, 90, 180, 270 and 365 days after the index date or the end of the study period January 31st, 2023, whichever occurred earlier.”* (Page. 7, Lines 152-154)

5. Time from vaccine - it is informative for the infected vaccinated groups to add the average time from last vaccine , since there is a waning affect it could be that the time between vaccine booster to infection was lower that 2 dose to infection hence the difference between 2 dose and booster is due to unsimilar time difference to infection

Thank you for raising the possible difference in the time from the latest dose of vaccines between patients with different vaccination statuses. We have evaluated and reported the median time from the latest dose of vaccines to the date of COVID-19 infection in our results. In consideration of the waning immunity conferred by COVID-19 vaccination over time (based on previous studies), we have conducted additional sensitivity analysis by excluding patients with COVID-19 infection over 6 months after the date of the last dose of vaccination and their corresponding controls based on previous studies which reported the waning of immunity six months after vaccination.^{1,2} The findings from the additional sensitivity analysis reported broadly consistent findings, indicating that the waning effects from COVID-19 vaccination would not have a major influence on the findings of this study. The relevant changes and revisions have been reflected in the method and presented under the results in our revised manuscript and supplementary table 6 and 7.

Method: *“Sensitivity analysis was performed by only including individuals with a positive PCR SARS-CoV-2 screening test results, cases of SARS-CoV-2 infection from the Omicron wave in Hong Kong, adjusting for the likely variant of SARS-CoV-2 responsible for the infection, excluding patients who received their last dose of vaccine more than 6 months before SARS-CoV-2 infection owing to the waning of immunity following vaccination, and controlling for the false discovery rate at 0.05 through Benjamin-Hochberg procedure.”* (Page. 9, Lines 186-191)

Results: *“The median time between the latest dose of vaccination and SARS-CoV-2 infection for patients who have received one, two and three or more doses of COVID-19 vaccination was 18 (interquartile range 12-42), 175 (119-208) and 101 (37-170) days, respectively.”* (Page. 10, Lines 207-210)

References:

- [1] Feikin DR, Higdon MM, Abu-Raddad LJ et al. Duration of effectiveness of vaccines against SARS-CoV-2 infection and COVID-19 disease: results of a systematic review and meta-regression. *The Lancet* 2022; 399:924–44.
- [2] Ponticelli D, Antonazzo IC, Caci G et al. Dynamics of antibody response to BNT162b2 mRNA COVID-19 vaccine after 6 months. *J Travel Med* 2021; 28:taab173.

6. Data source - it is not clear from the paragraph (data source) “in-patients electronic medical records were retrieved from the Hong Kong Hospital Authority (HKHA)” and (strength and limitation) “owing to the in-patient diagnosis code used to identify disease outcomes in the HKHA”. Does the data contain community patients , patients with mild covid infection that were not attending to hospital due to covid or any other reason but only visit/report his family doctor will be included in the analysis ? If included then the paragraph (data source) is a little bit confusing try to rephrase ?

Thank you for raising such potential confusion on the descriptions of the data source used for this study. We would like to clarify that the records of confirmed COVID-19 diagnoses were provided by the Centre for Health Protection (CHP) under the Department of Health of Hong Kong which consisted of a comprehensive coverage of cases in Hong Kong whilst routine electronic medical records were retrieved from the Hong Kong Hospital Authority (HKHA). In view of the potential

confusion caused by the current description of the data source used in this study, we have revised the relevant section in the manuscript.

Method: *“In this retrospective cohort study, routine electronic medical records were retrieved from the Hong Kong Hospital Authority (HKHA). The Hospital Authority is a statutory body that manages all public hospitals and their ambulatory clinics in Hong Kong. The service is available to all HK residents (>7.2 million) covering approximately 80% of all routine hospital admissions. Electronic medical records from the HKHA database consisted of disease diagnoses recorded in planned or unplanned doctor consultations from in- and out-patient hospital and emergency visits, thus allowing timely capture of all medical records of all users of the public health services in HK. Records were obtained from the Hong Kong Deaths Registry to identify mortality in this study. Information on vaccination status was provided by the Department of Health, The Government of Hong Kong Special Administrative Region whilst records of confirmed cases of SARS-CoV-2 infection were obtained from the Centre for Health Protection of the Government, the Hong Kong Special Administrative Region and HKHA. Anonymized unique patient identifiers were used to integrate these databases. These population-based databases have been used in previous studies on the long-term sequelae of COVID-19 infection, COVID-19 vaccines safety surveillance and effectiveness.”* (Page.6-7, Lines 126-139)

7. Treatment after infection - do the authors have any information regarding medication taken after infection such as Paxlovid ? Is there a difference between the groups in the chance of taking such treatment ?

Thank you for the questions. Whilst we are aware that the use of oral anti-viral COVID-19 agents may prevent severe COVID-19 conditions and subsequent health consequences, its effect on the risk of post-infection clinical sequelae is beyond the scope of this study. Nevertheless, we have evaluated the number of patients who have received Paxlovid and Molnupiravir treatment, the current licensed oral antiviral COVID-19 medication in Hong Kong, following SARS-CoV-2 infection. Given the small standard mean difference (SMD) of less than 0.2 and the small number of treatment users across the study groups as shown below, the number of users for these treatments is considered balance across the study groups and should not major influence on our main findings.

Oral antiviral COVID-19 medication	Unvaccinated (N / %)	Incompletely vaccinated (N / %)	Fully vaccinated (N / %)	Received booster dose (N / %)	SMD
Paxlovid	3993 (3.2)	1618 (1.6)	9428 (2.1)	45713 (9.3)	0.188
Molnupiravir	5437 (4.4)	3735 (3.7)	6904 (1.5)	16698 (3.4)	0.088

8. FDR - as the number of tests big i would recommend applying multiple hypothesis correction such as FDR.

Thank you for raising your concern about the possible false positive discovery arising from multiple comparisons in our study. We have conducted further sensitivity analysis to control for the false discovery rate (FDR) in our study through the Benjamini-Hochberg procedure. With a false discovery level controlled at 0.05 (5%), the results of additional sensitivity analysis reported largely consistent findings where the null hypotheses were rejected in comparisons where a significant difference in risk was observed in our main analysis, thus supporting the study conclusion drawn from our main analysis.

The described method and results from the additional sensitivity analysis have been included in the revised manuscript and supplementary table 9, respectively.

Method: *“Sensitivity analysis was performed by only including individuals with a positive PCR SARS-CoV-2 screening test results, cases of SARS-CoV-2 infection from the Omicron wave in Hong Kong, adjusting for the likely variant of SARS-CoV-2 responsible for the infection, excluding patients who received their last dose of vaccine more than 6 months before SARS-CoV-2 infection owing to the waning of immunity following vaccination, and controlling for the false discovery rate at 0.05 through Benjamin-Hochberg procedure.”* (Page.9, Lines 186-191)

9. Table 1, Supplementary table 2 - In the method the authors mention that for every covid patient there is matched control looking at Supplementary table 2 the number of controls is 2,649,069 where is it coming from ? if there are a total of 1,166,987 cases ? In general the numbers in table 1 are after propensity score weighting. It might be more informative to replace Supplementary table 2 with table 1 as they reflect the study cohort before manipulation.

Thank you for your comments and questions. As you have correctly mentioned that cases and controls were initially matched by birth-year and sex. Random birth-year- and sex-matching was conducted so that the COVID-19 infection date of a randomly chosen case was mapped to each control of the same birth-year and sex and served as the pseudo index date for the control group.

Furthermore, we have presented the baseline demographics of the study population prior and after carrying out Inverse Probability Treatment Weighting (IPTW) in the main text to provide more comprehensive demographic data on the study population to the readers and demonstrate the performance and the effectiveness in minimizing the difference in these potential confounding factor through our weighting model. The baseline demographics of our study population before and after IPTW as Table 1a and Table 1b in our revised manuscript, respectively.

Reviewer #3 (Remarks to the Author):

This study by Lam, Zhang et al. evaluated the incidence of various clinical sequelae of SARS-CoV-2 infection and all-cause mortality over the course of the first year post-infection in a large population-based cohort in Hong Kong. The authors leveraged a large amount of available routine healthcare data and administrative data to estimate the protective effect of COVID-19 vaccination on these outcomes. They found that COVID-19 vaccination led to a reduction in the incidence of most evaluated clinical sequelae as well as all-cause mortality compared to a non-infected control population, with a stronger reduction with higher levels of vaccination and a decreasing risk overall with increasing time since infection. While the authors provide an extensive analysis and population-based studies including control groups are crucial to better estimate the longer-term risks of post-COVID-19 condition and clinical sequelae after SARS-CoV-2 infection for public health policy, there are several concerns that should be addressed:

Thank you for taking your time to review our manuscript. We have studied your comments and made necessary changes to the manuscript. Please find our response and the revision made based on your comments.

Major points

1. There are four very important aspects when evaluating the potential effect of COVID-19 vaccination on post-acute sequelae of SARS-CoV-2 infection (PASC) over time, which are currently insufficiently discussed:

a. Timing of vaccination: Some existing studies have evaluated the effect of COVID-19 vaccination on incident PASC (infection and PASC onset after vaccination), while others have evaluated the effect of vaccination on prevalent PASC (infection and PASC onset prior to vaccination). The authors do not discuss this aspect and make a generalized statement that findings regarding the effect of vaccination are unknown and inconsistent, referring to two systematic reviews (Notarte et al. 2022, eClinMed; Byambasuren et al. 2023, BMJ Med). While I agree that the evidence is not fully clear to date, I think it can be stated that most prior studies have found a preventive effects of vaccination on (incident) PASC. Since the authors are primarily interested in this effect, the existing evidence should be discussed in greater detail both in the introduction and in the discussion, and the findings of the study should be discussed in the context of these previous findings. The authors may focus especially on population-based studies and consider evidence that emerged after the search window of the cited systematic reviews.

Thank you for your suggestions for the improvement of the contents of our manuscript. We have considered your suggestions and expanded our discussion on the available evidence surrounding the effects of COVID-19 vaccination on the post-infection sequelae of COVID-19. The relevant pieces of evidence reported in previous population-based studies have been summarised and further discussed in both the introduction and discussion of our revised manuscript.

Introduction: *“Despite the cumulative evidence of the covid-19 vaccines’ ability to reduce disease burden during acute infection, its effect in preventing the adverse outcome in the post-acute phase of SARS-CoV-2 infection remained largely unknown owing to the inconsistent findings from the existing studies. Whilst a protective effect of COVID-19 vaccination on incident health outcomes has been reported in several population and community-based studies, the extent of risk reduction, especially from the booster dose of vaccines remains to be evaluated.”* (Page. 6, Lines 113-118)

Discussion: *“Previous studies investigating the impact of COVID-19 vaccines on the risk of developing clinical sequelae associated with SARS-CoV-2 infection have reported inconsistent findings due to the difference in the methodological approach employed to evaluate two key aspects: the effect of vaccination on the risk of incident clinical sequelae following SARS-CoV-2 infection and the prevalence of post-infection sequelae outcomes among COVID-19 survivors. Further to the current evidence supporting the protection against both acute and post-acute health consequences of SARS-CoV-2 infection conferred by COVID-19 vaccination, the findings of this study demonstrated a graded reduction in the incidence and the persistence in risk according to the number of vaccines doses received prior to infection. The protective effect was more pronounced amongst the elder population and patients with a greater degree of morbidity. Nevertheless, further studies is warranted to evaluate the effect of COVID-19 vaccination in reducing the prevalence of COVID-19 to gain deeper understanding on the potential protective effect of COVID-19 vaccination in patients who have developed onset of clinical sequelae associated with SARS-CoV-2 infection.”* (Page.13-14, Lines 291-303)

b. Reinfections: Reinfections may affect the occurrence of PASC both through acute sequelae

of infection/COVID-19 and through presenting a repeated exposure that may lead to PASC in the post-acute phase. The authors do not comment on reinfections, whether such have occurred over the study timeframe, how reinfections were handled in the analysis (e.g., censoring may be appropriate), and how reinfections may have influenced the findings. This aspect needs to be explicitly addressed in the analysis and discussed by the authors.

Thank you for raising the gap in our current study. In view of your comments and that from another reviewer, we have adapted our study design to censor patients upon re-infection of COVID-19 to account for the implication COVID-19 re-infection in our analysis. Nevertheless, the implication of re-infection on the risk of post-acute clinical sequelae is out of the scope of this current study owing to the small proportion (5.5%) of the patients in our study who experienced a subsequent reinfection of COVID-19. We have therefore refrained from discussion on this factor due to the lack of supported findings from this study and have removed any unsupported claims and content.

Method: *“All subjects were followed up from the index date until the date of death, the occurrence of outcome, SARS-CoV-2 re-infection events or the latest date of the respective observation period end of the separate observation periods at 30, 90, 180, 270 and 365 days after the index date or the end of the study period January 31st, 2023, whichever occurred earlier.”* (Page.7, Lines 152-154)

c. Emergence of novel variants: Over the study timeframe, novel SARS-CoV-2 variants have emerged that have resulted in a large number of cases. As the authors state, the Omicron variant has been shown to result in milder disease. Various studies have evaluated the changes in the risk of PASC with novel strains and have found at least suggestive evidence that Delta and Omicron led to a lower risk of PACS. Since the emergence of these viruses is strongly correlated with better vaccination coverage and booster vaccination, this potentially leads to important confounding. While the authors also acknowledge this in the limitations, it is a point that requires attention in the analysis. Even in the absence of viral genetic data, the authors should perform a sensitivity analysis that attempts to account for the dominant strain at different infection timepoints and discuss this point in greater detail.

Thank you for your comment. We agreed that there has been emergence of novel variants of COVID-19 given the extensive study period of this study and that this could correlate with the vaccination coverage which could have had a considerable implication on our findings. In response to your comment and your suggestion, we have conducted additional sensitivity analysis by adjusting for the likely variant of COVID-19 responsible for the infection. The results from the analysis were largely consistent with the main analysis, suggesting that such potential confounding bias should not impact the overall conclusion of this study. The methodology of the relevant analysis and the discussion of the findings have been included in the revised manuscript.

The methods and the results of the additional sensitivity analysis has been described in the relevant section shown below and included as part of our revised manuscript under supplementary table 8.

Method: *“Sensitivity analysis was performed by only including individuals with a positive PCR SARS-CoV-2 screening test results, cases of SARS-CoV-2 infection from the Omicron wave in Hong Kong,³³ adjusting for the likely variant of SARS-CoV-2 responsible for the infection, excluding patients who received their last dose of vaccine more than 6 months before SARS-CoV-2 infection owing to the waning of immunity following vaccination,^{34,35} and controlling for the false discovery rate at 0.05 through Benjamin-Hochberg procedure.”* (Page., Lines 186-191)

Strength and Limitation: *“Secondly, the emergence of novel variant of SARS-CoV-2 is strongly correlated with better vaccination coverage and booster dose of vaccination. Given the lower severity and risk of health consequences associated with Delta and Omicron variant emerged later in the pandemic, this could lead to potentially confounding bias in our findings. Nevertheless, sensitivity analysis which adjusted for the likely variant of SARS-CoV-2 amongst the infected patients have reported a largely consistent result as the main analysis This suggests that any potential bias should not impact the study’s overall conclusions.”* (Page. 15, Lines 344-350)

2. From the provided descriptions, it is unclear how the statistical analyses to calculate incidence rates and hazard ratios were conducted. Since the 1-year follow-up timeframe was separated into different periods (0-30, 31-90, 91-180, 181-270, 271-365 days), how were the incidences calculated for each of the period? For example, what was the population at risk for each period; e.g., were individuals with an incident PASC outcome in previous periods excluded for later periods? Were separate Cox regression analyses performed for each period or were period-specific hazard ratios estimated using one model including all data? Were models with time-varying hazard considered (see e.g. Taquet et al. 2022, Lancet Psych)? The authors have to provide more details about the estimation method for incidence rates and the models used to estimate relative effects and justify their approach. A visualization of relevant cumulative incidence curves should be provided.

Thank you for your comments and questions. We would like to clarify that our current analysis model estimates the incidence rate and the risk of clinical sequelae separately for each observation period. The incidence rate was therefore estimated by the number of event outcomes that occurred in each of the time periods. The number of individuals at risk is defined as the number of patients without a history of the specific disease outcome up to the start of the specific observation period. Individuals with a history of particular clinical sequelae from an earlier observation period will be excluded from analysis for the specific clinical sequelae in subsequent observation windows.

We agreed that the cumulative incident curves would provide a useful graphic illustration of any change in hazard over the observation window and have presented the plots as part of our manuscript.

However, it is important to note that our current study design with separate Cox regression analysis performed for every time period share the same statistical methodology in principle as a time-varying model which differs only in their defined intervals. Therefore, findings generated from either of these methods should report consistent findings which would lead to the same conclusion. A three-monthly time interval was chosen for this study based on the clinical evidence which reported a gradual improvement of lung functions and exercise capacity over one year measured at three, six, nine and twelve months following COVID-19 infection. Hence, we believe that the incorporation of a time-varying hazard model would provide limited additional value in addressing our study objectives from a clinical perspective. Based on your suggestions, the following revision on the description on the statistical analysis and the number of patients at risk have been included in our revised manuscript. The cumulative incident plot of clinical sequelae has also been presented under Supplement Figure 1.

Method: *“Subjects with a history of outcome of interest were excluded from the analysis of the specific conditions whilst continued to be considered at risk for other disease outcomes. The incidence rate (per 1,000 person-years), hazard ratio (HR) and 95% confidence interval (CI) of*

each outcome were estimated between COVID and non-COVID-19 cohorts separately for each of the observation period using Cox proportional hazard regressions.” (Page.9 , Lines 181-186)

Results: *“The number of patients at risk at each observation period between day 0-30, 31-90, 91-180, 181-270, 271-365 were 2,815,023, 2,801,396, 2,764,243, 2,119,507 and 1,876,858, respectively.” (Page.10, Lines 215-217)*

Minor points

3. While perhaps often used interchangeably in the literature, I consider the distinction between COVID-19 and SARS-CoV-2 infection an important one. In the article, the authors refer both to SARS-CoV-2 infection and to COVID-19 (compared to a "non-COVID-19 group"). Since the authors also state that they used "in-patient electronic medical records" (page 6, line 122), it is unclear to me whether they indeed included all SARS-CoV-2 infected individuals in the "COVID-19" group, or only those hospitalized with COVID-19. I suspect that the former is the case, but this should be clearly reported since it does have an impact on the interpretation of the findings (COVID-19 would be those with moderate to severe disease, while the other may include mildly symptomatic and asymptomatic individuals). I would strongly suggest to use the term "SARS-CoV-2 infection" consistently throughout the article if the comparison is between infected vs. non-infected individuals. If the comparison is between individuals with COVID-19 vs. no COVID-19, this should be explained and justified.

Thank you for your comments and the potential confusion from the inconsistent use of wording in our manuscript. As you have correctly interpreted, our COVID-19 infected cohort includes all individuals with a SARS-CoV-2 infection from both hospital and community settings defined by a positive record of SARS-CoV-2 screening test by PCR or an RAT test regardless of the severity or symptoms developed from the infection. We have revised the relevant wording used in the manuscript when referring to patients with SARS-CoV-2 infection to ensure a consistent use of wording throughout the manuscript.

4. Both in introduction and discussion, the authors repeatedly and primarily refer to the work of Al-Aly and colleagues who have performed various analyses within the US Veterans Health Administration database. While their work is highly important in this context, the population underlying their study is not representative of the general population, which is an important limitation of their findings. Further evidence exists that is not discussed in the article. The authors should more comprehensively discuss the available literature, especially focusing on studies that have addressed population-based samples. This may include Ballering et al. 2022 (Lancet, [https://doi.org/10.1016/S0140-6736\(22\)01214-4](https://doi.org/10.1016/S0140-6736(22)01214-4)), Wu et al. 2022 (Sci Rep, <https://doi.org/10.1038/s41598-022-15727-0>), Ballouz et al. 2023 (BMJ, <https://doi.org/10.1136/bmj-2022-074425>), Holm et al. 2023 (Comm Med, <https://doi.org/10.1038/s43856-023-00326-5>), and Hastie et al. 2023 (Nat Comm, <https://doi.org/10.1038/s41467-023-39193-y>) as a starting point, as well as e.g. the GBD analysis (Global Burden of Disease Long COVID Collaborators 2022, JAMA). I would also like to alert the authors that they cite a study concerned with reinfection (Bowe et al. 2022, Nat Med; which is otherwise not addressed, see above), and a study that has been retracted (Huang et al. 2021, Lancet).

Thank you for raising an important limitation on the findings from earlier studies and suggestions on the areas for improvement in our manuscript. As you have correctly highlighted previous work conducted by Al-Aly and colleagues using the findings generated using certain

databases including the US Veterans Health Administration database and discharged hospitalised patients could suffer from such underlying limitations due to the under-representativeness of the general population from their study population in terms of the frailty and disease prognosis of the population which contribute to the discrepancies on the prevalence and risk of clinical sequelae following infection. We have highlighted such limitations when citing the relevant studies in our manuscript to facilitate the interpretation of such findings by the readers as well as comprehensively discussed and compared the findings from the large-scale, nationwide population-based studies you have kindly suggested. In addition, we have further revised the list of references and removed any irrelevant and out-of-date references you have kindly pointed out.

In response to your comments and suggestions, we have included the following revision on our manuscript.

Introduction: *“Nevertheless, evidence emerged from the earlier stage of the pandemic could be limited due to the selection of under-represented samples as the study population.”* (Page.5, Lines 95-98)

Discussion: *“Nevertheless, evidence that emerged in the early stage of the pandemic has been generated based on data with limited representation of the general population such as discharged hospitalized and older patients with SARS-CoV-2 infection. Since these factors were indicative of patients with more severe disease outcomes compared to the general population, the findings from these studies may result in discrepancies in the prevalence and risk of clinical sequelae following infection. More recent evidence from several large-scale, nationwide population-based studies have indicated a gradual improvement in their recovery status characterized by the reduction in prevalence of self-reported symptoms and proportion of infected individuals reporting non-recovery from health outcomes associated with SARS-CoV-2 infection.”* (Page.12, Lines 266-274)

5. It is unclear why the timeframe from 1 Apr 2020 to 31 Oct 2022 was used for the study. To my knowledge, the first case in Hong Kong occurred in Jan 2020. Cases from Sep-Oct 2022 at this point could not yet have reached one year of follow-up. The authors should thus justify the timeframe of the study. Furthermore, they should comment on the follow-up time of participants (median follow-up, consider person-time at risk) as well as the number individuals at risk for each analysis timeframe (0-30d, 31-90d, ...).

Thank you for your question and suggestion. The medication history within 3 months of the index date was included as covariates in our propensity score model. To ensure the accuracy and consistency in identifying prescription records for each individual 3-months prior to their index date, we have opted to include only data on prescription records of individuals containing the exact date of the specific record which has only been made available for records from January 1st, 2020 onward by the Hospital Authority in Hong Kong (HKHA). Hence, the April 1st, 2020 was chosen as the start of the inclusion period to allow the prescription records within 3 months of their index date to be identified accurately. Similarly, the end of the inclusion period was chosen as three months prior to the latest date of the cut-off for data extraction on the October 23rd 2022 to ensure a minimum follow-up of approximately 3 months for all individuals in this study.

The median time of follow-up and the number of patients at risk were also been reported in the revised manuscript under the following relevant section:

Results:

“The median follow-up period for controls, patients with SARS-CoV-2 who were unvaccinated, have received one, two and three or more doses of COVID-19 vaccination were 318 (173-329), 320 (253-331), 324 (312-329), 325 (312-330) and 171 (135-316) days, respectively.” (Page.10, Lines 210-212)

“The number of patients at risk at each observation period between day 0-30, 31-90, 91-180, 181-270, 271-365 were 2,815,023, 2,801,396, 2,764,243, 2,119,507 and 1,876,858, respectively.” (Page.10, Lines 215-217)

6. It is unclear why individuals with no COVID-19 vaccination prior to infection (0 doses) and individuals with incomplete vaccination (1 dose) were pooled into one group. To enable better comparability with other studies, it would be good to see the results for these two groups separately, since even incomplete vaccination may be protective compared to no vaccination. Otherwise, the authors should justify their choice.

Thank you for your comments. Individuals who have received 0 dose and 1 dose were identified as not fully vaccinated to achieve the clinical efficacy profile in preventing primary infection and severe condition of both vaccines which has been established from their respective clinical trials as detailed in the introduction of the manuscript (section included below). Taking your kind suggestion into consideration, we have revised our study design to separate the unvaccinated cohort (received 0 dose) and incompletely vaccinated cohort (received 1 dose) in all the analyses conducted in this study.

Method: *“Patients with SARS-CoV-2 infection were further stratified into (1) unvaccinated (0 dose), (2) incompletely vaccinated (1 dose), (3) completely (2 doses), and (4) vaccinated with booster doses (≥ 3 doses) according to the number of BioNtech or CoronaVac vaccines received prior to first SARS-COV-2 infection.”* (Page. 7, Lines 146-149)

7. No information is provided on the linkage of routine health data. The authors should make an explicit statement that data were linked and how this was achieved (unique identifier, name & date of birth, probability matching)? The authors state that longitudinal health data used to determine prior existing (co)morbidities was "anonymized" (page 7, line 148). How could the linkage be achieved with anonymized data? The authors should also elaborate whether linkage failed for any participant and how many were excluded at this stage.

Thank you for raising the potential source for confusion on data source for this study. The linking of records from the separate databases used in this study was achieved through an anonymized unique patient identifier assigned to each individual in each database. The linkage can be achieved as the same anonymized unique patient identifier is assigned to each unique individual across the separate databases used in this study. The details on the methods of the record linkage from separate databases have been detailed as part of the revised manuscript.

Data source: *“In this retrospective cohort study, routine electronic medical records were retrieved from the Hong Kong Hospital Authority (HKHA). The Hospital Authority is a statutory body that manages all public hospitals and their ambulatory clinics in Hong Kong. The service is available to all HK residents (>7.2 million) covering approximately 80% of all routine hospital admissions and all patients with SARS-CoV-2 infection in HK.²⁴ Electronic medical records from the HKHA database consisted of disease diagnoses recorded in planned or unplanned doctor consultations from*

in- and out-patient hospital and emergency visits, thus allowing timely capture of all medical records of all users of the public health services in HK. Records were obtained from the Hong Kong Deaths Registry to identify mortality in this study. Information on vaccination status was provided by the Department of Health, The Government of Hong Kong Special Administrative Region whilst records of confirmed cases of SARS-CoV-2 infection were obtained from the Centre for Health Protection of the Government of the Hong Kong Special Administrative Region. Anonymized unique patient identifiers were used to integrate these databases. These population-based databases have been used in previous studies on the long-term sequelae of COVID-19 infection, COVID-19 vaccines safety surveillance and effectiveness." (Page. 6-7, Lines 126-139)

8. The matching for the comparator group of uninfected individuals ("non-COVID-19 group") is insufficiently described. What was the matching ratio (1:1, 2:1) or were all individuals with matching sex and date of birth as those in the "COVID-19 group" included? Furthermore, the authors should justify why they used sex and date of birth for the matching. Other factors may also play a role (socioeconomic environment, comorbidities), while the exact date of birth may not be the most important factor (age in years would probably be sufficient and allow for a greater number of matches so that other factors could be incorporated more easily). The authors should elaborate this in the methods and justify their choice.

Thank you for your question. Patients with COVID-19 and controls were not matched to a fixed ratio. Instead, our model separates each subject into separate strata of birth-year and sex before identifying the number of cases and control within each strata. Patients with COVID-19 are matched to the maximum number of non-COVID-19 controls within each stratum. The number of controls matched to each case would therefore differ based on the number of subjects within each stratum.

The main objective for matching targets and controls by their birth year and sex is to allow the identical index date of a target to be assigned to their corresponding controls as a pseudo-index date to the controls. It also ensures that targets are paired with controls with identical demographic risk factors in dictating the pathogenesis and severity of disease associated with COVID-19 infection as age is highly associated with the disease outcomes evaluated in this study. The year of birth was chosen as a criterion instead of age to match the exposed (patients with COVID-19) to controls due to the infeasibility of determining the age for the non-exposed controls prior to a pseudo-index date being assigned. Such limitation also limits the possibility of deriving the co-morbidities and other potential factors prior to the assignment of pseudo-index date for the controls. Other important confounding variables including the comorbidities, prescription history and the number of hospital and doctor visits prior to the index date were subsequently adjusted through the Inverse Probability Treatment Weighting (IPTW) model.

Unfortunately, certain potential confounders including the socioeconomic environment cannot be accounted for in this study as the relevant data on the residential area of individuals within the data source were insufficiently detailed to serve as a meaningful proxy for the socioeconomic status of the individual in order to mitigate the risk of re-identification of individuals.

The following details on the matching process between patients with SARS-CoV-2 infection and the comparator group as well as the limitation of this study due to unmeasured confounders has been included in the relevant section of the revised manuscript.

Method: “Patients with an incident SARS-CoV-2 infection (confirmed by rapid antigen test [RAT] or polymerase chain reaction [PCR] test in throat swab, nasopharyngeal aspirate, or deep throat sputum specimens) between April 1st, 2020 and October 31st, 2022 were matched to non-infected controls without a positive SARS-CoV-2 test record throughout the study period with the exact birth-year and sex.” (Page. 7, Lines 142-146)

The index date of patients with SARS-COV-2 infection was defined as the date of first diagnosis date of SARS-CoV-2 infection. The identical index date was assigned to their corresponding matched controls as the pseudo-index date. (Page. 7, Line. 149-151)

Strength and Limitation: “Several important unmeasured confounders, namely socioeconomic, smoking status, obesity and strains of SARS-CoV-2 virus found in individual patient, could not be accounted for in this study owing to data availability, which may have introduced bias to our results.” (Page. 6, Lines 355-357)

9. It is not fully clear how censoring was performed. What is meant by "the latest date of the respective observation period" (page 7, line 146-147); does this refer to the total follow-up observation period for each individual after diagnosis (or pseudo-diagnosis) of infection or to the separate observation periods for calculating incidences (0-30d, 31-90d, ...)? Were individuals censored if they had any PASC outcome event (censored at the same timepoint for all outcomes), or were they censored only for the respective evaluated PASC outcome event analyzed (varying censoring for each PASC outcome)? The authors should specify this more clearly.

Thank you for your question. We would like to clarify that the wording “the latest date of the respective observation period” refers to the end of the separate observation periods at 30, 90, 180, 270 and 365 days after SARS-CoV-2 infection. All subjects were followed up from the index date until the date of death, the occurrence of outcome, SARS-CoV-2 re-infection events or the latest date of the respective observation period of the separate observation periods at 30, 90, 180, 270 and 365 days after the index date or the end of the study period January 31st, 2023, whichever occurred earlier. We conducted separate survival analyses for each outcome separately, which means that the censoring varies for each post-acute clinical sequelae outcome instead of censoring at the same timepoint for all outcomes.

We have included the revised description of the follow-up period to avoid any potential confusion from the readers.

Method: “All subjects were followed up from the index date until the date of death, occurrence of outcome, SARS-CoV-2 re-infection or the end of the separate observation periods at 30, 90, 180, 270 and 365 days after the index date or the end of the study period January 31st, 2023, whichever occurred earlier.” (Page.7, Lines 152-154)

10. In line with the above, the authors should provide a flowchart of the selection of individuals for analysis, including any exclusions at the linkage and matching stages as well as due to missing information and attrition.

Thank you for your kind suggestion. We have taken your advice and included a flowchart to illustrate the selection of both patients with COVID-19 and the non-COVID-19 controls. The relevant diagram has been included as Figure 1 in our revised manuscript.

11. The authors used inverse probability treatment weighting (IPTW) to control for potential confounding, using age, sex, CCI, and medication history to estimate IPTW weights. The authors should outline how these covariables were included (age as groups or as continuous, CCI as groups or continuous, medication history as yes/no for all medications or yes/no for individual medications) in the weights estimation. As the authors also state, other confounding factors such as socioeconomic status, educational level and health-related behaviors may be further important factors. Given the detail of the described administrative data, why could this not be included at least partially e.g. through the use of social area codes? Last, SMDs are presented to evaluate balance between "case and control" groups (page 8, line 173; Table 2). Do these SMDs relate to the comparison between "non-COVID-19" and "COVID-19" groups? If so, SMDs should be evaluated and ideally presented for all relevant group comparisons made (i.e., I would also want to see how well the IPTW method performed for individuals with booster compared to incompletely vaccinated, etc.).

Thank you for your comments. We would like to clarify that age, CCI and the number of hospital and doctor visits prior to the index date were included as a continuous variable in the Inverse Probability Treatment Weighting (IPTW) model. Each individual medication included in the model was also counted as a separate variable (ie. Yes/No for individual medications listed in the manuscript and supplementary table 1). In view of the potential confusion raised on the IPTW model of this study, we have included further details in the relevant section to provide further details and clarification to the readers.

We agree with your speculation regarding the potential presence of unmeasured confounders in our study, such as socioeconomic status, educational level, and health-related behaviors of individuals. However, adjusting for individual geographical location may have limited effectiveness in accounting for these factors. The available data on the residential area of individuals within the data source were insufficiently detailed to serve as a meaningful proxy for the socioeconomic status of individuals in order to mitigate the risk of re-identification of individuals within the database. Considering the relatively small geographic area of Hong Kong, any variations in socioeconomic status across different classified areas within the population should be minimal, and adjusting for such non-confounding factors could result in further selection bias in our results. Such potential limitation due to unmeasured confounders has been acknowledged in the relevant section of the manuscript

Lastly, the SMD listed in table 1a and 1b evaluated the difference in the difference in the baseline characteristics across all five study groups. Given that the SMD for all variables were less than 0.1 after weighting, the distribution of baseline characteristics between all study groups is considered balanced after IPTW.

Based on the limitations discussed above, we have revised the following section in the manuscript to acknowledge the potential limitations of this study.

Strength and Limitation: *“Lastly, residual confounding bias can remain even after weighing subjects according to their propensity scores. Several important unmeasured confounders, namely obesity, smoking, socioeconomic status, educational level and strains of SARS-CoV-2 virus found in individual patients, could not be accounted for in this study owing to data availability, which may have introduced bias to our results.”* (Page. 16, Lines 354-357)

12. The authors state that "subjects with a history of outcome of interest were excluded

from the analysis of specific conditions" (page 8, lines 173-174). Does this mean that participants with a pre-existing specific condition (e.g. myocardial infarction) were excluded from analyses for that specific PASC outcome but not for others (myocardial infarction, but not for stroke), or for all PASC outcomes (myocardial infarction, stroke, etc.)? The authors should make clearer which individuals were excluded for which outcome analysis. They should also consider presenting how many individuals (or person-time) were included in each outcome analysis.

Thank you for the comments and highlighting the potential misleading wordings in our manuscript. We would like to clarify that patients with a history of a particular outcome were excluded only from the corresponding analyses for the specific outcome (ie. Myocardial infarction) whilst continuing to be considered at risk of other clinical sequelae (such as stroke). We have rephrased the paragraph in the manuscript and presented the person-time in each outcome analysis based on the confusion raised.

Method: *"Subjects with a history of outcome of interest were excluded from the analysis of the specific conditions whilst continued to be considered at risk for other disease outcomes."* (Page. 9, Lines 182-184)

13. The uncertainties regarding the statistical approach also impact the interpretation of the findings. For example, the authors state that the study found that "the uptake of the booster dose, was found to be effective in reducing the risk and persistence of PASC" (page 11, lines 238-239). If the authors indeed evaluated the incidence of PASC, no inference about its persistence can be made (this would be a statement about prevalence among infected over time).

Thank you for raising such potential confusion on the interpretation on our findings. We would like to clarify that the word "persistence" refers to the persistence in the risk difference between patients with SARS-CoV-2 infection and the non-infected controls. In view of such potential confusion, we have avoided the use of such term wherever possible or made clear on the message and interpretation in the main text as well as the title to avoid any confusion or misleading interpretation to the readers.

14. The list of pre-existing morbidities, evaluated PASC outcomes, and the outcome list in Supplementary Table 1 do not align. Based on this, it is not possible to judge which definitions were used for the baseline characteristics and the outcome evaluations.

Thank you for your suggestion, we have re-organised the Supplementary table1 to differentiate the disease definitions of baseline characteristics from the health outcome evaluations the list in Supplementary Table 1 based on your suggestion.

15. Authors state to have performed subgroup analyses based on age groups ≤ 65 years and >65 years (in addition to sex and CCI <4 vs. ≤ 4). In the results, results are reported for "patients aged above 40" (page 11, line 228). Is this a typo or were additional analyses conducted?

Thank you for your questions and apologies for the mistake in the manuscript. We would like to clarify that the section stated should read "patients aged above 65" instead. We have corrected the typo in the following section of the revised manuscript.

Results: *“Patients aged above 65 or with a CCI of four or above are at a greater risk of clinical sequelae during the acute and post-acute phase of SARS-CoV-2 infection especially amongst unvaccinated or incompletely vaccinated patients.”* (Page. 11, Lines 247-249)

16. It is not fully clear how the sensitivity analysis including only individuals with a positive PCR was conducted. I assume that for SARS-CoV-2 infected individuals, the population from the primary analysis was restricted to those individuals. However, how was the control population defined? Was a new matched comparator sample selected (which could theoretically include individuals that had a positive RAT or aspirate) or was the same comparator group used as in primary analysis? The authors should make this more clear and consider presenting a population characteristics table for the sensitivity analysis in the supplementary material.

Thank you for your comments and apologies for the confusion. This particular sensitivity analysis was conducted by only including patients with a positive COVID-19 screening test record identified through a laboratory-confirmed Polymerase Chain Reaction (PCR) COVID-19 screening test, with the same matched group of controls retained for the comparison. We have taken your suggestions and included a population characteristics table for each sensitivity analyses where the study population differ from the population included in the main analysis have also been included as Supplementary table 2, 4 and 6 for the manuscript.

17. The discussion should be revised to make sure that all statements are supported by the findings. For example, the statement "Such findings are consistent with the existing literatures which reported a reduction in risk of PASC conditions 6 months following mild and re-infection of COVID-19" (page 12, lines 252-253) is unrelated to the finding stated in the previous sentence. Furthermore, the discussion focuses somewhat on the use of mRNA vaccinations (e.g., "The promotion of the uptake of the booster dose of the bivalent messenger RNA (mRNA) should also be encouraged" (page 13, lines 276-277)), which were not separately analyzed. Last, the strengths and limitations section contains several repetitions of the results that do not directly provide arguments on strengths and limitations (page 14, lines 297-301, lines 303-305) and could thus be shortened.

Thank you for your suggestions. We have conducted a comprehensive review of our manuscript to identify any unsupported statements or claims made in the discussion section. Any statements in the discussion which lacked support from our findings including those that you have kindly raised have been carefully revised or removed to avoid any misleading information to the readers.

Discussion: *“A reduction in risk of post-infection clinical sequelae was also reported 6 months following their initial infection amongst individuals with a mild SARS-CoV-2 infection”* (Page. 12-13, Lines 275-277)

“The bivalent omicron-containing booster vaccine was also shown to confer broader immunity against the Omicron variant of SARS-CoV-2 and reduce hospitalizations or deaths due to SARS-CoV-2 infection.” (Page. 13, Lines 290-292)

“The findings of this study demonstrated a gradual reduction in long-term health consequences associated with SARS-COV-2 over one year, indicating a lesser disease burden compared to that reported in earlier studies as well as the effect of COVID-19 vaccination in reducing the risk and the persistence of clinical sequelae beyond the acute phase of SARS-COV-2 infection.” (Page. 14-15, Lines 322-325)

18. The authors should also check the statements related to detection bias. Especially the statement "undercapturing of COVID-19 diagnosis of asymptomatic cases could result in potential misclassification between existing co-morbidities and post-infection sequelae" (page 14, lines 306-320) is not clear to me. Perhaps I misunderstood, but if the authors indeed evaluated newly diagnosed incident PASC and exclude individuals with pre-existing conditions (as the authors state), then pre-existing co-morbidities should not influence the results? Where it does make sense is that overattribution may happen among those with diagnosed SARS-CoV-2 infection since the infection-induced healthcare contacts may result in an increased likelihood of receiving further examinations and consecutively a diagnosis of other conditions.

Thank you for raising such a potential source of confusion. We would like to clarify that this particular sentence you have highlighted described a similar situation you have mentioned in the second part of the comments where the potential increased healthcare contacts including further examination of health conditions could lead to the increased diagnosis of condition which could have persisted prior to SARS-CoV-2 infection. Given that these patients with a history of a specific disease cannot be identified and excluded from subsequent analysis, this could result in the over-attribution of disease diagnosis to the infection itself. Nevertheless, we would like to emphasize that the history of chronic diseases in the HKHA has been recorded with high completeness as demonstrated in previous study, thus ensuring the accuracy and reliability of data to distinguish existing co-morbidities and sequelae of SARS-CoV-2 infection as reported in previous study.¹ Clinical sequelae reported in this study including stroke, MI and seizure were largely of great disease severity which would typically result in distinct symptoms upon the onset of disease. Thus, the incidence of such sequelae would not be affected by the increased surveillance and healthcare contacts on patients following SARS-CoV-2 infection.

In view of confusion with the wordings used to describe the aforementioned potential bias in this study, we have revised the wording in the relevant section taking your comments into consideration:

Strength and Limitation: "In addition, the increased healthcare contacts from receiving further examination amongst patients diagnosed with SARS-CoV-2 infection could result in the increased diagnosis of condition which could have persisted prior to infection. For instance, patients presenting with sequelae may have developed certain diseases prior to their diagnosis of SARS-CoV-2 infection; yet they did not receive a diagnosis for those conditions until a confirmed diagnosis of SARS-CoV-2 infection, resulting in the over-attribution of disease diagnosis as post-infection sequelae. Nevertheless, the history of chronic diseases in the HKHA has been recorded with high completeness as demonstrated in previous study, thus ensuring the accuracy and reliability of data to distinguish existing co-morbidities and sequelae of SARS-CoV-2 infection. Given the sufficiently long observation period, any existing co-morbidities of subjects that were not captured is considered unlikely. Such error would also have a minimal effect on sequelae observed during the post-acute phase of infection. Furthermore, sequelae reported in this study including stroke, MI and seizure were largely of great disease severity which would typically result in distinct symptoms upon the onset of disease. Thus, the incidence of such sequelae would not be affected by the increased surveillance and healthcare contacts on patients following SARS-CoV-2 infection." (Page 15, Lines 332-346)

Reference

[1] Wong, M. C. S. et al. Health services research in the public healthcare system in Hong Kong: An analysis of over 1 million antihypertensive prescriptions between 2004–2007 as an example of the potential and pitfalls of using routinely collected electronic patient data. *BMC Health Services Research* 8, 138 (2008).

19. The population sizes reported in Table 2 do not align with the number of individuals included in the analysis. I assume that the numbers correspond to the pseudopopulation size derived from IPTW. The authors should make this explicit and also report the actual number of individuals (as in Supplementary Table 2).

Thank you for your suggestion. As you have correctly speculated, the population size reported in Table 2 refers to the weighted population after IPTW. In view of the potential confusion for the readers, we have reported the actual number of individuals in the manuscript and in Table 1a. Further clarification was also included in the footnote of Table 1b to emphasise one the population size reported in Table 1b refers to the pseudo-population of each stratum of patients with COVID-19 after propensity-score weighting.

Results: *“The number of individuals presented represent the pseudo-population of each stratum of patients with COVID-19 after propensity-score weighting”* (Table 1b)

20. The language of the article is in many places imprecise and ambiguous. While I empathize with the authors and the difficulties involved in scientific writing in foreign languages, it precludes a thorough interpretation of the methods and results of the article. This is likely reflected in my comments above. The article should be thoroughly checked for grammar, spelling, and the meaning conveyed by the wording by a native English speaker. Also consistency in terminology should be ensured (e.g. "sex" vs. "gender").

Thank you for raising the current issue in the language and writing of the manuscript. In view of your comment, we have thoroughly checked for any imprecision and inconsistencies in the language used in the manuscript. The revised manuscript has also been proofread by multiple co-authors and other native English speakers for any mistakes in the language and to enhance the readability and precision of the message delivered in this manuscript.

We sincerely hope that you will consider this revised manuscript favourably. Should there be further corrections or clarifications needed, we are more than happy to make the necessary changes and provide additional information. We look forward to hearing from you soon.

Yours sincerely,

Ian C. K. Wong, PhD, MSc, B.Pharm
Professor and Head, Department of Pharmacology and Pharmacy
Li Ka Shing Faculty of Medicine
The University of Hong Kong
2/F, Li Ka Shing Faculty of Medicine, Laboratory Block, Faculty of Medicine Building
21 Sassoon Road, Pokfulam, Hong Kong

REVIEWER COMMENTS

Reviewer #2 (Remarks to the Author):

Thank you for your detailed answers and efforts. Several concerns have been adequately addressed. The manuscript shows a very strong effect of long covid for many serious outcomes lasting up to 1 year (such as All cause mortality, Heart failure and major CVD). This effect is observed mostly in the unvaccinated group, whereas in the fully vaccinated and booster groups mostly only a transient effect is observed in the 0-30 days. These are very strong and interesting results however they are susceptible to potential biases and require strong proofs.

Major issues:

1. Covid-19 Infected vs control comparison

Regarding the major concern of testing bias toward patients attending hospital and clinics and therefore bias toward higher detection of covid19 within patients with elevated risk for health outcomes (major comment 1 from previous review). The authors have included better explanation of the covid testing policy and have also added hospital and doctor visits from the previous year as confounders. However I am still worried about the potential effect of testing bias. In the previous review I requested to add sanity checks that can increase the confidence in the results from the paper:

Adding negative control outcomes (such as Cancer , Fracture etc.). The purpose of this request is to show that non related covid outcomes do not show increased risk for covid infected patients being hospitalized (whereas testing bias can lead to such increase). The authors suggested that they cannot identify any such not covid related outcome. While I agree that it is impossible to be 100% sure , I believe some outcomes are very unlikely to show strong covid effect. Specifically, I would be very surprised covid can increase the risk for cancer in the periods of 0-30d, 30-90d post infection. If not cancer or fracture other outcomes that lead to hospitalization should be examined.

This testing bias can lead to higher risk especially in the closer windows to infection (such as the 0-30d) where the manuscript shows many findings, while these negative controls are not an absolute proof for the validity of the results, they can increase the confidence in the intriguing results presented in this manuscript.

2. Vaccinated status comparison

The authors addressed the concern regarding the variant vaccination status however I worried regarding the difference in the health state of the vaccinated and unvaccinated groups. As can be seen in table 1a the unvaccinated group has higher incidence of many pre existing morbidities such as (Cardiovascular disease, Diabetes ,Respiratory disease, etc.) while the authors try to correct for these differences by propensity-score weighting, there could be other health related differences that are not listed and therefore not corrected. (As the authors noted several important confounders are missing in the database).

One way to overcome this potential bias is to perform a sensitivity analysis using case and control comparison only for groups with the same vaccination status. i.e. match the unvaccinated cases to unvaccinated controls and calculate the hazard ratio on these groups.

I think this may increase confidence that the unvaccinated group do not have inherent bias toward increased risk for health outcomes.

Minor issues:

1. I could not find in the text what is the eligibility criteria for patients in the cohort.
2. In figure 1 it is written that 187 cases died before the index date. Since cases determined the index date (by date of infection) how could they die before they were infected ?

Reviewer #3 (Remarks to the Author):

I would like to thank the authors for their detailed replies and revisions to the manuscript and for their consideration of my and the other reviewers' comments. The additional clarifications helped to understand their analytical approach, and the justifications for their choices appear sensible to me. I especially also appreciate the conduct of further sensitivity analyses and presenting results for incompletely vaccinated individuals, which help to further contextualize their results.

Of course, it is unfortunate that further potentially important confounding variables (e.g. socioeconomic status, smoking, obesity) could not be adjusted for. But as I understand there is not much that can be done about it, and the consistency in results after taking healthcare use into account in IPTW increases the confidence in the results. Meanwhile, it is good that the authors clearly outline this limitation in the discussion.

While the authors have sufficiently addressed my primary concerns in this revision, I would still like to ask them to consider a few changes and clarifications in the manuscript:

1. Matching: Please clarify in the manuscript that you selected all individuals without positive test and with the same birth-year and sex which were available in the dataset as matched controls, and assigned to them the (pseudo-)index date from a randomly selected corresponding case. The current wording does not make the approach sufficiently clear in my view.
2. Population at risk for individual outcomes: Thank you for adding a corresponding statement in the first paragraph of the results. However, based on your replies I understood that the population at risk is different for each outcome (since individuals with the outcome prior to infection are excluded), so it remains unclear for me which outcome this statement relates to. If this information is included, it may be most valuable to include an additional column for the population at risk for each of the observation periods, outcomes, and groups. However, I would also agree with omitting this information if this renders the Tables too large and is deemed acceptable by the Editorial Board.
3. Analytical approach for cumulative risk & hazard ratios: Thank you also for outlining your analytical approach for different observation periods more clearly. With the chosen approach, it needs to be considered that the analysis may be subject to a built-in selection bias (see e.g. Hernan 2010, <https://doi.org/10.1097/EDE.0b013e3181c1ea43>; Bartlett 2020, <https://doi.org/10.1080/19466315.2020.1755722>). For each observation period, the analysis is conditioned on individuals being event-free up to that point ("survivors"). This makes the results difficult to interpret. There are some alternative analytical approaches that aim to overcome this issue. Meanwhile, given that the findings are consistent with the existing literature, I consider it unlikely that the chosen approach would bias the results in a way that it would majorly change the main conclusions. Nevertheless, I suggest that you either consider an alternative analytical approach as outlined in the sources above (at least in a sensitivity analysis), or clearly state this limitation and its potential implications in the corresponding section in the discussion.
4. Results, line 215: Please adjust to "The SMDs of all baseline characteristics after weighting were less than 0.1, [...]".

RE: NCOMMS-23-39376A– The persistence in risk and effect of COVID-19 vaccination on long-term health consequences one year after SARS-CoV-2 infection: a territory-wide cohort study in Hong Kong

On behalf of the co-authors, I would like to thank the editors and the reviewers for their further comments and suggestions. In response, we have made the necessary adjustments in the revised manuscript, and hope that you will now find it suitable for publication. We provide here, our point-to-point responses to the earlier version of this paper reviewed.

REVIEWER COMMENTS

Reviewer #2 (Remarks to the Author):

Thank you for your detailed answers and efforts. Several concerns have been adequately addressed. The manuscript shows a very strong effect of long covid for many serious outcomes lasting up to 1 year (such as All cause mortality, Heart failure and major CVD). This effect is observed mostly in the unvaccinated group, whereas in the fully vaccinated and booster groups mostly only a transient effect is observed in the 0-30 days. These are very strong and interesting results however they are susceptible to potential biases and require strong proofs.

Thank you for the valuable feedback and support on the revision made in response to the earlier comments. Please find the following responses to your further comments below:

Major issues:

1. Covid-19 Infected vs control comparison

Regarding the major concern of testing bias toward patients attending hospital and clinics and therefore bias toward higher detection of covid19 within patients with elevated risk for health outcomes (major comment 1 from previous review). The authors have included better explanation of the covid testing policy and have also added hospital and doctor visits from the previous year as confounders. However I am still worried about the potential effect of testing bias. In the previous review i requested to add sanity checks that can increase the confidence in the results from the paper:

Adding negative control outcomes (such as Cancer , Fracture etc.). The purpose of this request is to show that non related covid outcomes do not show increased risk for covid infected patients being hospitalized (whereas testing bias can lead to such increase). The authors suggested that they cannot identify any such not covid related outcome. While I agree that it is impossible to be 100% sure , I believe some outcomes are very unlikely to show strong covid effect. Specifically, I would be very surprised covid can increase the risk

for cancer in the periods of 0-30d, 30-90d post infection. If not cancer or fracture other outcomes that lead to hospitalization should be examined.

This testing bias can lead to higher risk especially in the closer windows to infection (such as the 0-30d) where the manuscript shows many findings, while these negative controls are not an absolute proof for the validity of the results, they can increase the confidence in the intriguing results presented in this manuscript.

Thank you for your further comments and suggestions on the use of negative outcomes as controls for potential testing bias in patients with COVID-19. Although we believe that the possible clinical consequences of COVID-19 remain largely unknown and that any clinical outcomes which have a completely null association with infection cannot be identified based on the current knowledge, we have considered your suggestion and analysed the risk of a number of cancer outcomes as negative control outcomes to evaluate the possible testing bias in this study. The results of the analysis presented below showed an increased incidence of lung cancer and lymphoma amongst patients with COVID-19 during the acute and early post-acute phase of infection which may indicate a level of testing bias in our findings. Given that COVID-19 is primarily a respiratory related infection, heightened screening and testing focused on lung and respiratory function, particularly during the early stages of infection, may result in an elevated detection of respiratory-related diseases, including lung cancer, among COVID-19 patients. However, the marginal increased risk observed in other types of cancer suggests that this potential bias primarily impacts the risk of respiratory-related diseases.

Additionally, although statistical significance in the increase in risk of these clinical sequelae was observed, the actual number of events and incidence rate observed remained relatively low, with only 2,558, 66, and 1,096 cases of incident lung cancer, brain cancer and lymphoma observed within the whole study period, respectively. This is further reflected in the wide 95% confidence interval of the results. Consequently, the reliability of these findings in assessing the potential testing bias is limited due to the small number of events analyzed. Nonetheless, given that we could not completely exclude the possibility of testing bias arise from the increase clinical attention and screening test following the diagnosis of COVID-19, we have acknowledged such potential bias in our study as a limitation of this study.

Discussion: *“In addition, the increased healthcare contacts from receiving further examination amongst patients diagnosed with SARS-CoV-2 infection could result in the increased diagnosis of condition which could have persisted prior to infection. For instance, patients presenting with sequelae may have developed certain diseases prior to their diagnosis of SARS-CoV-2 infection; yet they did not receive a diagnosis for those conditions until a confirmed diagnosis of SARS-CoV-2 infection, resulting in the over-attribution of disease diagnosis as post-infection sequelae.”* (Page 15, Lines 337-343)

Considering that a suitable negative control outcome should have a similar confounding structure as the outcome of interest, the difference in the risk factors between cancer and the outcomes of interest have poses uncertainty regarding the use of cancer as negative controls. In view of the uncertainty around the possible clinical outcomes associated with COVID-19 infection, including the use of cancers as negative outcome and the reliability of our findings due to the limited number of events observed as mentioned above, we have withheld reporting such findings in this study to avoid any potential misleading information to the readers.

Incidence rate and hazard ratio of clinical sequelae in COVID-19 and non-COVID-19 cohort after weighting during 0-30, 31-90, 91-180, 181-270, 271-365 days of COVID-19 infection

Clinical sequelae	Controls (REF)		COVID-19											
	Event	Incidence per 1,000 person-years	Unvaccinated			Received single dose			Received two doses			Received booster dose(s)		
			Event	Incidence per 1,000 person-years	Hazard ratio	Event	Incidence per 1,000 person-years	Hazard ratio	Event	Incidence per 1,000 person-years	Hazard ratio	Event	Incidence per 1,000 person-years	Hazard ratio
0-30 days														
Lung Cancer	221	0.68 (0.60,0.78)	384	1.21 (1.10,1.34)	1.77 (1.08,2.90)	312	0.98 (0.87,1.09)	1.43 (0.79,2.58)	297	0.92 (0.82,1.03)	1.35 (0.90,2.03)	247	0.76 (0.67,0.86)	1.12 (0.70,1.77)
Brain Cancer	12	0.04 (0.02,0.06)	30	0.09 (0.06,0.13)	2.44 (0.31,19.42)	42	0.13 (0.10,0.18)	3.46 (0.43,27.75)	9	0.03 (0.01,0.05)	0.74 (0.09,5.95)	19	0.06 (0.03,0.09)	1.52 (0.32,7.19)
Lymphoma	112	0.35 (0.29,0.42)	309	0.97 (0.87,1.09)	2.81 (1.46,5.41)	114	0.36 (0.30,0.43)	1.03 (0.40,2.68)	183	0.57 (0.49,0.65)	1.64 (0.98,2.73)	205	0.63 (0.55,0.72)	1.82 (1.06,3.11)
31-90 days														
Lung Cancer	505	0.80 (0.73,0.87)	1,102	1.78 (1.68,1.89)	2.23 (1.60,3.11)	609	0.97 (0.90,1.05)	1.22 (0.75,1.98)	716	1.14 (1.06,1.22)	1.43 (1.10,1.86)	483	0.76 (0.70,0.84)	0.96 (0.72,1.29)
Brain Cancer	17	0.03 (0.02,0.04)	62	0.10 (0.08,0.13)	3.67 (0.77,17.44)	0	NA	NA	0	NA	NA	7	0.01 (0.00,0.02)	0.38 (0.05,2.91)
Lymphoma	243	0.38 (0.34,0.43)	319	0.51 (0.46,0.57)	1.34 (0.68,2.66)	206	0.33 (0.29,0.38)	0.86 (0.36,2.06)	345	0.55 (0.49,0.61)	1.43 (0.99,2.06)	396	0.62 (0.56,0.69)	1.63 (1.11,2.38)
91-180 days														
Lung Cancer	844	1.03 (0.96,1.10)	1,236	1.43 (1.35,1.51)	1.38 (1.00,1.92)	1,258	1.38 (1.30,1.46)	1.34 (0.97,1.84)	1,021	1.14 (1.07,1.21)	1.10 (0.89,1.37)	562	0.82 (0.75,0.89)	0.79 (0.60,1.04)
Brain Cancer	20	0.02 (0.02,0.04)	0	NA	0.00 (0.00,0.00)	0	NA	NA	10	0.01 (0.01,0.02)	0.47 (0.06,3.70)	22	0.03 (0.02,0.05)	1.27 (0.36,4.50)
Lymphoma	288	0.35 (0.31,0.39)	571	0.66 (0.61,0.71)	1.86 (1.10,3.15)	285	0.31 (0.28,0.35)	0.88 (0.46,1.68)	397	0.44 (0.40,0.48)	1.24 (0.88,1.74)	341	0.49 (0.44,0.55)	1.43 (0.96,2.15)
181-270 days														
Lung Cancer	619	0.96 (0.89,1.04)	876	1.33 (1.25,1.42)	1.39 (0.98,1.96)	917	1.35 (1.27,1.44)	1.41 (1.00,1.98)	654	0.98 (0.91,1.06)	1.02 (0.79,1.32)	961	1.72 (1.61,1.83)	1.79 (0.82,3.89)
Brain Cancer	14	0.02 (0.01,0.04)	36	0.06 (0.04,0.08)	2.47 (0.32,19.32)	49	0.07 (0.05,0.10)	3.25 (0.70,14.99)	20	0.03 (0.02,0.05)	1.35 (0.35,5.20)	0	NA	NA
Lymphoma	267	0.41 (0.37,0.46)	209	0.32 (0.28,0.36)	0.77 (0.38,1.55)	234	0.34 (0.30,0.39)	0.83 (0.43,1.60)	288	0.43 (0.38,0.48)	1.04 (0.72,1.50)	241	0.42 (0.37,0.48)	1.04 (0.64,1.69)
271-365 days														
Lung Cancer	261	0.70 (0.62,0.79)	306	0.77 (0.69,0.86)	1.20 (0.69,2.09)	334	0.93 (0.83,1.03)	1.31 (0.75,2.29)	366	1.00 (0.90,1.11)	1.42 (1.02,1.99)	162	0.45 (0.39,0.52)	0.63 (0.33,1.19)
Brain Cancer	6	0.02 (0.01,0.03)	0	NA	NA	0	NA	NA	0	NA	NA	0	NA	NA
Lymphoma	140	0.38 (0.32,0.44)	207	0.52 (0.46,0.60)	1.50 (0.70,3.20)	113	0.32 (0.26,0.38)	0.83 (0.33,2.08)	115	0.31 (0.26,0.37)	0.83 (0.46,1.47)	28	0.07 (0.05,0.11)	0.20 (0.05,0.81)

Note: CI: confidence interval, REF: Reference group; Major CVD: Major Cardiovascular Disease; Composite outcome of stroke, heart failure and coronary artery disease
Hazard Ratio (HR) and 95% Confidence Interval (95% CI) were estimated by Cox regression, HR >1 (or <1) indicates patients with COVID-19 had higher (lower) risk of sequelae compared to the non-COVID-19 control cohort
Un-vaccinated, Incomplete vaccinated, Fully Vaccinated and received booster dose of COVID-19 vaccines refers to patients who have received 0 dose, 1 doses, 2 doses and ≥3 doses of BioNtech or CoronaVac COVID-19 vaccines, respectively

2. Vaccinated status comparison

The authors addressed the concern regarding the variant vaccination status however i worried regarding the difference in the health state of the vaccinated and unvaccinated groups. As can be seen in table 1a the unvaccinated group has higher incidence of many pre existing morbidities such as (Cardiovascular disease, Diabetes ,Respiratory disease, etc.) while the authors try to correct for these differences by propensity-score weighting, there could be other health related differences that are not listed and therefore not corrected. (As the authors noted several important confounders are missing in the database).

One way to overcome this potential bias is to perform a sensitivity analysis using case and control comparison only for groups with the same vaccination status. i.e. match the unvaccinated cases to unvaccinated controls and calculate the hazard ratio on these groups.

I think this may increase confidence that the unvaccinated group do not have inherent bias toward increased risk for health outcomes.

Thank you for the further elaboration on the concerns in the difference in the baseline demographics between patients of different vaccination status and your suggestion to account for such potential bias in our analysis. Taking your further advice into consideration, we have conducted additional sensitivity analysis by matching unvaccinated cases to unvaccinated controls as the study population. The findings of the analyses showed below reported consistent findings as our main analysis which has led to the same conclusion. This lends further support that the current propensity-score weighting model is sufficient in minimizing any selection bias caused by the health-related differences across separate study cohorts.

The results of the current sensitivity analysis were presented in supplementary tables 6-7 and summarized below. The description of the added sensitivity analysis was included in the method of the revised manuscript.

Method: *“Sensitivity analysis was performed by only including individuals with a positive PCR SARS-CoV-2 screening test results, cases of SARS-CoV-2 infection from the Omicron wave in Hong Kong, unvaccinated patients with COVID-19 and matched control with the same vaccination status, adjusting for the likely variant of SARS-CoV-2 responsible for the infection, excluding patients who received their last dose of vaccine more than 6 months before SARS-CoV-2 infection owing to the waning of immunity following vaccination, and controlling for the false discovery rate at 0.05 through Benjamin-Hochberg procedure.”* (Page 9, Lines 190-197)

Incidence rate and hazard ratio of clinical sequelae in COVID-19 and non-COVID-19 cohort after weighting during 0-30, 31-90, 91-180, 181-270,271-365 days of COVID-19 infection (unvaccinated patients with COVID-19 and matched control with the same vaccination status)

Clinical sequelae	Day 0-30					Day 31-90				
	Non-COVID-19		COVID-19			Non-COVID-19		COVID-19		
	Event	Incidence per 1,000 person-years (95% CI)	Event	Incidence per 1,000 person-years (95% CI)	Hazard Ratio ^a (95% CI)	Event	Incidence per 1,000 person-years (95% CI)	Event	Incidence per 1,000 person-years (95% CI)	Hazard Ratio ^a (95% CI)
Major CVD	68	12.07 (9.50,15.26)	309	36.96 (32.99,41.24)	3.03 (2.29,4.01)	127	11.40 (9.58,13.56)	274	17.29 (15.31,19.41)	1.52 (1.21,1.89)
Stroke	42	6.73 (4.87,8.95)	188	20.64 (17.86,23.76)	3.02 (2.12,4.31)	72	5.91 (4.69,7.44)	147	8.51 (7.20,9.95)	1.44 (1.06,1.95)
Myocardial infarction	22	3.33 (2.17,4.96)	123	12.38 (10.37,14.75)	3.67 (2.32,5.79)	39	3.00 (2.18,4.06)	77	4.11 (3.29,5.13)	1.37 (0.91,2.06)
Heart Failure	25	3.83 (2.59,5.62)	85	8.77 (7.07,10.80)	2.27 (1.43,3.62)	54	4.18 (3.16,5.39)	143	7.81 (6.61,9.17)	1.86 (1.33,2.61)
Atrial fibrillation	3	0.53 (0.17,1.35)	44	4.53 (3.31,6.00)	8.42 (2.88,24.58)	28	2.21 (1.52,3.17)	63	3.43 (2.65,4.36)	1.55 (0.97,2.49)
Coronary Artery Disease	26	4.12 (2.81,6.02)	126	13.39 (11.22,15.90)	3.22 (2.05,5.05)	57	4.58 (3.49,5.87)	118	6.63 (5.53,7.93)	1.45 (1.03,2.04)
Deep vein thrombosis	7	0.98 (0.41,1.91)	15	1.52 (0.90,2.45)	1.54 (0.47,5.03)	8	0.57 (0.26,1.08)	20	1.07 (0.68,1.62)	1.87 (0.81,4.33)
Cardiovascular mortality	152	22.24 (18.94,26.01)	467	45.96 (41.88,50.22)	2.05 (1.67,2.51)	141	10.52 (8.88,12.36)	194	10.13 (8.76,11.61)	0.96 (0.76,1.22)
Chronic pulmonary disease	6	0.87 (0.33,1.77)	23	2.34 (1.50,3.43)	2.67 (0.80,8.91)	14	1.10 (0.65,1.82)	41	2.25 (1.65,3.04)	2.04 (1.02,4.11)
Acute respiratory distress syndrome	7	1.03 (0.42,1.93)	64	6.43 (5.02,8.18)	6.16 (2.59,14.69)	22	1.64 (1.04,2.43)	44	2.35 (1.74,3.13)	1.43 (0.84,2.42)
Seizure	6	0.85 (0.32,1.72)	32	3.23 (2.27,4.52)	3.78 (1.41,10.14)	15	1.15 (0.69,1.87)	26	1.39 (0.95,2.02)	1.21 (0.58,2.53)
End-stage renal disease	5	0.79 (0.32,1.71)	17	1.68 (1.05,2.69)	2.12 (0.78,5.75)	5	0.36 (0.12,0.77)	6	0.34 (0.15,0.68)	0.95 (0.29,3.08)
Acute kidney injury	5	0.78 (0.33,1.73)	54	5.43 (4.15,7.06)	6.97 (2.64,18.40)	18	1.36 (0.81,2.06)	47	2.49 (1.83,3.26)	1.83 (1.04,3.25)
Pancreatitis	1	0.13 (0.00,0.54)	14	1.40 (0.83,2.32)	10.36 (1.34,80.17)	2	0.16 (0.05,0.54)	18	0.92 (0.56,1.43)	5.69 (1.66,19.56)
All-cause mortality	835	121.86 (113.77,130.30)	7,445	732.67 (716.13,749.42)	5.97 (5.50,6.49)	813	60.82 (56.72,65.08)	1,950	101.90 (97.42,106.47)	1.67 (1.52,1.83)

Incidence rate and hazard ratio of clinical sequelae in COVID-19 and non-COVID-19 cohort after weighting during 0-30, 31-90, 91-180, 181-270,271-365 days of COVID-19 infection (unvaccinated patients with COVID-19 and matched control with the same vaccination status; continued)

Clinical sequelae	Day 91-180					Day 181-270				
	Non-COVID-19		COVID-19			Non-COVID-19		COVID-19		
	Event	Incidence per 1,000 person-years (95% CI)	Event	Incidence per 1,000 person-years (95% CI)	Hazard Ratio ^a (95% CI)	Event	Incidence per 1,000 person-years (95% CI)	Event	Incidence per 1,000 person-years (95% CI)	Hazard Ratio ^a (95% CI)
Major CVD	159	9.84 (8.37,11.43)	275	12.60 (11.16,14.13)	1.28 (1.04,1.56)	173	11.21 (9.65,13.00)	261	13.34 (11.78,15.01)	1.19 (0.97,1.46)
Stroke	94	5.31 (4.31,6.46)	144	6.06 (5.15,7.13)	1.14 (0.87,1.50)	105	6.24 (5.14,7.53)	154	7.24 (6.17,8.46)	1.16 (0.89,1.51)
Myocardial infarction	45	2.37 (1.73,3.12)	72	2.76 (2.18,3.46)	1.17 (0.78,1.74)	44	2.43 (1.81,3.25)	59	2.55 (1.97,3.28)	1.05 (0.70,1.58)
Heart Failure	61	3.28 (2.56,4.21)	124	4.87 (4.07,5.79)	1.48 (1.09,2.02)	75	4.22 (3.36,5.28)	130	5.70 (4.77,6.74)	1.35 (0.99,1.85)
Atrial fibrillation	36	1.93 (1.37,2.64)	58	2.30 (1.75,2.93)	1.19 (0.76,1.87)	45	2.54 (1.86,3.34)	56	2.47 (1.87,3.17)	0.97 (0.63,1.50)
Coronary Artery Disease	64	3.57 (2.75,4.50)	120	4.90 (4.08,5.84)	1.37 (0.99,1.91)	59	3.47 (2.67,4.44)	96	4.35 (3.53,5.27)	1.26 (0.89,1.77)
Deep vein thrombosis	5	0.28 (0.11,0.61)	13	0.48 (0.26,0.80)	1.75 (0.62,4.96)	7	0.41 (0.19,0.78)	9	0.38 (0.20,0.72)	0.94 (0.34,2.56)
Cardiovascular mortality	153	7.91 (6.72,9.23)	187	7.07 (6.10,8.13)	0.89 (0.71,1.12)	131	7.07 (5.93,8.35)	132	5.59 (4.70,6.61)	0.79 (0.60,1.03)
Chronic pulmonary disease	34	1.82 (1.26,2.48)	41	1.63 (1.20,2.20)	0.90 (0.55,1.46)	17	0.94 (0.55,1.45)	50	2.18 (1.64,2.86)	2.32 (1.28,4.23)
Acute respiratory distress syndrome	38	2.01 (1.45,2.73)	61	2.33 (1.79,2.97)	1.16 (0.75,1.80)	29	1.59 (1.11,2.28)	44	1.90 (1.40,2.53)	1.20 (0.73,1.98)
Seizure	21	1.10 (0.72,1.68)	42	1.62 (1.20,2.19)	1.46 (0.82,2.61)	22	1.22 (0.80,1.82)	45	1.95 (1.45,2.59)	1.59 (0.91,2.78)
End-stage renal disease	11	0.59 (0.32,1.02)	7	0.28 (0.13,0.55)	0.48 (0.18,1.29)	2	0.08 (0.01,0.30)	6	0.25 (0.09,0.49)	2.97 (0.58,15.12)
Acute kidney injury	32	1.66 (1.15,2.31)	38	1.47 (1.07,2.01)	0.88 (0.53,1.47)	35	1.90 (1.33,2.60)	37	1.57 (1.12,2.14)	0.83 (0.48,1.42)
Pancreatitis	9	0.46 (0.21,0.82)	13	0.48 (0.26,0.80)	1.04 (0.40,2.69)	2	0.16 (0.05,0.54)	17	0.71 (0.42,1.10)	1.72 (0.70,4.26)
All-cause mortality	979	50.76 (47.66,54.02)	1,427	54.04 (51.31,56.92)	1.06 (0.97,1.16)	813	60.82 (56.72,65.08)	1,192	50.31 (47.50,53.21)	0.93 (0.85,1.02)

Incidence rate and hazard ratio of clinical sequelae in COVID-19 and non-COVID-19 cohort after weighting during 0-30, 31-90, 91-180, 181-270,271-365 days of COVID-19 infection (unvaccinated patients with COVID-19 and matched control with the same vaccination status; continued)

Day 271-365					
Clinical sequelae	Non-COVID-19		COVID-19		Hazard Ratio ^a (95% CI)
	Event	Incidence per 1,000 person-years (95% CI)	Event	Incidence per 1,000 person-years (95% CI)	
Major CVD	130	11.03 (9.24,13.03)	134	11.25 (9.44,13.25)	0.92 (0.71,1.19)
Stroke	71	5.59 (4.37,6.97)	68	5.28 (4.12,6.63)	0.86 (0.60,1.24)
Myocardial infarction	40	2.95 (2.12,3.95)	32	2.28 (1.58,3.18)	0.72 (0.44,1.16)
Heart Failure	58	4.37 (3.38,5.64)	75	5.53 (4.40,6.91)	1.14 (0.79,1.65)
Atrial fibrillation	18	1.38 (0.87,2.15)	27	2.02 (1.38,2.91)	1.41 (0.76,2.61)
Coronary Artery Disease	56	4.40 (3.36,5.67)	44	3.30 (2.43,4.39)	0.68 (0.45,1.03)
Deep vein thrombosis	6	0.41 (0.16,0.85)	8	0.59 (0.29,1.12)	1.30 (0.41,4.13)
Cardiovascular mortality	39	2.82 (2.02,3.81)	13	0.92 (0.54,1.58)	0.34 (0.17,0.70)
Chronic pulmonary disease	21	1.58 (1.04,2.41)	22	1.60 (1.02,2.37)	0.90 (0.48,1.72)
Acute respiratory distress syndrome	21	1.57 (1.02,2.36)	19	1.36 (0.82,2.05)	0.78 (0.40,1.50)
Seizure	6	0.41 (0.16,0.86)	14	1.00 (0.55,1.60)	2.20 (0.77,6.29)
End-stage renal disease	5	0.39 (0.16,0.85)	9	0.61 (0.29,1.12)	1.38 (0.40,4.70)
Acute kidney injury	20	1.48 (0.96,2.27)	18	1.32 (0.82,2.05)	0.81 (0.37,1.76)
Pancreatitis	6	0.45 (0.21,0.96)	10	0.68 (0.34,1.22)	1.41 (0.53,3.76)
All-cause mortality	628	45.80 (42.30,49.47)	735	52.23 (48.53,56.08)	1.04 (0.93,1.17)

Note: CI: confidence interval, REF: Reference group; Major CVD: Major Cardiovascular Disease; Composite outcome of stroke, heart failure and coronary artery disease

Hazard Ratio (HR) and 95% Confidence Interval (95% CI) were estimated by Cox regression, HR >1 (or <1) indicates patients with COVID-19 had higher (lower) risk of sequelae compared to the non-COVID-19 control cohort

Minor issues:

1. I could not find in the text what is the eligibility criteria for patients in the cohort.

Thank you for your comment. All individuals with data linkage to electronic medical records of Hong Kong Hospital Authority from January 1st, 2018 to January 23rd, 2023 were eligible for this study. We have included further description on the eligibility criteria for patients in this study.

Method: *“Individuals with data linkage to electronic medical records of Hong Kong Hospital Authority from January 1st, 2018 to January 23rd, 2023 were eligible for this study.”* (Page 7, Lines 142-143)

2. In figure 1 it is written that 187 cases died before the index date. Since cases determined the index date (by date of infection) how could they die before they were infected ?

Thank you for comment and raising the question. We would like to clarify that the 187 number of cases who died prior to index date referred to patients have died prior to a positive record of COVID-19 test was recorded. As the data used in this study were obtained from various sources, such impossible cases could arise due to the misalignment of data entry from the different data sources. These subjects were therefore excluded from our final study population to ensure data quality control. We have revised the wording used in the Figure 1 to avoid any potential confusion by the readers.

Figure 1: *“Died before record of positive SARS-CoV-2 infection (N=187)”*

Reviewer #3 (Remarks to the Author):

I would like to thank the authors for their detailed replies and revisions to the manuscript and for their consideration of my and the other reviewers' comments. The additional clarifications helped to understand their analytical approach, and the justifications for their choices appear sensible to me. I especially also appreciate the conduct of further sensitivity analyses and presenting results for incompletely vaccinated individuals, which help to further contextualize their results.

Of course, it is unfortunate that further potentially important confounding variables (e.g. socioeconomic status, smoking, obesity) could not be adjusted for. But as I understand there is not much that can be done about it, and the consistency in results after taking healthcare use into account in IPTW increases the confidence in the results. Meanwhile, it is good that the authors clearly outline this limitation in the discussion.

While the authors have sufficiently addressed my primary concerns in this revision, I would still like to ask them to consider a few changes and clarifications in the manuscript:

1. Matching: Please clarify in the manuscript that you selected all individuals without positive test and with the same birth-year and sex which were available in the dataset as matched controls, and assigned to them the (pseudo-)index date from a randomly selected

corresponding case. The current wording does not make the approach sufficiently clear in my view.

Thank you for your suggestion. We have revised the relevant section in the manuscript based on your suggestion to clarify the methodology in the matching process.

Method: *“All individuals without a record of positive test record of the same birth-year and sex were selected as matched controls.”* (Page 7, Lines 149-150)

“The identical index date was assigned to randomly selected corresponding matched controls as the pseudo-index date.” (Page 8, Lines 154-155)

2. Population at risk for individual outcomes: Thank you for adding a corresponding statement in the first paragraph of the results. However, based on your replies I understood that the population at risk is different for each outcome (since individuals with the outcome prior to infection are excluded), so it remains unclear for me which outcome this statement relates to. If this information is included, it may be most valuable to include an additional column for the population at risk for each of the observation periods, outcomes, and groups. However, I would also agree with omitting this information if this renders the Tables too large and is deemed acceptable by the Editorial Board.

Thank you for your comments. The number of patients at risk reported for the individual time interval refers to the total number subjects who were not censored in the preceding time interval and were continued to be followed up at the start of the corresponding interval. We acknowledged that the report of the number of patients at risk for separate outcome would further enhance the clarity on our results. However, it was thought that such information carries scant relevance in interpreting the findings of this study, we have opted to report the total number of patients at risk at the separate observation windows currently included in the manuscript.

Result: *“The number of patients at risk at each observation period between day 0-30, 31-90, 91-180, 181-270, 271-365 were 2,815,023, 2,801,396, 2,764,243, 2,119,507 and 1,876,858, respectively.”* (Page 10, Lines 221-223)

3. Analytical approach for cumulative risk & hazard ratios: Thank you also for outlining your analytical approach for different observation periods more clearly. With the chosen approach, it needs to be considered that the analysis may be subject to a built-in selection bias (see e.g. Hernan 2010, <https://doi.org/10.1097/EDE.0b013e3181c1ea43>; Bartlett 2020, <https://doi.org/10.1080/19466315.2020.1755722>). For each observation period, the analysis is conditioned on individuals being event-free up to that point ("survivors"). This makes the results difficult to interpret. There are some alternative analytical approaches that aim to overcome this issue. Meanwhile, given that the findings are consistent with the existing literature, I consider it unlikely that the chosen approach would bias the results in a way that it would majorly change the main conclusions. Nevertheless, I suggest that you either consider an alternative analytical approach as outlined in the sources above (at least in a sensitivity analysis), or clearly state this limitation and its potential implications in the corresponding section in the discussion.

Thank you for your comments and raising the potential built-in selection bias associated with the period-specific Hazard Ratio reported. We acknowledged that our current study design which involve the censoring of subjects who have developed specific outcome of interest within a given observation window would inevitably result in a degree of selection bias due to the existence of

ubiquitous factors which could be systematically different in their distribution between survivors from separate cohorts towards the later observation windows as described by Hernan and Bartlett [1,2] which could contribute to the reduction of magnitude of hazard estimated between two study cohorts observed in later observation windows.

With consideration of the potential selection bias discussed and your suggestion, we have discussed such potential limitation and implication on the findings in full in the discussion of the revised manuscript and suggested future research direction to evaluate the effect of such selection bias on the findings reported in this current study and other existing literatures.

Discussion: *“Thirdly, the estimation of period-specific HR is subjected to possible built-in selection bias from the censoring of patients upon the incident of clinical sequelae and the systematic difference in distribution of unknown ubiquitous factors between survivors from separate cohorts exist generally especially towards the later stage of follow-up. This could contribute to the reduction in the magnitude of HR estimated in the later observation windows. Further study is warranted to evaluate the effect of the built-in selection bias described.”* (Page 16, Lines 357-366)

References:

[1] Hernán, M. A. The Hazards of Hazard Ratios. *Epidemiology* 21, 13-15 (2010).

[2] Bartlett, J. W. et al. The Hazards of Period Specific and Weighted Hazard Ratios. *Statistics in Biopharmaceutical Research* 12, 518-519 (2020).

4. Results, line 215: Please adjust to "The SMDs of all baseline characteristics after weighting were less than 0.1, [...]".

Thank you for your suggestion. We have revised the relevant wording according to your suggestion.

Results: *“The SMDs of all baseline characteristics after weighting were less than 0.1, indicating a good balance between groups of patients with different vaccination status.”* (Page 10, Lines 219-221)

We sincerely hope that you will consider this revised manuscript favourably. Should there be further corrections or clarifications needed, we are more than happy to make the necessary changes and provide additional information. We look forward to hearing from you soon.

Yours sincerely,

Ian C. K. Wong, PhD, MSc, B.Pharm
Professor and Head, Department of Pharmacology and Pharmacy
Li Ka Shing Faculty of Medicine
The University of Hong Kong
2/F, Li Ka Shing Faculty of Medicine, Laboratory Block, Faculty of Medicine Building
21 Sassoon Road, Pokfulam, Hong Kong

REVIEWERS' COMMENTS

Reviewer #2 (Remarks to the Author):

I appreciate the effort done by the authors regarding the analysis of potential testing by the use of negative control.

My concern was a testing bias toward patients attending hospital and clinics and therefore bias toward higher detection of covid19 within patients with elevated risk for health outcomes as noted in my last review (major issue 1), analyzing negative control outcomes was a way to reveal if such bias exists.

The authors applied the analysis using 3 negative control outcomes (Lung Cancer , Brain Cancer , Lymphoma) and showed the results in a table on the previous review.

To the best of my understanding the results show that a testing bias exists, for example Lung Cancer and Lymphoma showed in unvaccinated group at window 0-30 days hazard ratios of 1.77 (1.08,2.90) , 2.81 (1.46,5.41) respectively and hazard ratios significantly higher than 1 are observed in other groups and periods.

Considering the long latent period for cancer development (several years), the authors' finding that covid was associated with immediate increased risk for the tested cancers types, is evident of a bias. The bias may stem from 2 possible reasons: detection bias (more testing for presence of CVD/cancer/etc. among infected vs among uninfected) or selection bias (more testing for covid among those with prevalent comorbidities, i.e. reverse causality).The text added by the authors to the discussion only lists one of the reasons and should list both.

While this testing bias effect seems lower than the long covid associated effects shown in Table 2, I do not think it is legitimate to not show these results and discuss them.

The existence of bias does not mean that the research is not publishable , it is almost impossible to expect a clean analysis on the Omicron variant with its widespread and the question of the vaccine performance is important. Analyzing the presence of bias and quantifying it strengthens the reliability of the results.

I think the results for the negative outcome analysis (eg. the table shown in the last author response) should be a part of the manuscript (at least as a supplementary table) and should be mentioned in the result section and the discussion.

If the authors think that the specific outcomes chosen suffer limitations (such as low numbers of cases) I may suggest taking other conditions (such as any cancer or fracture) as negative controls.

Additionally the authors may consider using statistical methods for effect estimates after accounting for their observed bias.

RE: NCOMMS-23-39376B– Persistence in risk and effect of COVID-19 vaccination on long-term health consequences after SARS-CoV-2 infection

On behalf of the co-authors, I would like to thank the editors and the reviewer for their further comments and suggestions. In response, we have made the necessary adjustments in the revised manuscript, and hope that you will now find it suitable for publication. We provide here, our point-to-point response to the earlier version of this paper reviewed.

REVIEWER COMMENTS

Reviewer #2 (Remarks to the Author):

I appreciate the effort done by the authors regarding the analysis of potential testing by the use of negative control.

My concern was a testing bias toward patients attending hospital and clinics and therefore bias toward higher detection of covid19 within patients with elevated risk for health outcomes as noted in my last review (major issue 1), analyzing negative control outcomes was a way to reveal if such bias exists.

The authors applied the analysis using 3 negative control outcomes (Lung Cancer , Brain Cancer , Lymphoma) and showed the results in a table on the previous review. To the best of my understanding the results show that a testing bias exists, for example Lung Cancer and Lymphoma showed in unvaccinated group at window 0-30 days hazard ratios of 1.77 (1.08,2.90) , 2.81 (1.46,5.41) respectively and hazard ratios significantly higher than 1 are observed in other groups and periods.

Considering the long latent period for cancer development (several years), the authors' finding that covid was associated with immediate increased risk for the tested cancers types, is evident of a bias. The bias may stem from 2 possible reasons: detection bias (more testing for presence of CVD/cancer/etc. among infected vs among uninfected) or selection bias (more testing for covid among those with prevalent comorbidities, i.e. reverse causality).The text added by the authors to the discussion only lists one of the reasons and should list both.

While this testing bias effect seems lower than the long covid associated effects shown in Table 2, I do not think it is legitimate to not show these results and discuss them.

The existence of bias does not mean that the research is not publishable , it is almost impossible to expect a clean analysis on the Omicron variant with its widespread and the question of the vaccine performance is important. Analyzing the presence of bias and quantifying it strengthens the reliability of the results.

I think the results for the negative outcome analysis (eg. the table shown in the last author response) should be a part of the manuscript (at least as a supplementary table) and should be mentioned in the result section and the discussion.

If the authors think that the specific outcomes chosen suffer limitations (such as low numbers of cases) I may suggest taking other conditions (such as any cancer or fracture) as negative controls.

Additionally the authors may consider using statistical methods for effect estimates after accounting for their observed bias.

Thank you for your further comments on the findings on the negative control outcomes and further suggestions on the presentation of the results and discussion of the potential detection and selection bias as indicated from our findings. We agreed that the current findings on the increased incidence of the three types of cancer which are considered to have a prolonged latent period for their development raises the potential of detection bias from the increased healthcare contact in patients following SARS-CoV-2 infection and selection bias arise from the increased SARS-CoV-2 testing amongst individuals with prevalent comorbidities, particular during the acute phase of infection.

Despite the indication of potential detection and selection bias from the increased risk of cancer outcomes during the acute phase of infection, the extent of their implications is likely to differ across study outcomes. Our current analysis focuses on the incidence of disease diagnosis and that a wide range of confounders, including existing morbidities, medication history, previous healthcare visits and vaccination records have been adjusted for in the analysis model employed to minimize the potential of selection bias. The consistent direction of findings in patients of different levels of Charlson comorbidity index, a proxy of their degree of comorbidities, as the findings of main analysis reported in this study further suggested that the potential selection bias incurred may not have a major influence on the overall conclusion of this study.

We understand that the use of statistical modelling, such as bias factor derived from the effect estimates of the chosen negative control cancer outcome, to accurately account for such observed biases would require further evaluation and adjustment of specific confounding factors associated with cancers which are considered to differ greatly from the clinical sequelae evaluated as the main analysis of this study. Due to the lack of specific relevant data in our current data source, such analysis may not be considered as part of this current study.

Taking your suggestions and the discussion above into consideration, we have mentioned findings from the negative control analysis in the results section and presented the results in full in Supplementary Table 12 of this manuscript. The potential of both the detection and selection bias along with their implication has been acknowledged and discussed in detail in our revised manuscript.

Results: *“A moderate increase in risk of lung cancer and lymphoma was observed during the first 30 days of infection. (Supplementary Tables 12)”* (Page 7, Lines 150-152)

Discussion: *“Firstly, detection bias might be inherent in this study due to the potential under-reporting of existing underlying conditions prior to a diagnosis of SARS-CoV-2 infection. In addition, the increased healthcare contacts from receiving further examination amongst patients diagnosed with SARS-CoV-2 infection could result in the increased diagnosis of condition which*

could have persisted prior to infection. For instance, patients presenting with sequelae may have developed certain diseases prior to their diagnosis of SARS-CoV-2 infection; yet they did not receive a diagnosis for those conditions until a confirmed diagnosis of SARS-CoV-2 infection, resulting in the over-attribution of disease diagnosis as post-infection sequelae.” (Page 10-11, Lines 235-242)

Discussion: “Secondly, *potential selection bias may also arise from the increased SARS-CoV-2 testing amongst individuals with prevalent comorbidities.*” (Page 11, Lines 250-251)

Method: “*Lung cancer, brain cancer and lymphoma which were considered to have a prolonged latent period for their development were included as negative control outcomes to detect possible testing bias.*” (Page 15, Lines 348-350)

We sincerely hope that you will consider this revised manuscript favourably. Should there be further corrections or clarifications needed, we are more than happy to make the necessary changes and provide additional information. We look forward to hearing from you soon.

Yours sincerely,

Ian C. K. Wong, PhD, MSc, B.Pharm
Professor and Head, Department of Pharmacology and Pharmacy
Li Ka Shing Faculty of Medicine
The University of Hong Kong
2/F, Li Ka Shing Faculty of Medicine, Laboratory Block, Faculty of Medicine Building
21 Sassoon Road, Pokfulam, Hong Kong